# Understanding the Expressive Power and Mechanisms of Transformer for Sequence Modeling

**Mingze Wang**
School of Mathematical Sciences, Peking University, Beijing, China
`mingzewang@stu.pku.edu.cn`

**Weinan E** †
Center for Machine Learning Research and School of Mathematical Sciences, Peking University, Beijing, China
AI for Science Institute, Beijing, China
`weinan@math.pku.edu.cn`

## Abstract

We conduct a systematic study of the approximation properties of Transformer for sequence modeling with long, sparse and complicated memory. We investigate the mechanisms through which different components of Transformer, such as the dot-product self-attention, positional encoding and feed-forward layer, affect its expressive power, and we study their combined effects through establishing explicit approximation rates. Our study reveals the roles of critical parameters in the Transformer, such as the number of layers and the number of attention heads. These theoretical insights are validated experimentally and offer natural suggestions for alternative architectures.

## 1 Introduction

In recent years, Transformer networks (Vaswani et al., 2017) have emerged as foundational models, setting new benchmarks across various domains, including natural language processing (NLP), computer vision (CV), and protein folding. Despite their impressive practical achievements, the underlying mechanisms and theoretical foundations of Transformer networks remain largely elusive.

Transformer networks encompass various components, posing challenges to their comprehensive understanding. A typical Transformer comprises multiple layers, each consisting of a multi-head self-attention (Attn) sub-layer and a feed-forward network (FFN) sub-layer, integrated with residual blocks. FFN is a two-layer nonlinear network, while Attn includes dot-product (DP) and positional encoding (PE). To get a better understanding of how Transformer works in practice, we need to study several key issues. These include:

**(i)** *How do the key hyper-parameters, for example, the number of layers, the number of Attn heads and the with of FFN layers, affect the performance of the Transformer network?*

**(ii)** *How do the Attn and FFN layers contribute differently to the overall performance?*

**(iii)** *How does DP attention work, and is the DP structure necessary?*

**(iv)** *How efficient is PE in modeling long-range correlations?*

Extensive empirical research on Transformer components has led to the proposal of numerous alternatives to the current structure of Transformer. For example, several relative positional encodings (RPE) (Shaw et al., 2018; Raffel et al., 2020; Su et al., 2024; Press et al., 2022) have been proposed

38th Conference on Neural Information Processing Systems (NeurIPS 2024).

to substitute the original absolute positional encoding (APE), yielding superior performance in challenging tasks like length generalization (Ontanón et al., 2022; Csordás et al., 2021; Anil et al., 2022). Additionally, the necessity of the computationally expensive DP in Attn layers has been widely questioned, and researchers proposed numerous alternatives of DP that show considerable efficacy in specific tasks (Kitaev et al., 2020; Wang et al., 2020; Choromanski et al., 2020; Tay et al., 2021; Allen-Zhu and Li, 2023). Nonetheless, these explorations have not yielded a satisfactory theoretical understanding of the mechanisms of these components.

**In this work**, we investigate the expressive power of Transformer and the underlying mechanisms of its components for sequence modeling. **Our contributions** are summarized as follows:

We categorize three types of sequence modeling tasks with varying complexity, which are relevant to a broad spectrum of application areas. *Task I: Modeling fixed, long but sparse memories*. This is relevant to sparse Boolean functions and the traditional $n$-gram model in NLP. *Task II: Modeling adaptive, long but sparse memories*. This is relevant to multi-step reasoning tasks as well as various NLP tasks such as dependency parsing, sentiment analysis, and continuation writing. *Task III: Modeling essentially sparse memories*. Examples include feature representation in CV and wavelet analysis in classical signal processing.

For these sequence modeling tasks, we theoretically investigate the expressive power of Transformer and its variants, establishing explicit approximation rates. Our meticulous analysis provides *theoretical insights* into the underlying mechanisms of Transformer components. Specifically,

- **The distinct roles of the number of layers, the number of Attn heads, and the width of FFN layers.** Deeper Transformer are capable of handling memories with more intricate interrelationships, such as nested relationships (Thm 4.4). In contrast, for memories lacking such interrelationships, single-layer Transformer with sufficient number of Attn heads and FFN width should suffice (Thm 4.1). This is quite intuitive: If the content of the next token relies on a few previous tokens in an independent way, we can treat each such dependence by a separate attention head. There is no need for many layers. Additionally, increasing the depth can also alleviate the reliance on the number of heads and width (Prop 4.5).

- **The different roles of Attn layers and FFN layers.** Our results consistently suggest that: FFN layers are tasked with approximating nonlinear memory functions and the readout function, while Attn layers are responsible for extracting the tokens from these memory locations.

- **The functionality and necessity of DP.** For the relatively simple Task I, DP is not necessary and can be omitted (Thm 3.1). However, for the more complex Task II, the cooperation between DP and RPE provides the needed interaction between the temporal space and the token space, crucial for the extraction of adaptive memories (Thm 4.1 and 4.4). Additionally, for Task II, while the nonlinearity provided by DP is necessary (Prop 4.2), a computationally efficient alternative to DP exists, as we show in Prop 4.3.

- **The efficiency of RPE in modeling long-range correlations.** Our results consistently suggest that the primary role of RPE is to approximate the memory kernels. Specifically, for Task III, we demonstrate that Transformer with suitable RPE can handle heavy-tailed memories, thus overcoming the Curse of Memory faced by recurrent neural networks (Thm 5.1). Moreover, our findings give theoretical support to the choice of RPE in practice.

Finally, we conduct experiments to validate our theoretical insights.

## 2 Preliminaries

**Basic notations.** We use bold-faced letters for vectors or matrices and lowercase letters for scalars, e.g. $\boldsymbol{x} = (x_1, \cdots, x_d)^\top \in \mathbb{R}^d$ and $\boldsymbol{W} = (W_{ij})_{m \times n} \in \mathbb{R}^{m \times n}$. The standard Euclidean inner product between two vectors is denoted by $\langle \cdot, \cdot \rangle$, and the $l_p$ norm of a vector is represented by $\|\cdot\|_p$.

We employ standard big-O notations $\mathcal{O}, \Omega, \Theta$ to hide absolute positive constants and use $\tilde{\mathcal{O}}, \tilde{\Omega}, \tilde{\Theta}$ to further hide logarithmic constants. For any positive integer $n$, let $[n] = \{1, \cdots, n\}$. Denote by $\mathbb{I}\{E\}$ the indicator function for an event $E$. Denote by $a \vee b = \max\{a, b\}$ for real number $a, b$.

## 2.1 Sequence modeling with long but sparse memories

**Sequence modeling.** For convenience, we consider input sequences of infinite length ($t \in \mathbb{Z}$). It is important to note, however, that our theoretical framework can be adapted to finite-length input sequences by masking distant tokens. Formally, the output sequence $\boldsymbol{Y} = (\boldsymbol{y}_t)_{t \in \mathbb{Z}} \in \mathbb{R}^{c \times \mathbb{Z}}$ is generated from the input sequence $\boldsymbol{X} = (\boldsymbol{x}_t)_{t \in \mathbb{Z}} \in \mathcal{X} \subset \mathbb{R}^{d \times \mathbb{Z}}$ via an unknown mapping $\mathbf{H}.(\cdot)$ dependent on the input sequence up to the prediction time, and this can be expressed as:

$$\boldsymbol{y}_t = \mathbf{H}_t(\boldsymbol{X}) = \boldsymbol{f}(\boldsymbol{x}_t, \boldsymbol{x}_{t-1}, \boldsymbol{x}_{t-2}, \cdots), \quad t \in \mathbb{Z}. \tag{1}$$

Our objective is to learn the mapping $\mathbf{H}.(\cdot)$. Additionally, we define the norm $\|\|\mathbf{H}\|\| := \sup_{t \in \mathbb{Z}} \sup_{\boldsymbol{X} \in \mathcal{X}} \|\mathbf{H}_t(\boldsymbol{X})\|$. Without loss of generality, we assume $\|\boldsymbol{x}_t\|_2 \leq 1$ for any $\boldsymbol{X} \in \mathcal{X}$ and set the output dimension $c = 1$ for simplicity.

**Long but sparse memories.** To model such sequences, we define three types of memories: fixed, long but sparse memories; adaptive, long but sparse memories; and essentially sparse memories. These memory types are prevalent in sequence modeling tasks across diverse domains such as NLP, CV, signal processing, and sparse function representation. In Section 3, 4, and 5, we will formally define these different types and investigate Transformer's capacity to model them.

## 2.2 Transformer architecture

**Transformer network.** Transformer (Vaswani et al., 2017) is a network architecture designed for processing sequences and generating predictions. Given an input sequence $\boldsymbol{X}$, Transformer executes the following steps. Initially, each $d$-dimensional (dim) input token is transformed into a $D$-dim vector through an embedding mapping such as $\boldsymbol{x}_t^{(0)} = \boldsymbol{W}_E \boldsymbol{x}_t + \boldsymbol{b}_E$, where $\boldsymbol{W}_E \in \mathbb{R}^{D \times d}, \boldsymbol{b}_E \in \mathbb{R}^D$. Subsequently, a typical $L$-layer Transformer with residual block operates according to the formulation:

$$\begin{aligned} \boldsymbol{X}^{(l-\frac{1}{2})} &= \boldsymbol{X}^{(l-1)} + \mathbf{Attn}^{(l)}(\boldsymbol{X}^{(l-1)}), \quad l \in [L]; \\ \boldsymbol{X}^{(l)} &= \boldsymbol{X}^{(l-\frac{1}{2})} + \mathbf{FFN}^{(l)}(\boldsymbol{X}^{(l-\frac{1}{2})}), \quad l \in [L]. \end{aligned} \tag{2}$$

At the $l$-th layer, $\mathbf{FFN}^{(l)}(\cdot)$ denotes a standard (point-wise) two-layer ReLU networks with $m$ neurons: for a given input $\boldsymbol{x} \in \mathbb{R}^D$, $\mathbf{FFN}^{(l)}(\boldsymbol{x}) = \sum_{k=1}^m \boldsymbol{a}_k^{(l)} \sigma(\boldsymbol{b}_k^{(l)\top} \boldsymbol{x} + c_k^{(l)})$, where $\sigma(\cdot)$ is the activation function such as ReLU. Additionally, in the final ($L$-th) FFN layer, the residual block is omitted, commonly referred to as the readout function. Moreover, $\mathbf{Attn}^{(l)}(\cdot)$ refers to a multi-head self-attention, as elaborated below.

**Multi-head self-attention.** Our focus lies on standard dot-product Attn, denoted as $\mathbf{Attn}^{(l)}(\cdot)$ and consisting of $H$ heads. When applied to an input sequence $\boldsymbol{X}$, Attn operates as follows:

$$\mathbf{Attn}^{(l)}(\boldsymbol{X}) = \boldsymbol{W}_O^{(l)} \sum_{h=1}^H \boldsymbol{W}_V^{(l,h)} \boldsymbol{X} \operatorname{softmax}_c \left( \left\langle \boldsymbol{W}_Q^{(l,h)} \boldsymbol{X}, \boldsymbol{W}_K^{(l,h)} \boldsymbol{X} \right\rangle + \boldsymbol{R}^{(l,h)} \right). \tag{3}$$

Here, the parameters $\boldsymbol{W}_Q^{(l,h)}, \boldsymbol{W}_K^{(l,h)}, \boldsymbol{W}_V^{(l,h)}, \boldsymbol{W}_O^{(l,h)}$ correspond to the query, key, value, output matrices of the $(l, h)$-th head, respectively. $\operatorname{softmax}_c$ represents taking softmax normalization across column. Furthermore, $\boldsymbol{R}^{(l,h)} \in \mathbb{R}^{\mathbb{Z} \times \mathbb{Z}}$ denotes the relative positional encoding matrix, which satisfies $R_{t,s}^{(l,h)} = -\infty$ for $t < s$ in the next-token prediction paradigm. Consequently, the $t$-th output of Attn is expressed as:

$$\mathbf{Attn}_t^{(l)}(\boldsymbol{X}) = \boldsymbol{W}_O^{(l)} \sum_{h=1}^H \sum_{s=0}^{+\infty} \frac{\boldsymbol{W}_V^{(l,h)} \boldsymbol{x}_{t-s} \exp\left( \left\langle \boldsymbol{W}_Q^{(l,h)} \boldsymbol{x}_t, \boldsymbol{W}_K^{(l,h)} \boldsymbol{x}_{t-s} \right\rangle + R_{t,t-s}^{(l,h)} \right)}{\sum_{j=0}^{+\infty} \exp\left( \left\langle \boldsymbol{W}_Q^{(l,h)} \boldsymbol{x}_t, \boldsymbol{W}_K^{(l,h)} \boldsymbol{x}_{t-j} \right\rangle + R_{t,t-j}^{(l,h)} \right)}.$$

**Logarithmic and Power relative positional encoding.** As highlighted in Section A, among various types of RPEs, the RPEs used in T5 and KERPLE(log) demonstrate superior performance over Alibi, significantly outperforming other RPEs and APEs in the length generalization task (Kazemnejad et al., 2023; Chi et al., 2022). This finding motivates our focus on the T5-type, KERPLE(log), and Alibi-type RPEs throughout this paper. All of these RPE matrices are Toeplitz, with the form of

$R_{t,s} = r(t-s)$. Notably, for T5 and KERPLE(log), $r(t-s)$ undergoes an initial linear decrease followed by a logarithmic decrease as the relative distance $t-s$ increases (Please refer to Section G.1 for more details). In contrast, for Alibi, $r(t-s)$ decreases linearly. Inspired by these discussions, we examine the following RPEs with different decay rates:

$$\phi_{\log}(z) = \begin{cases} -\log z, & z \geq 1 \\ -\infty, & \text{otherwise} \end{cases}; \quad \phi_{\text{lin}}(z) = \begin{cases} -z, & z \geq 0 \\ -\infty, & \text{otherwise} \end{cases}.$$

We will study Transformer with $\phi_{\text{type}}$ RPE ($\text{type} \in \{\log, \text{lin}\}$). Specifically, the RPE in the $(l, h)$-th head (3) is as follows:

$$R_{t,s}^{(l,h)} := p^{(l,h)} \phi_{\text{type}}(t-s), \tag{4}$$

where $p^{(l,h)} \in \mathbb{R}_+$ is a trainable parameter.

**Remark 2.1.** For standard Transformer (2) incorporating Attn (3) with RPE (4), the parameters are: the embedding matrix $\boldsymbol{W}_E$; $\boldsymbol{a}_k^{(l)}, \boldsymbol{b}_k^{(l)}, c_k^{(l)}$ in the FFN layers; $\boldsymbol{W}_Q^{(l,h)}, \boldsymbol{W}_K^{(l,h)}, \boldsymbol{W}_V^{(l,h)}, p^{(l,h)}, \boldsymbol{W}_O^{(l)}$ in the Attn layers. Notably, the number of parameters is independent of the sequence length, thus enabling the model to handle input sequences of arbitrary length.

**Remark 2.2.** In the subsequent sections, we will analyze Transformer and its variants. For the sake of brevity, some shorthand notations are introduced here. For examples, Transformer (2) using $\phi_{\log}/\phi_{\text{lin}}$ RPE (4) is referred to as "Transformer with log/lin-RPE"; Transformer with $\boldsymbol{W}_Q^{(l,h)}, \boldsymbol{W}_K^{(l,h)} = \boldsymbol{0}$ is called "dot-product-free Transformer".

## 2.3 Expressive power via approximation theory

This paper delves into the expressive power of Transformer through the lens of approximation theory, with a specific focus on establishing explicit approximation rates for Transformers in modeling long but sparse memories.

**Approximation rates v.s. universal approximation.** In approximation theory, results are generally categorized into two types: universal approximation (density-type) and approximation rates (Jackson-type) (Jackson, 1930). Universal approximation investigates whether the hypothesis class is dense in the target class. Although this property is fundamental, it does not offer detailed insights into approximation efficiency. In contrast, approximation rates go deeper, emphasizing the efficiency of the approximation. A typical example within this framework is the approximation theory of two-layer neural networks (2NNs).

**Barron space of 2NNs.** The well-known universal approximation result for 2NNs asserts that 2NNs can approximate any continuous function (Barron, 1992; 1993; 1994). Nonetheless, this result lacks a characterization of the approximation efficiency, i.e., how many neurons are needed to achieve a certain approximation accuracy? This gap was addressed by the Barron space theory (E et al., 2019; 2021; Ma et al., 2020). It is established that for any function within Barron space $f \in \mathcal{B}$ (Appendix G.2), 2NNs with $m$ neurons (denoted by $\mathcal{H}_m$) can approximate them efficiently, at a rate of $\inf_{f_m \in \mathcal{H}_m} \|f - f_m\| \leq \mathcal{O}(\|f\|_{\mathcal{B}} / \sqrt{m})$, remarkably independent of the input dimension $d$, thus avoiding the *Curse of Dimensionality* (Bellman, 1966; Bach, 2017).

# 3 Fixed, long but $M$-sparse memories

## 3.1 Problem formulation

**Fixed, long but $M$-sparse memories.** In this section, we investigate a fundamental category of long but sparse memories. Our focus is on scenarios where the positions of the sparse memories remain fixed and are independent of the tokens. The target function is represented by:

$$y_t = f(\boldsymbol{x}_t, \boldsymbol{x}_{t-T_1}, \cdots, \boldsymbol{x}_{t-T_M}), \tag{5}$$

where $1 \leq T_1 < \cdots < T_M < +\infty$ signify the fixed positions of the memories. Despite the memories being fixed (token-independent) and sparse (finite $M$), the task can still be complex due to the potentially long-range memories ($T_1, \cdots, T_M$ can be large enough).

**Examples.** (I) For Boolean inputs, (5) aligns with *sparse Boolean functions*, also studied in (Edelman et al., 2022; Bhattamishra et al., 2022). Notably, Bhattamishra et al. (2022) observed that Transformers outperform LSTMs in learning sparse parities. (II) Selecting the simplest case of $T_i = i$ in (5) corresponds to the traditional $n$-*gram model*, which consists of short and sparse memories.

**Target class.** We focus on target functions described in (5). The readout function $f$ is considered within the standard Barron space $\mathcal{B}$, i.e., which can be effectively approximated by 2NNs. Moreover, we assume that $f$ is Lipschitz, denoted by $f \in \mathcal{L}$. Thus, we can focus more on investigating the memory extraction power of Transformer. Formally, we define the target class for modeling fixed, long but $M$-sparse memories as:

$$\mathcal{H}^{\text{Fix}} := \big\{ \mathbf{H} : \ \mathbf{H}_t(\boldsymbol{X}) = (5), \text{ where } 1 \leq T_1 < \cdots < T_M < +\infty, f \in \mathcal{B} \cap \mathcal{L} \big\}. \tag{6}$$

**Transformer hypothesis class.** As mentioned in Section 1, one of our main aims is to study the necessity and roles of different components in Transformer, such as DP and RPE. This section focuses on the "simplest" one-layer Transformer and investigates whether it can effectively model this task. Formally, our hypothesis class includes all one-layer *DP-free* Transformers, configured with $H$ Attn heads and FFN width $m$:

$$\mathcal{TF}_{(1,H,m)}^{\text{DPF,type}} := \big\{ \mathbf{TF} : \mathbf{TF} \text{ is a 1-layer, } H\text{-head, } m\text{-width}$$
$$\text{dot-product-free Transformer with type-RPE} \big\}. \tag{7}$$

## 3.2 Theoretical results and insights

**Theorem 3.1** (Approximation rate). *For any target* $\mathbf{H} \in \mathcal{H}^{\text{Fix}}$ *(6), rate* $n \in \mathbb{N}_+$, *and* $H, m \in \mathbb{N}_+$, *there exists a* 1*-layer Transformer* $\mathbf{TF} \in \mathcal{TF}_{(1,H,m)}^{\text{DPF,type}}$ *(7) and a constant* $C(n)$ *such that*

$$\||\mathbf{H} - \mathbf{TF}\|| \leq \mathcal{E}_{\text{FFN}} + \|f\|_{\text{Lip}} \, \mathcal{E}_{\text{Attn}}(\texttt{type}),$$

*where* $\mathcal{E}_{\text{FFN}} = \tilde{\mathcal{O}}\left( \frac{\|f\|_{\mathcal{B}}}{\sqrt{m}} \right)$ *and* $\mathcal{E}_{\text{Attn}}(\texttt{type}) = \begin{cases} \mathcal{O}\left( \frac{C(n)}{H^n} \left( \sum_{i=1}^{M} e^{0.01 T_i} \right)^{n+1} \right), \texttt{type} = \texttt{lin} \\ \mathcal{O}\left( \frac{C(n)}{H^n} \left( \sum_{i=1}^{M} T_i^{1.01} \right)^{n+1} \right), \texttt{type} = \texttt{log} \end{cases}$.

Theorem 3.1 establishes the *approximation rate* of one-layer DP-free Transformer for modeling fixed, long but sparse memories. Here, the model complexity is governed by the number of Attn heads $H$ and the width of FFN layers $m$, while the target complexity arises from the lengths of the memories $T_1, \cdots, T_M$ and the complexity of the readout function $f$. The approximation error comprises two components: the error in the FFN component $\mathcal{E}_{\text{FFN}}$ and the error in the Attn component $\mathcal{E}_{\text{Attn}}(\texttt{type})$. The error $\mathcal{E}_{\text{FFN}}$ aligns with classical results, showcasing its effectiveness in approximating Barron functions. On the other hand, $\mathcal{E}_{\text{Attn}}(\texttt{type})$ hinges on the capacity of the Attn block for modeling long-range memories. Specifically, with increasing memory length, the necessary number of Attn heads grows at a small exponential rate for lin-RPE and at a polynomial rate for log-RPE.

The proof of Theorem 3.1 is deferred to Appendix B. We can draw some insights from Theorem 3.1 and its proof.

**Different roles of the Attn layer and the FFN layer.** The Attn and FFN layers fulfill distinct roles in this task. Specifically, the FFN layer efficiently approximates the nonlinear readout function $f$, while the Attn layer is responsible for extracting the token $\boldsymbol{x}_{t-T_i}$ by approximating the memory kernel $\mathbb{I}\{\cdot = T_i\}$. These components together enable effective modeling of fixed, long, but sparse memories.

**Non-necessity of DP.** Theorem 3.1 suggests that the DP component in Attn is not necessary and can be omitted for modeling fixed, long but sparse memories. This is due to the relative simplicity of modeling fixed memory kernels. In a more complex scenario in Section 4, the role of the dot-product becomes important. In contrast to Edelman et al. (2022), which utilizes the property of DP to prove that Transformer can model sparse Boolean functions, our result reveals that one-layer Transformer can successfully tackle the same task *even without the dot product in the attention layer*.

**Effect of RPE types on expressivity.** Our result indicates that the type of the RPE used in the Attn layer subtly influences the Transformer's ability to model long-range memories. As the range of the memory increases, the required head number grows at a slightly exponential rate for lin-RPE

and at a polynomial rate for $\log$-RPE. The subtle difference is attributed to the relative simplicity of approximating the memory kernel $\mathbb{I}\{\cdot = T_i\}$. We will explore a more complex task in Section 5, where the impact of different types of RPE becomes even more pronounced.

## 4 $K$-Adaptive, long but $M$-sparse memories

### 4.1 Problem formulation

In this section, we delve into a more complex modeling scenario closely aligned with typical language processing tasks.

$K$**-Adaptive, long but** $M$**-sparse memories.** This section investigates the scenario where the positions of the sparse memories are "adaptive", meaning they depend on the input tokens. The target function is formulated as:

$$y_t = f(\boldsymbol{x}_t, \boldsymbol{x}_{t-t_1}, \cdots, \boldsymbol{x}_{t-t_M}), \tag{8}$$

where the positions of the memory tokens $t_1, \cdots, t_M$ follow a nested relationship:

$$t_1 = g_1(\boldsymbol{x}_t); t_2 = g_2(\boldsymbol{x}_t, \boldsymbol{x}_{t-t_1}); \cdots; t_{K+1} = g_{K+1}(\boldsymbol{x}_t, \boldsymbol{x}_{t-t_1}, \cdots, \boldsymbol{x}_{t-t_K});$$
$$\cdots; t_M = g_M(\boldsymbol{x}_t, \boldsymbol{x}_{t-t_1}, \cdots, \boldsymbol{x}_{t-t_K}).$$

Here, $M$ denotes the number of memory tokens, and $K$ measures the nesting complexity in the memory structure. We assume that memory functions $g_i$ generate positive integers for the input tokens, and there exist maximum values $T_i$ such that $g_i \leq T_i$. In this adaptive framework, each position of the memory token depends on multiple input tokens and is nested within other memory structures, leading to potential influence of later memory tokens by the earlier ones.

To facilitate understanding, we first consider a warm-up case, i.e., $K = 0$ in (8). In this case, the positions of memories only depend on the current token, without interaction with each other. It can be represented as:

$$y_t = f(\boldsymbol{x}_t, \boldsymbol{x}_{t-t_1}, \cdots, \boldsymbol{x}_{t-t_M}), \tag{9}$$

where $t_i = g(\boldsymbol{x}_i), i \in [M]$.

**Target class.** The target classes for modeling adaptive, long but sparse memories in both warm-up and general cases are as follows:

$$\mathcal{H}^{\text{Adap}}_{(1,M)} := \left\{ \mathbf{H} : \ \mathbf{H}_t(\boldsymbol{X}) = (9), \ \text{where} \ g_i \in \mathcal{B}, 1 \leq g_i \leq T_i, i \in [M]; f \in \mathcal{B} \cap \mathcal{L} \right\}. \tag{10}$$

$$\mathcal{H}^{\text{Adap}}_{(K,M)} := \left\{ \mathbf{H} : \ \mathbf{H}_t(\boldsymbol{X}) = (8), \ \text{where} \ g_i \in \mathcal{B}, 1 \leq g_i \leq T_i, i \in [M]; f \in \mathcal{B} \cap \mathcal{L} \right\}. \tag{11}$$

**Examples.** Adaptive memories are commonly encountered in practical scenarios. (I) *Adaptive sparse Boolean functions*, e.g., $y_t = x_t \cdot x_{t-g(x_t)} \cdot x_{t-g(x_{t-g(x_t)})}$, where $\boldsymbol{X} \in \{\pm 1\}^{\mathbb{Z}}$, $g(x) = 1$ for $x = 1$ and $g(x) = 2$ for $x = -1$. This fits within our framework (8) with $K = M = 2$. (II) *Multi-step reasoning*, e.g., modeling the $K$-adaptive, long, but $K$-sparse memories contains a complicated $K$-step reasoning task, which require the sequential search following the rule $((\cdots((x_t \mapsto x_{t-t_1}) \mapsto x_{t-t_2} \cdots) \mapsto x_{t-t_{K-1}}) \mapsto x_{t-t_K}$. (III) In *NLP tasks* like dependency parsing, part-of-speech tagging, sentiment analysis, or continuation writing, the positions of relevant prefix tokens usually depend on the context itself, and can vary depending the content. Additionally, the nested structure is a fundamental characteristic of natural language (Hawkins, 2021).

**Transformer hypothesis class.** Some previous works Yun et al. (2019); Kim et al. (2022) treated the softmax with normalization as an approximation of hardmax, suggesting the potential importance of the normalization. In contrast, in this section, we remove the normalization in the denominator of softmax and investigate its ability for sequence modeling. Additionally, to address the discreteness of time and memory values, we consider Transformer with specific precision, as detailed in Appendix C. The precision technique is widely used in LLM training (Kalamkar et al., 2019), such as BFloat16. Formally, the hypothesis class is defined as follows, encompassing all normalization-free $L$-layer Transformer, configured with $H$ Attn heads and FFN width $m$ and using type-RPE and specific precision.

$$\mathcal{TF}^{\text{type}}_{(L,H,m)} := \big\{ \mathbf{TF} : \mathbf{TF} \text{ is an } L\text{-layer, } H\text{-head, } m\text{-width}$$
$$\text{Transformer with type-RPE and specific precision}\big\}. \tag{12}$$

## 4.2 Theoretical results and insights: The warm-up case

**Theorem 4.1** (Approximation rate, warm-up case). *For any target $\mathbf{H} \in \mathcal{H}_{(1,M)}^{\mathrm{Adap}}$ (8), rate $n \in \mathbb{N}_+$, and $H, m \in \mathbb{N}_+$, there exists a two-layer Transformer $\mathbf{TF} \in \mathcal{TF}_{(2,H,m)}^{\mathtt{type}}$ (12) and a constant $C(n)$ such that: if the width satisfies $m \geq \begin{cases} \tilde{\Omega}\big(\sum_{i=1}^{M} \|g_i\|_{\mathcal{B}}^2\big) & ,\mathtt{type} = \mathrm{lin} \\ \tilde{\Omega}\big(\sum_{i=1}^{M} \|\log g_i\|_{\mathcal{B}}^2 T_i^2\big) & ,\mathtt{type} = \log \end{cases}$, then the following approximation rate holds:*

$$\|\mathbf{H} - \mathbf{TF}\| \leq \mathcal{E}_{\mathrm{FFN}} + \|f\|_{\mathrm{Lip}} \, \mathcal{E}_{\mathrm{Attn}}(\mathtt{type}),$$

*where $\mathcal{E}_{\mathrm{FFN}} = \tilde{\mathcal{O}}\left(\frac{\|f\|_{\mathcal{B}}}{\sqrt{m}}\right)$ and $\mathcal{E}_{\mathrm{Attn}}(\mathtt{type}) = \begin{cases} \mathcal{O}\left(\frac{C(n)}{H^n}\left(\sum_{i=1}^{M} e^{0.01 T_i}\right)^{n+1}\right) & ,\mathtt{type} = \mathrm{lin} \\ \mathcal{O}\left(\frac{C(n)}{H^n}\left(\sum_{i=1}^{M} T_i^{1.01}\right)^{n+1}\right) & ,\mathtt{type} = \log \end{cases}.$*

In Theorem 4.1, we present the *approximation rate* of two-layer Transformer for the warm-up case: modeling 1-adaptive, long but $M$-sparse memories. This theorem reveals that the approximation error comprises two distinct components: the error in the FFN component $\mathcal{E}_{\mathrm{FFN}}$ and the error in the Attn component $\mathcal{E}_{\mathrm{Attn}}(\mathtt{type})$. A critical difference from 3.1 is the presence of the condition related to the width $m$ of FFN layers. This term arises from using the FFN layer to approximate the memory function $g_i$. Owing to the discreteness of memory $g_i$ and the implementation of rounding operations, the approximation within rounding accuracy all achieves zero error after rounding, while it can not get correct rounding beyond this accuracy. In contrast, the error $\mathcal{E}_{\mathrm{FFN}}$ is caused by using FFN to approximate the readout function $f$, the same as $\mathcal{E}_{\mathrm{FFN}}$ in Theorem 3.1.

The proof of Theorem 4.1 can be found in Appendix C.1. Theorem 4.1 and its proof offer several critical insights into the underlying mechanism of Transformer.

**Distinct roles of Attn layers and FFN layers.** Our proof elucidates that the FFN layers are tasked with approximating the readout function $f$ and memory functions $g_i$, while the Attn layers are responsible for the extraction of the adaptive memories. It is essential to clarify the difference between "approximating memory functions" and "memory extraction". The former refers to utilizing some function to estimate the memory function $g_i$, whereas the latter pertains to extracting the token $\boldsymbol{x}_{t-g_i(\boldsymbol{x}_t)}$ from the memory location.

**Cooperation between DP and RPE.** In the 2-nd Attn layer, the extraction of the memory functions is achieved through an interplay between DP and RPE. Specifically, this is done through *a nice interaction between the temporal space (provided by RPE) and the token space (provided by DP)*. Please refer to Appendix C.1 for more details.

**Rethinking DP in Attn.** Our proof highlights that the core mechanism of Attn is to provide a nice interaction between the temporal space and the token space through the cooperation of DP and RPE. This leads us to the following question: *Is DP in Attn necessary and replaceable?* The following two propositions provide some hints.

**Proposition 4.2** (DP vs. DP-free (informal)). *There exists a target $\mathbf{H} \in \mathcal{H}_{(1,1)}^{\mathrm{Adap}}$ (10) such that:*

*(A) For any $\epsilon > 0$, there exists a 1-layer Attn $\mathbf{Attn}^{\mathrm{DP}}$ such that $\left\|\mathbf{H} - \mathbf{Attn}^{\mathrm{DP}}\right\| \leq \epsilon$.*

*(B) For any 1-layer DP-free Attn $\mathbf{Attn}^{\mathrm{DPF}}$, a uniform lower bound holds: $\left\|\mathbf{H} - \mathbf{Attn}^{\mathrm{DPF}}\right\| \geq \frac{2}{3}$.*

Proposition 4.2 reveal a significant distinction in the expressiveness of two network types for modeling adaptive, long, but sparse memories. Specifically, 1-layer Attn with DP can effectively model this task, while 1-layer DP-free Attn provably fails. This finding underscores the essential role of DP in providing the necessary nonlinearity for Attn to model adaptive memories. The formal version of Proposition 4.2 and its proof can be found in Appendix C.2.

**Proposition 4.3** (Substitute for DP (informal)). *There exists a substitute structure for DP, requiring only $\mathcal{O}(D)$ parameters (compared to $\mathcal{O}(D^2)$ in standard DP) that can effectively model $\mathbf{H} \in \mathcal{H}_{(1,M)}^{\mathrm{adap}}$ (10). Specifically, if we substitute DP with this structure, 1-layer Transformer can achieve the same approximation rate as stated in Section 4.1.*

Proposition 4.3 demonstrates the existence of a structurally simpler yet effective alternative to traditional DP for modeling (10). This alternative is proposed based on our insights into the role of Attn in facilitating the interaction between the temporal space and the token space. Specifically, we propose a more direct structure to achieve this interaction. More details are deferred to Appendix C.3.

## 4.3 Theoretical results and insights: The general case

**Theorem 4.4** (Approximation rate, general case). *For any target* $\mathbf{H} \in \mathcal{H}_{(K,M)}^{\mathrm{Adap}}$, *rate* $n \in \mathbb{N}_+$, *and* $H, m \in \mathbb{N}_+$, *there exists an* $L$-*layer* ($L = K + 1 + \mathbb{I}\{M \geq K + 1\}$) *Transformer* $\mathbf{TF} \in \mathcal{TF}_{(L,H,m)}^{\mathtt{type}}$ (12) *and a constant* $C(n)$ *such that: if the width satisfies if the width satisfies* $m \geq \begin{cases} \tilde{\Omega}\big( \max_{i \in [K]} \vee \sum_{i=K+1}^{M} \|g_i\|_{\mathcal{B}}^2 \big) & , \mathtt{type} = \mathrm{lin}, \\ \tilde{\Omega}\big( \max_{i \in [K]} \vee \sum_{i=K+1}^{M} \|\log g_i\|_{\mathcal{B}}^2 T_i^2 \big) & , \mathtt{type} = \log \end{cases}$, *then the following approximation rate holds:*

$$\|\mathbf{H} - \mathbf{TF}\| \leq \mathcal{E}_{\mathrm{FFN}} + \|f\|_{\mathrm{Lip}} \, \mathcal{E}_{\mathrm{Attn}}(\mathtt{type}), \text{ where}$$

$$\mathcal{E}_{\mathrm{FFN}} = \tilde{\mathcal{O}}\left( \frac{\|f\|_{\mathcal{B}}}{\sqrt{m}} \right), \mathcal{E}_{\mathrm{Attn}}(\mathtt{type}) = \begin{cases} \mathcal{O}\left( \frac{C(n)}{H^n} \sqrt{\sum_{l=1}^{K} e^{0.02(n+1)T_l} + \left(\sum_{l=K+1}^{M} e^{0.01T_l}\right)^{2n+2}} \right), \mathtt{type} = \mathrm{lin} \\ \mathcal{O}\left( \frac{C(n)}{H^n} \sqrt{\sum_{l=1}^{K} T_l^{2.02(n+1)} + \left(\sum_{l=K+1}^{M} T_l^{1.01}\right)^{2n+2}} \right), \mathtt{type} = \log \end{cases}.$$

In Theorem 4.4, we establish the *approximation rate* of deep Transformer for modeling $K$-adaptive, long but $M$-sparse memories. Similar to that in Theorem 4.1, the approximation error divides into two distinct terms. A key difference from Theorem 4.1 is the impact of the nested relationships among the memory functions on the required number of layers, Attn heads, and the width of FFN layers. The nested structure within the initial $K$ memories mandates sequential processing in the first $K$ layers one by one. If $M \geq K + 1$, then in the $K + 1$-th layer, the remaining $M - K$ non-nested memory functions $t_{K+1}, \cdots, t_M$ are concurrently processed. The proof of Theorem 4.4 is deferred to Appendix D.1.

**Distinct roles of the number of layers** $L$, **the number of Attn heads** $H$, **and the width of FFN layers** $m$. Theorem 4.4 and its proof highlight the distinct roles of three key hyper-parameters of Transformer: $L$, $H$, and $m$. Deeper Transformer are capable of handling the memories with more intricate nested relationships, requiring a $K + 1$ layer network for a nesting complexity of $K$. In contrast, the number of heads and width needed is dictated by the individual complexity of memory functions themselves ($\|g_i\|_{\mathcal{B}}, \|\log g_i\|_{\mathcal{B}}, T_i$ for memory $g_i$), necessitating that each layer's Attn heads and FFN width are sufficient to capture the memory functions extracted in that layer. This understanding is quite intuitive: If the content of the next token relies on a few previous tokens in an independent way, we can treat each such dependence with a separate attention head. There is no need for many layers.

**Mitigating required head and width with depth.** Recalling Theorem 4.1, the memories lacking nested relationships can be efficiently approximated by 2-layer Transformer with a sufficient number of heads and width. The subsequent proposition further explores how increasing the depth of Transformer can influence its efficiency for modeling memories without nested relationships.

**Proposition 4.5** (Deep network, warm-up case). *For any target* $\mathbf{H} \in \mathcal{H}_{(1,M)}^{\mathrm{Adap}}$ (8), *rate* $n \in \mathbb{N}_+$, *and* $H, m \in \mathbb{N}_+$, *there exists an* $M + 1$-*layer Transformer* $\mathbf{TF} \in \mathcal{TF}_{(M+1,H,m)}^{\mathtt{type}}$ (12) *and a constant* $C(n)$ *such that: if the width satisfies* $m \geq \begin{cases} \tilde{\Omega}\big( \max_{i \in [K]} \|g_i\|_{\mathcal{B}}^2 \big) & , \mathtt{type} = \mathrm{lin}, \\ \tilde{\Omega}\big( \max_{i \in [K]} \|\log g_i\|_{\mathcal{B}}^2 T_i^2 \big) & , \mathtt{type} = \log \end{cases}$, *then the following approximation rate holds:*

$$\|\mathbf{H} - \mathbf{TF}\| \leq \mathcal{E}_{\mathrm{FFN}} + \|f\|_{\mathrm{Lip}} \, \mathcal{E}_{\mathrm{Attn}}(\mathtt{type}),$$

*where* $\mathcal{E}_{\mathrm{FFN}} = \tilde{\mathcal{O}}\left( \frac{\|f\|_{\mathcal{B}}}{\sqrt{m}} \right)$ *and* $\mathcal{E}_{\mathrm{Attn}}(\mathtt{type}) = \begin{cases} \mathcal{O}\left( \frac{C(n)}{H^n} \sqrt{\sum_{l=1}^{K} e^{0.02(n+1)T_l}} \right), \mathtt{type} = \mathrm{lin} \\ \mathcal{O}\left( \frac{C(n)}{H^n} \sqrt{\sum_{l=1}^{K} T_l^{2.02(n+1)}} \right), \mathtt{type} = \log \end{cases}.$

Upon comparing Proposition 4.5 with Theorem 4.1, a notable distinction becomes evident between 2-layer and $M + 1$-layer Transformer in terms of the requirement of the number of Attn heads and the width of FFN layers. Specifically, for 2-layer Transformer, the required width is proportionally linked

to the *sum* of all the memory functions' complexity ($\|g_i\|_{\mathcal{B}}$, $\|\log g_i\|_{\mathcal{B}}$, $T_i$ for memory function $g_i$). In contrast, for $M + 1$-layer Transformer, the required width correlates with the *maximum* complexity of the memory functions, much lower than that for 2-layer Transformer. Similarly, the required number of heads for $M + 1$-layer Transformer is much fewer than that for 2-layer Transformer. Please refer to Appendix D.2 for a detailed comparison. The observation suggests that increased depth can significantly reduce the demands on the number of heads and the width. The underlying reason is that deep networks can distribute the memories across different layers for processing, with each layer focusing on approximating only a single memory function.

## 5   Essentially $M$-sparse memories

### 5.1   Problem formulation

In language tasks, each token possesses clear semantic meaning. As a result, the structure of the memory is sparse in the original space. This aligns well with our modeling assumptions discussed in Section 3 and 4. However, in other machine learning tasks, we may encounter situations where the input tokens lack distinct semantic meaning. This might happen in image processing or classical signal processing. In these situations, the memory structure could potentially be dense in the original space. Nonetheless, the memory structure might exhibit sparsity in some transformed domain. We call such memory structure "essentially sparse". In this section, we study the situation in which the memory structure in long-ranged but essentially sparse. For simplicity, we consider the situation in which the positions of the memory kernels are fixed. The analysis can be easily extended to the situation with an adaptive memory structure.

**Fixed, essentially $M$-sparse memory.** Consider the following situation:

$$y_t = f\left(\left(\boldsymbol{X} * \rho_1\right)(t), \cdots, \left(\boldsymbol{X} * \rho_M\right)(t)\right), \tag{13}$$

where $\rho_1(\cdot), \cdots, \rho_M(\cdot) \in \ell^1(\mathbb{N})$ serve as memory kernels, and $(\boldsymbol{X} * \rho_k)(t) = \sum_{s=0}^{+\infty} \boldsymbol{x}_{t-s} \rho_k(s)$ denotes the convolution of the inputs with kernel $\rho_k$.

**Target class** and **Transformer hypothesis class.** The target class for modeling essentially sparse memories is defined as:

$$\mathcal{H}^{\mathrm{Ess}} := \left\{\mathbf{H} : \ \mathbf{H}_t(\boldsymbol{X}) = (13), \ \text{where } \rho_1, \cdots, \rho_M \in \ell^1(\mathbb{N}), f \in \mathcal{B} \cap \mathcal{L}\right\}. \tag{14}$$

For the hypothesis class, we consider one-layer dot-product-free Transformer with Attn head number $H$ and FFN width $m$, as defined in (7).

**Examples.** Essentially sparse memories are prevalent in real-world scenarios:

(I) *Image Tasks.* In CV, a fundamental objective is identifying and representing meaningful "features", such as ears, nose, etc. These features can often be modeled using convolution kernels, leading to a task in the form $y = f\left(\boldsymbol{X} * \rho_{\mathrm{eye}}, \boldsymbol{X} * \rho_{\mathrm{nose}}, \boldsymbol{X} * \rho_{\mathrm{ear}}\right)$. This is an extension of the task we discussed above, in which the kernel functions $\{\rho_j\}$ are data-dependent ("adaptive" in the terminology used in the previous section).

(II) *Signal processing.* In signal processing, it is commonly the case that the signals are highly sparse under Wavelet or Fourier transforms. For instance, let $\psi(\cdot)$ be a wavelet function and define $\psi_{a,b}(t) := \psi(\frac{t-b}{a})/\sqrt{|a|}$. Then we have $y = f\left(\boldsymbol{X} * \psi_{a_1,b_1}, \cdots, \boldsymbol{X} * \psi_{a_M,b_M}\right)$ where $(a_1, b_1), \cdots, (a_M, b_M)$ might be data-dependent.

(III) *Mathematical calculation.* Consider algebraic operations where memory exhibits sparsity under specific linear transformations. For example, $y_t = 10x_t + x_{t-4}/(\sum_{s=0}^{100} w_s x_{t-10-s}) - \sum_{s=0}^{+\infty} v_s x_{t-100-s}$ can be represented in our framework as $y = f\left(\boldsymbol{X} * \rho_1, \cdots, \boldsymbol{X} * \rho_4\right)$, where each $\rho_i$ represents a specific linear transformation.

### 5.2   Theoretical results and insights

**Theorem 5.1** (Approximation rates)**.**
*(A) Consider $\mathcal{H}^{\mathrm{Ess}}$ (14) with exponentially decayed memory kernels, i.e., there exists $\beta > 0$ such that*

$\rho_1(t), \cdots, \rho_M(t) = \mathcal{O}(e^{-\beta t})$. *Then for any target* $\mathbf{H} \in \mathcal{H}^{\mathrm{Ess}}$, *rate* $n \in [\lfloor 99\beta \rfloor]$, *and* $H, m \in \mathbb{N}_+$, *there exists a 1-layer DP-free Transformer* $\mathbf{TF} \in \mathcal{TF}_{(1,H,m)}^{\mathrm{DPF,lin}}$ (7) *and a constant* $C(n)$ *such that*

$$\|\|\mathbf{H} - \mathbf{TF}\|\| \leq \mathcal{E}_{\mathrm{FFN}} + \|f\|_{\mathrm{Lip}} \cdot \mathcal{E}_{\mathrm{Attn}};$$

*(B) Consider* $\mathcal{H}^{\mathrm{Ess}}$ *(14) with polynomially decayed memory kernels, i.e., there exists* $\beta > 1$ *such that* $\rho_1(t), \cdots, \rho_M(t) = \mathcal{O}(t^{-\beta})$. *Then for any target* $\mathbf{H} \in \mathcal{H}^{\mathrm{Ess}}$, *rate* $n \in [\lfloor 0.99\beta \rfloor - 1]$, *and* $H, m \in \mathbb{N}_+$, *there exists a 1-layer DP-free Transformer* $\mathbf{TF} \in \mathcal{TF}_{(1,H,m)}^{\mathrm{DPF,log}}$ (7) *and a constant* $C(n)$ *such that*

$$\|\|\mathbf{H} - \mathbf{TF}\|\| \leq \mathcal{E}_{\mathrm{FFN}} + \|f\|_{\mathrm{Lip}} \cdot \mathcal{E}_{\mathrm{Attn}};$$

*where* $\mathcal{E}_{\mathrm{FFN}} = \tilde{\mathcal{O}}\left(\frac{\|f\|_{\mathcal{B}}}{\sqrt{m}}\right), \mathcal{E}_{\mathrm{Attn}} = \mathcal{O}\left(\frac{C(n)M^{n+1}}{H^n}\right).$

Theorem 5.1 illustrates that one-layer DP-free Transformer with lin-RPE is effective in modeling essentially sparse memories with exponentially decayed kernels, and one-layer DP-free Transformer with log-RPE can efficiently model the memories with polynomially decayed kernels. A key difference between Theorem 5.1 and Theorem 3.1 lies in the memory kernels they address. In Theorem 5.1, the Attn layer should approximate general memory kernels $\rho_i(\cdot)$, instead of approximating indicator kernels $\mathbb{I}\{\cdot = T_i\}$ in Theorem 3.1. The proof of Theorem 5.1 can be found in Appendix E.

**Overcoming the Curse of Memory (CoM).** For recurrent neural networks (RNN), it was discovered (Li et al., 2021; 2022) that both approximation and optimization become exceedingly difficult when the target has long-term memory. This phenomenon is referred as the "*curse of memory*", or "CoM". It was shown in (Li et al., 2021; 2022) that RNN requires an exponentially large number of neurons to approximate targets with heavy-tailed memory kernels, such as the ones that exhibit polynomial decay. In contrast, Theorem 5.1 reveals that Transformer with log-RPE efficiently handles polynomial decaying memory kernels, requiring only a polynomial number of neurons for effective approximation. This finding theoretically elucidates the superior performance of T5's RPE and KERPLE(log) in length generalization task in practice (Section G.1).

## 6 Experimental Validation

As summarized in Section 1, our theoretical analysis reveals novel insights into the expressive power and mechanisms of Transformer. To validate these insights, we conduct experiments ranging from simple toy models to more complex language model pre-training. Due to space constraints, detailed experimental validation and practical implications of our insights are presented in Appendix H.

## 7 Conclusion and Future Work

In this work, we investigate theoretically the expressive power and the mechanisms of Transformer for modeling long but sparse memories. Our analysis establishes explicit approximation rates and offers much-needed insights into the functionalities of the various components of Transformer. However, we still have a long way to go for a full theoretical understanding of Transformer. For instance, although we have investigated the mechanisms of Transformer in terms of expressive power, the evolution of the mechanisms during the training process remains elusive. Recent studies revealed that Transformer exhibits multi-phase learning dynamics (Boix-Adsera et al., 2023) and undergoes phase transitions (Olsson et al., 2022) during training, akin to the phenomenon of learning with increasing complexity in classical neural networks (Kalimeris et al., 2019; Xu et al., 2019; Rahaman et al., 2019; Abbe et al., 2023a; Wang and Ma, 2023). These and other issues will be studied in future work.

## Acknowledgments

This work is supported in part by the National Key Basic Research Program of China (No. 2015CB856000). We thank Prof. Qianxiao Li, Prof. Lei Wu, Dr. Zhong Li, and Dr. Hongkang Yang for helpful discussions and anonymous reviewers for their valuable suggestions.

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

# Appendix

## A  Detailed Related Works

**Theoretical results of Transformer.** We first review the expressive power results of Transformer. Yun et al. (2019) first proved the universal approximation property (UAP) of Transformer, highlighting the crucial role of PE in breaking permutation invariance. Edelman et al. (2022) demonstrated that Transformer can approximate fixed sparse functions. Dehghani et al. (2019); Pérez et al. (2021); Wei et al. (2022a) explored the Turing-completeness of infinite-precision and finite-precision Transformer. Giannou et al. (2023) showed that looped Transformer can implement practical computer programs.

Jiang and Li (2023) provided explicit approximation rates for Transformer in sequences modeling with inherent graph structures. Liu et al. (2022) found that Transformer can execute finite-state automata. Ma and Ying (2022) asserted the natural suitability of Attn for achieving permutation equivariance. Besides these affirmative results, several studies characterized the expressivity limitation of Transformers, particularly in modeling formal languages or simulating circuits (Hahn, 2020; Weiss et al., 2021; Bhattamishra et al., 2020; Merrill and Sabharwal, 2023b; Merrill et al., 2022). Additionally Feng et al. (2023); Merrill and Sabharwal (2023a) examined the expressivity of Transformer using Chain of Thought prompting (Wei et al., 2022b). Moreover, some studies showed that the in-context learning ability of Transformer is attainable by simulating gradient-based iterations across various layers (Garg et al., 2022; Akyürek et al., 2022; von Oswald et al., 2023; Von Oswald et al., 2023; Mahankali et al., 2023; Bai et al., 2023; Shen et al., 2023). Besides, experimental studies also provide insights into the mechanisms of Transformer through induction head (Elhage et al., 2021; Olsson et al., 2022), information flow (Wang et al., 2023), anchor functions (Zhang et al., 2024), etc.

**Positional encoding.** One core component of Transformer is the PE, which facilitates the representation of input sequence order. Theoretically, Transformer without PE lacks UAP and is restricted to representing permutation-invariant functions. PE was first introduced in Vaswani et al. (2017). It has limitations in encoding unseen positions. To overcome this difficulty, Shaw et al. (2018) introduced RPE. Subsequent studies proposed various different RPE types. Notable examples include T5's RPE (Raffel et al., 2020), Rotary RPE (Su et al., 2024) (utilized in PaLM (Chowdhery et al., 2023) and LlaMA (Touvron et al., 2023)), Alibi RPE (Press et al., 2022) (employed in BLOOM (Workshop et al., 2022)), and KERPLE (Chi et al., 2022). A prevailing belief is that RPEs can outperform APEs in the "length generalization task" (Ontañón et al., 2022; Csordás et al., 2021)– the ability to generalize from smaller training contexts to larger ones, a critical challenge for Large Language Models (Anil et al., 2022; Abbe et al., 2023b). However, Press et al. (2022) revealed that the commonly used Rotary RPE may exhibit suboptimal performance in this task. The recent work (Kazemnejad et al., 2023; Chi et al., 2022) conducted systematic experiments comparing the length generalization capabilities of Transformers with various RPEs and APEs, suggesting that *the RPEs used in T5 and KERPLE(log) demonstrate superior performance over other types*.

**Rethinking dot-product.** Another critical component of Transformer is the DP structure. Due to its quadratic cost as a function of the sequence length, the necessity of DP has always been questioned. Numerous variants of DP have been proposed, demonstrating competitive performance across diverse tasks. Representative examples include Longformer (Beltagy et al., 2020), Big Bird (Zaheer et al., 2020), Reformer (Kitaev et al., 2020), Linformer (Wang et al., 2020), Performer (Choromanski et al., 2020), Synthesizer (Tay et al., 2021), etc. In particular, a recent study (Allen-Zhu and Li, 2023) compared standard and DP-free Transformers in modeling "context-free grammar". Their findings suggested that the presence of DP has a marginal impact on performance. *These evidences motivate us to rethink the necessity of DP in Transformer*.

**Sparsity** (Donoho, 2006; Candès and Wakin, 2008) has gained considerable attention in sequence modeling. In classical signal processing, there is a prevailing notion that valuable signals are extremely sparse. For example, when representing an image, one often finds that only a few wavelet coefficients hold significant values in wavelet space (Meyer, 1992). In NLP, the starting point off the traditional $n$-gram model (Shannon, 1948) is that the next token only relies on a few previous tokens. Such models, however, overlook long-range information, often resulting in suboptimal performance. For NLP tasks such as dependency parsing (Nivre and Scholz, 2004), sentiment analysis (Nasukawa and Yi, 2003), part-of-speech tagging (Francis and Kucera, 1979), and continuation writing (Brown et al., 2020; OpenAI, 2023), it is indeed often the case that only a limited subset of preceding information is crucial for accurate prediction. However, these relevant information can be quite distant. For instance, the resolution of a mystery novel may hinge on elements introduced at the outset. Moreover, for Transformer networks, extensive research into the visualization and interpretability has revealed that (i) the learned activation maps of FFN layers are extremely sparse (Li et al., 2023); (ii) the learned self-attention matrices exhibit notable sparsity, yet it does not closely resemble a diagonal configuration (Elhage et al., 2021). These observations suggest that the prediction of the next token is influenced by a small number of previous tokens which might be far away. Therefore, *being able to represent sparse but long-range dependence is important for sequence modeling*.

# B  Proof of Section 3

## B.1  Proof of Theorem 3.1

In this subsection, we give the detailed proofs of fixed, long but sparse memory:

$$\boldsymbol{y}_t = \boldsymbol{f}(\boldsymbol{x}_t, \boldsymbol{x}_{t-T_1}, \cdots, \boldsymbol{x}_{t-T_M}),$$

where $1 \leq T_1 < \cdots < T_M < +\infty$ signify the fixed positions of the memories.

**Theorem B.1** (Restatement of Theorem 3.1). *For any target $\mathbf{H} \in \mathcal{H}^{\mathrm{Fix}}$ (6), rate $n \in \mathbb{N}_+$, and $H, m \in \mathbb{N}_+$, there exists a 1-layer Transformer $\mathbf{TF} \in \mathcal{TF}^{\mathrm{DPF,type}}_{(1,H,m)}$ (7) and a constant $C(n)$ such that*

$$\|\mathbf{H} - \mathbf{TF}\| \leq \mathcal{E}_{\mathrm{FFN}} + \|f\|_{\mathrm{Lip}} \, \mathcal{E}_{\mathrm{Attn}}(\mathtt{type}),$$

*where $\mathcal{E}_{\mathrm{FFN}} = \tilde{\mathcal{O}}\left(\frac{\|f\|_{\mathcal{B}}}{\sqrt{m}}\right)$ and*

$$\mathcal{E}_{\mathrm{Attn}}(\mathtt{type}) = \begin{cases} \mathcal{O}\left(\frac{C(n)}{H^n}\left(\sum_{i=1}^{M} e^{0.01 T_i}\right)^{n+1}\right), & \mathtt{type} = \mathrm{lin} \\ \mathcal{O}\left(\frac{C(n)}{H^n}\left(\sum_{i=1}^{M} T_i^{1.01}\right)^{n+1}\right), & \mathtt{type} = \mathrm{log} \end{cases}.$$

*Proof of Theorem B.1.*
First, we choose the embedding dimension $D = (M+1)d$, and select the simple embedding $\boldsymbol{W}_E = (\boldsymbol{I}_{d \times d}, \boldsymbol{0})^\top \in \mathbb{R}^{D \times d}, \boldsymbol{b}_E = \boldsymbol{0} \in \mathbb{R}^D$.

Then for any input sequence $\boldsymbol{X} = (\boldsymbol{x}_t)_{t \in \mathbb{Z}}$, the token after embedding satisfies:

$$\boldsymbol{x}_t^E = \boldsymbol{W}_E \boldsymbol{x}_t + \boldsymbol{b}_E = (\boldsymbol{x}_t^\top, \boldsymbol{0}^\top)^\top \in \mathbb{R}^D.$$

For one-layer Dot-product-free Transformer $\mathbf{TF} \in \mathcal{TF}^{\mathrm{DPF,type}}_{(1,H,m)}$ with $\phi_{\mathtt{type}}$, the output token $\mathbf{TF}_t(\boldsymbol{X})$ of $t$-th input token $\boldsymbol{x}_t$ satisfies:

$$\boldsymbol{x}_t^{(1/2)} = \boldsymbol{x}_t^{(0)} + \boldsymbol{W}_O^{(1)} \sum_{h=1}^{H} \mathbf{Attn}_t^{(1,h)}(\boldsymbol{X}^{(0)}),$$

$$\boldsymbol{x}_t^{(1)} = \mathbf{FFN}^{(1)}(\boldsymbol{x}_t^{(1/2)})$$

where

$$\mathbf{Attn}_t^{(1,h)}(\boldsymbol{X}) = \boldsymbol{W}_V^{(1,h)} \sum_{s=0}^{+\infty} \frac{\boldsymbol{x}_{t-s} \exp\left(p^{(1,h)} \phi_{\mathtt{type}}(s)\right)}{\sum_{j=0}^{+\infty} \exp\left(p^{(1,h)} \phi_{\mathtt{type}}(j)\right)}.$$

This proof can be summarized as the following process:

$$\cdots \quad \boldsymbol{x}_t^E \quad \cdots$$
$$\text{Step I. Attn layer } \downarrow$$
$$\cdots \quad \boldsymbol{x}_t^{(1/2)} \approx (\boldsymbol{x}_t^\top, \boldsymbol{x}_{t-T_1}^\top, \cdots, \boldsymbol{x}_{t-T_M}^\top)^\top \quad \cdots$$
$$\text{Step II. FFN layer } \downarrow$$
$$\cdots \quad \boldsymbol{x}_t^{(1)} \approx \boldsymbol{f}(\boldsymbol{x}_t, \boldsymbol{x}_{t-T_1}, \cdots, \boldsymbol{x}_{t-T_M}) \quad \cdots$$

Now we give the formal proof.

**Step I.** Extract the memory locations by (Dot-product-free) Attn layer.

We consider to use $H_k$ attention heads (from $\sum_{i=1}^{k-1} H_i + 1$-th head to $\sum_{i=1}^{k} H_i$-th head) to extract it, and it satisfies to $\sum_{k=1}^{M} H_k = H$.

For simplicity, we denote the following projection matrices:

$$\boldsymbol{P}^{(k)} := (\boldsymbol{0}_{d \times kd} \quad \boldsymbol{I}_{d \times d} \quad \boldsymbol{0}) \in \mathbb{R}^{d \times D}, \quad 1 \le k \le M.$$

$$\boldsymbol{P}_{\perp}^{(k)} := \begin{pmatrix} \boldsymbol{I}_{kd \times kd} & \boldsymbol{0}_{d \times d} & \boldsymbol{0} \\ \boldsymbol{0} & \boldsymbol{0}_{d \times d} & \boldsymbol{I}_{(M-k)d \times (M-k)d} \end{pmatrix} \in \mathbb{R}^{Md \times D}, \quad 1 \le k \le M.$$

Now we consider the extraction of $k$-th memory $\boldsymbol{x}_{t-T_k}$ $(1 \le k \le M)$.

- **Case** $\texttt{type} = \mathrm{lin}$.

  By Lemma F.1, for any rate $n \in \mathbb{N}_+$, there exists an constant $C(n)$ and a function

  $$\phi_k^{\mathrm{exp}}(t) = \sum_{\sum_{i=1}^{k-1} H_i + 1 \le h \le \sum_{i=1}^{k} H_i} \alpha_h e^{-\beta_h t}$$

  such that $\beta_h > 0$ and

  $$\|\mathbb{I}\{\cdot = T_k\} - \phi_k^{\mathrm{exp}}(\cdot)\|_{\ell_1(\mathbb{N})} = \sum_{s=0}^{+\infty} |\mathbb{I}\{s = T_k\} - \phi_k^{\mathrm{exp}}(s)| \le C(n) \frac{e^{0.01(n+1)T_k}}{H_k^n}.$$

  Therefore, for these attention heads ($\sum_{i=1}^{k-1} H_i + 1 \le h \le \sum_{i=1}^{k} H_i$), we can choose

  $$p^{(1,h)} = \beta_h, \quad \boldsymbol{W}_V^{(1,h)} = \alpha_h \left( \sum_{j=0}^{+\infty} \exp(-\beta_h j) \right) \boldsymbol{\delta}_{(k+1,1)}^{d \times d},$$

  where $\boldsymbol{\delta}^{(k+1,1)} \in \mathbb{R}^{D \times D}$ means that: it equals to $\boldsymbol{I}_{d \times d}$ for the $(k+1,1)$-th $d \times d$ blocks, and $\boldsymbol{0}_{d \times d}$ for the other $d \times d$ blocks.

  Then it holds that:

  $$\sum_{h=\sum_{i=1}^{k-1} H_i + 1}^{\sum_{i=1}^{k} H_i} \mathbf{Attn}_t^{(1,h)}(\boldsymbol{X}^{(0)}) = \sum_{h=\sum_{i=1}^{k-1} H_i + 1}^{\sum_{i=1}^{k} H_i} \alpha_h \sum_{s=0}^{+\infty} e^{-\beta_h s} \begin{pmatrix} \boldsymbol{0}_{kd} \\ \boldsymbol{x}_{t-s} \\ \boldsymbol{0} \end{pmatrix} \in \mathbb{R}^{D}.$$

  This implies:

  $$\boldsymbol{P}^{(k)} \sum_{h=\sum_{i=1}^{k-1} H_i + 1}^{\sum_{i=1}^{k} H_i} \mathbf{Attn}_t^{(1,h)}(\boldsymbol{X}^{(0)}) = \sum_{h=\sum_{i=1}^{k-1} H_i + 1}^{\sum_{i=1}^{k} H_i} \alpha_h \sum_{s=0}^{+\infty} e^{-\beta_h s} \boldsymbol{x}_{t-s},$$

  $$\boldsymbol{P}_{\perp}^{(k)} \sum_{h=\sum_{i=1}^{k-1} H_i + 1}^{\sum_{i=1}^{k} H_i} \mathbf{Attn}_t^{(1,h)}(\boldsymbol{X}^{(0)}) = \boldsymbol{0},$$

  moreover, the following estimate holds:

  $$\left\| \boldsymbol{P}^{(k)} \sum_{h=\sum_{i=1}^{k-1} H_i + 1}^{\sum_{i=1}^{k} H_i} \mathbf{Attn}_t^{(1,h)}(\boldsymbol{X}^{(0)}) - \boldsymbol{x}_{t-T_k} \right\|_2$$

  $$= \left\| \sum_{h=\sum_{i=1}^{k-1} H_i + 1}^{\sum_{i=1}^{k} H_i} \alpha_h \sum_{s=0}^{+\infty} e^{-\beta_h s} \boldsymbol{x}_{t-s} - \boldsymbol{x}_{t-T_k} \right\|_2$$

$$= \left\| \sum_{s=0}^{+\infty} \left( \sum_{h=\sum_{i=1}^{k-1} H_i+1}^{\sum_{i=1}^{k} H_i} \alpha_h e^{-\beta_h s} - \mathbb{I}\{s = T_k\} \right) \boldsymbol{x}_{t-s} \right\|_2$$

$$\leq \sum_{s=0}^{+\infty} \left| \sum_{h=\sum_{i=1}^{k-1} H_i+1}^{\sum_{i=1}^{k} H_i} \alpha_h e^{-\beta_h s} - \mathbb{I}\{s = T_k\} \right|$$

$$= \left\| \phi_k^{\exp}(\cdot) - \mathbb{I}\{\cdot = T_k\} \right\|_{\ell_1(\mathbb{N})} \leq C(n) \frac{e^{0.01(n+1)T_k}}{H_k^n}.$$

- **Case** `type` $= \log$.

  By Lemma F.4, for any rate $n \in \mathbb{N}_+$, there exists an constant $C(n)$ and a function

  $$\phi_k^{\mathrm{poly}}(t) = \sum_{\sum_{i=1}^{k-1} H_i+1 \leq h \leq \sum_{i=1}^{k} H_i} \alpha_h t^{-\beta_h},$$

  such that $\beta_h > 1$ and

  $$\left\| \mathbb{I}\{\cdot = T_k\} - \phi_k^{\mathrm{poly}}(\cdot) \right\|_{\ell_1(\mathbb{N}_+)} = \sum_{s=1}^{+\infty} \left| \mathbb{I}\{s = T_k\} - \phi_k^{\mathrm{poly}}(s) \right| \leq C(n) \frac{T_k^{1.01(n+1)} H_k^n}{.}$$

  Therefore, for these attention heads ($\sum_{i=1}^{k-1} H_i + 1 \leq h \leq \sum_{i=1}^{k} H_i$), we can choose

  $$p^{(1,h)} = \beta_h, \quad \boldsymbol{W}_V^{(1,h)} = \alpha_h \left( \sum_{j=1}^{+\infty} j^{-\beta_h} \right) \boldsymbol{\delta}^{(k+1,1)},$$

  where $\boldsymbol{\delta}^{(k+1,1)} \in \mathbb{R}^{D \times D}$ means that: it equals to $\boldsymbol{I}_{d \times d}$ for the $(k+1, 1)$-th $d \times d$ blocks, and $\mathbf{0}_{d \times d}$ for the other $d \times d$ blocks.

  Then it holds that:

  $$\boldsymbol{P}^{(k)} \sum_{h=\sum_{i=1}^{k-1} H_i+1}^{\sum_{i=1}^{k} H_i} \mathbf{Attn}_t^{(1,h)}(\boldsymbol{X}^{(0)}) = \sum_{h=\sum_{i=1}^{k-1} H_i+1}^{\sum_{i=1}^{k} H_i} \alpha_h \sum_{s=1}^{+\infty} s^{-\beta_h} \boldsymbol{x}_{t-s},$$

  $$\boldsymbol{P}_{\perp}^{(k)} \sum_{h=\sum_{i=1}^{k-1} H_i+1}^{\sum_{i=1}^{k} H_i} \mathbf{Attn}_t^{(1,h)}(\boldsymbol{X}^{(0)}) = \mathbf{0},$$

  moreover, the following estimate holds:

  $$\left\| \sum_{h=\sum_{i=1}^{k-1} H_i+1}^{\sum_{i=1}^{k} H_i} \boldsymbol{P}^{(k)} \mathbf{Attn}_t^{(1,h)}(\boldsymbol{X}^{(0)}) - \boldsymbol{x}_{t-T_k} \right\|_2$$

  $$= \left\| \sum_{h=\sum_{i=1}^{k-1} H_i+1}^{\sum_{i=1}^{k} H_i} \alpha_h \sum_{s=1}^{+\infty} s^{-\beta_h} \boldsymbol{x}_{t-s} - \boldsymbol{x}_{t-T_k} \right\|_2$$

  $$= \left\| \sum_{s=1}^{+\infty} \left( \sum_{h=\sum_{i=1}^{k-1} H_i+1}^{\sum_{i=1}^{k} H_i} \alpha_h s^{-\beta_h} - \mathbb{I}\{s = T_k\} \right) \boldsymbol{x}_{t-s} \right\|_2$$

$$\leq \sum_{s=1}^{+\infty} \left| \sum_{h=\sum_{i=1}^{k-1} H_i+1}^{\sum_{i=1}^{k} H_i} \alpha_h s^{-\beta_h} - \mathbb{I}\{s = T_k\} \right|$$

$$= \left\| \phi_k^{\mathrm{poly}}(\cdot) - \mathbb{I}\{\cdot = T_k\} \right\|_{\ell_1(\mathbb{N}_+)} \leq \mathcal{O}\left( C(n) \frac{T_k^{1.01(n+1)}}{H_k^n} \right).$$

Then we combine the results for all $k \in [M]$ for these two cases. By choose $\boldsymbol{W}_O = \boldsymbol{I}_D$, we have:

$$\left\| \boldsymbol{x}_t^{(1/2)} - \begin{pmatrix} \boldsymbol{x}_t \\ \boldsymbol{x}_{t-t_1} \\ \vdots \\ \boldsymbol{x}_{t-t_M} \end{pmatrix} \right\|_2$$

$$= \left\| \begin{pmatrix} \boldsymbol{x}_t \\ \boldsymbol{0}_d \\ \vdots \\ \boldsymbol{0}_d \end{pmatrix} + \sum_{h=1}^{M} \mathbf{Attn}_t^{(1,h)}(\boldsymbol{X}) - \begin{pmatrix} \boldsymbol{x}_t \\ \boldsymbol{x}_{t-t_1} \\ \vdots \\ \boldsymbol{x}_{t-t_M} \end{pmatrix} \right\|_2 = \left\| \sum_{h=1}^{M} \mathbf{Attn}_t^{(1,h)}(\boldsymbol{X}) - \begin{pmatrix} \boldsymbol{0}_d \\ \boldsymbol{x}_{t-t_1} \\ \vdots \\ \boldsymbol{x}_{t-t_M} \end{pmatrix} \right\|_2$$

$$= \left\| \sum_{k=1}^{M} \left( \sum_{h=\sum_{i=1}^{k-1} H_i+1}^{\sum_{i=1}^{k} H_i} \mathbf{Attn}_t^{(1,h)}(\boldsymbol{X}) - \begin{pmatrix} \boldsymbol{0}_{kd} \\ \boldsymbol{x}_{t-T_k} \\ \boldsymbol{0}_d \end{pmatrix} \right) \right\|_2$$

$$\leq \sum_{k=1}^{M} \left\| \sum_{h=\sum_{i=1}^{k-1} H_i+1}^{\sum_{i=1}^{k} H_i} \mathbf{Attn}_t^{(1,h)}(\boldsymbol{X}) - \begin{pmatrix} \boldsymbol{0}_{kd} \\ \boldsymbol{x}_{t-T_k} \\ \boldsymbol{0}_d \end{pmatrix} \right\|_2$$

$$= \sum_{k=1}^{M} \left\| \sum_{h=\sum_{i=1}^{k-1} H_i+1}^{\sum_{i=1}^{k} H_i} \boldsymbol{P}^{(k)} \mathbf{Attn}_t^{(1,h)}(\boldsymbol{X}) - \boldsymbol{x}_{t-T_k} \right\|_2$$

$$\leq \mathcal{E}_{\mathrm{Attn}}(\texttt{type}) := \begin{cases} C(n) \sum_{k=1}^{M} \frac{e^{0.01(n+1)T_k}}{H_k^n}, & \texttt{type} = \lin \\ C(n) \sum_{k=1}^{M} \frac{T_k^{1.01(n+1)}}{H_k^n}, & \texttt{type} = \log \end{cases}.$$

Consequently, one detail is to assign the head number $\{H_k\}_{k=1}^{M}$ such that the error's sum $\mathcal{E}_{\mathrm{Attn}}(\texttt{type})$ is as small as possible. Our way is solving the minimization problem:

$$\min_{H_1,\cdots,H_M} : \mathcal{E}_{\mathrm{Attn}}(\texttt{type})$$

$$\text{s.t.} \sum_{k=1}^{M} H_k = H,$$

which suggests that we should choose the head number:

$$H_k = \frac{e^{0.01 T_k}}{\sum_{j=1}^{M} e^{0.01 T_j}} H, \quad k \in [M], \quad \texttt{type} = \lin;$$

$$H_k = \frac{T_k^{1.01}}{\sum_{j=1}^{M} T_j^{1.01}} H, \quad k \in [M], \quad \texttt{type} = \log.$$

Thus, we obtain the bound in Step I:

$$\mathcal{E}_{\mathrm{Attn}}(\texttt{type}) \leq \begin{cases} \frac{C(n)}{H^n} \left( \sum_{k=1}^{M} e^{0.01 T_k} \right)^{n+1}, & \texttt{type} = \lin \\ \frac{C(n)}{H^n} \left( \sum_{k=1}^{M} T_k^{1.01} \right)^{n+1}, & \texttt{type} = \log \end{cases}.$$

Furthermore, by choosing $\mathcal{E}_{\text{Attn}}(\texttt{type}) \leq 1$, it holds that

$$\left\| \boldsymbol{x}_t^{(1/2)} \right\|_\infty \leq \left\| \boldsymbol{x}_t^{(1/2)} - \begin{pmatrix} \boldsymbol{x}_t \\ \boldsymbol{x}_{t-t_1} \\ \vdots \\ \boldsymbol{x}_{t-t_M} \end{pmatrix} \right\|_\infty + \left\| \begin{pmatrix} \boldsymbol{x}_t \\ \boldsymbol{x}_{t-t_1} \\ \vdots \\ \boldsymbol{x}_{t-t_M} \end{pmatrix} \right\|_\infty \leq \mathcal{E}_{\text{Attn}}(\texttt{type}) + 1 \leq 2.$$

**Step II.** Approximate the readout function by FFN layer.

In this step, we aim to approximate the function $f$ using two-layer network. By Lemma G.6, there exists a two layer neural network with $m$ neurons defined on $\mathbb{R}^D$

$$\text{FFN}^{(1)}(\boldsymbol{y}) = \sum_{k=1}^{m} a_k \sigma(\boldsymbol{b}_k^\top \boldsymbol{y} + c_k)$$

such that

$$\mathcal{E}_{\text{FFN}} := \left\| \text{FFN}^{(1)} - f \right\|_{L^\infty([-2,2]^D)} \leq \tilde{\mathcal{O}}\left( \frac{\|f\|_{\mathcal{B}}}{\sqrt{m}} \right).$$

**The final bound.**

For any $t$ and $\boldsymbol{X} \in \mathcal{X}$, it holds that

$$\left\| \mathbf{H}_t(\boldsymbol{X}) - \boldsymbol{x}_t^{(1)} \right\| = \left| f\left( \boldsymbol{x}_t, \boldsymbol{x}_{t-t_1}, \cdots \boldsymbol{x}_{t-t_M} \right) - \text{FFN}^{(1)}\left( \boldsymbol{x}_t^{(1/2)} \right) \right|$$

$$= \left| f\left( \boldsymbol{x}_t, \boldsymbol{x}_{t-t_1}, \cdots \boldsymbol{x}_{t-t_M} \right) - f\left( \boldsymbol{x}_t^{(1/2)} \right) + f\left( \boldsymbol{x}_t^{(1/2)} \right) - \text{FFN}^{(1)}\left( \boldsymbol{x}_t^{(1/2)} \right) \right|$$

$$\leq \left| f\left( \boldsymbol{x}_t, \boldsymbol{x}_{t-t_1}, \cdots \boldsymbol{x}_{t-t_M} \right) - f\left( \boldsymbol{x}_t^{(1/2)} \right) \right| + \left| f\left( \boldsymbol{x}_t^{(1/2)} \right) - \text{FFN}^{(1)}\left( \boldsymbol{x}_t^{(1/2)} \right) \right|$$

$$\leq \|f\|_{\text{Lip}} \left\| \left( \boldsymbol{x}_t^\top, \boldsymbol{x}_{t-t_1}^\top, \cdots \boldsymbol{x}_{t-t_M}^\top \right) - \boldsymbol{x}_t^{(1/2)} \right\|_2 + \left\| f - \text{FFN}^{(1)} \right\|_{L^\infty([-2,2]^D)}$$

$$\leq \|f\|_{\text{Lip}} \cdot \mathcal{E}_{\text{Attn}}(\texttt{type}) + \mathcal{E}_{\text{FFN}},$$

where

$$\mathcal{E}_{\text{FFN}} = \tilde{\mathcal{O}}\left( \frac{\|f\|_{\mathcal{B}}}{\sqrt{m}} \right);$$

$$\mathcal{E}_{\text{Attn}}(\texttt{type}) = \begin{cases} \mathcal{O}\left( \frac{C(n)}{H^n} \left( \sum_{k=1}^{M} e^{0.01 T_k} \right)^{n+1} \right), & \texttt{type} = \lin \\ \mathcal{O}\left( \frac{C(n)}{H^n} \left( \sum_{k=1}^{M} T_k^{1.01} \right)^{n+1} \right), & \texttt{type} = \log \end{cases}.$$

Due to the arbitrariness of $t$ and $\boldsymbol{X}$, the proof is completed.

$\square$

# C Proof of Section 4.2

In this section, we give the detailed proofs of the approximation theory of Transformer for modeling the warm-up case of adaptive, long but sparse memory:

$$\boldsymbol{y}_t = \boldsymbol{f}(\boldsymbol{x}_t, \boldsymbol{x}_{t-t_1}, \cdots, \boldsymbol{x}_{t-t_M}),$$

where the adaptive memory satisfies to:

$$t_k = g_k(\boldsymbol{x}_t), \quad k \in [M].$$

Moreover, $g_k(\cdot)$ generate positive integers for the input tokens, and there exist maximum values $T_k$ such that $1 \le g_k(\boldsymbol{x}_t) \le T_k$ holds for any $\boldsymbol{x}_t$ and $k \in [M]$.

To tackle the discrete values of the time and the memory values $g_k(\boldsymbol{x}_t)$, a modified version of standard FFN, termed "FFN with precision", us cibsudered. This approach ensures that the output of FFN undergoes a simple rounding operation. Notably, the precision technique is widely used in LLM training (Kalamkar et al., 2019), such as `BFloat16`. Specifically, for Transformer using RPE with `type`, we use the following FFN with precision:

$$
\begin{aligned}
\widetilde{\mathrm{FFN}}(\boldsymbol{x}) &:= [\mathrm{FFN}(\boldsymbol{x})], & \texttt{type} = \mathrm{lin}; \\
\widetilde{\mathrm{FFN}}(\boldsymbol{x}) &:= \log\left[\exp\left(\mathrm{FFN}(\boldsymbol{x})\right)\right], & \texttt{type} = \log,
\end{aligned}
\tag{15}
$$

where $[\cdot]$ signifies rounding to the nearest integer, i.e., $[x] = \arg\min_{n \in \mathbb{Z}} |n - x| \ (x \in \mathbb{R})$.

It is important to note that the rounding obtained by using the operator $\log[\exp(z)]$, used in (15), is *quite fine*, which is much finer than the vanilla rounding obtained by $[z]$. To elaborate, the following proposition is presented:

**Proposition C.1.** *For any $z \ge 1$, the following holds:*

$$(i) \ |\log[\exp(z)] - z| \le \frac{1}{2\min\{e^z, [e^z]\}}; \quad (ii) \ |[z] - z| \le \frac{1}{2}.$$

## C.1 Proof of Theorem 4.1

**Theorem C.2** (Restatement of Theorem 4.1). *For any target $\mathbf{H} \in \mathcal{H}^{\mathrm{Adap}}_{(1,M)}$ (8), rate $n \in \mathbb{N}_+$, and $H, m \in \mathbb{N}_+$, there exists a two-layer Transformer $\mathbf{TF} \in \mathcal{TF}^{\mathrm{NF,type}}_{(2,H,m)}$ (12) and a constant $C(n)$ such that: if the width satisfies*

$$
m \ge \begin{cases} \tilde{\Omega}\left(\sum_{i=1}^M \|g_i\|^2_{\mathcal{B}}\right), & \texttt{type} = \mathrm{lin} \\ \tilde{\Omega}\left(\sum_{i=1}^M \|\log g_i\|^2_{\mathcal{B}} T_i^2\right), & \texttt{type} = \log \end{cases},
$$

*then the following approximation rate holds:*

$$\|\mathbf{H} - \mathbf{TF}\| \le \mathcal{E}_{\mathrm{FFN}} + \|f\|_{\mathrm{Lip}} \mathcal{E}_{\mathrm{Attn}}(\texttt{type}),$$

*where $\mathcal{E}_{\mathrm{FFN}} = \tilde{\mathcal{O}}\left(\frac{\|f\|_{\mathcal{B}}}{\sqrt{m}}\right)$ and*

$$
\mathcal{E}_{\mathrm{Attn}}(\texttt{type}) = \begin{cases} \mathcal{O}\left(\frac{C(n)}{H^n}\left(\sum_{i=1}^M e^{0.01 T_i}\right)^{n+1}\right), & \texttt{type} = \mathrm{lin} \\ \mathcal{O}\left(\frac{C(n)}{H^n}\left(\sum_{i=1}^M T_i^{1.01}\right)^{n+1}\right), & \texttt{type} = \log \end{cases}.
$$

*Proof of Theorem C.2.*
First, we choose the embedding dimension $D = (M+1)(d+1)$, and select a simple embedding $\boldsymbol{W}_E = (\boldsymbol{I}_{d\times d}, \boldsymbol{0})^\top \in \mathbb{R}^{D\times d}, \boldsymbol{b}_E = \boldsymbol{0} \in \mathbb{R}^D$.

Then for any input sequence $\boldsymbol{X} = (\boldsymbol{x}_t)_{t\in\mathbb{Z}}$, the token after embedding satisfies:

$$\boldsymbol{x}_t^{(0)} = \boldsymbol{W}_E \boldsymbol{x}_t + \boldsymbol{b}_E = (\boldsymbol{x}_t^\top, \boldsymbol{0}^\top)^\top \in \mathbb{R}^D.$$

To tackle the discrete values of $g_m(\boldsymbol{x}_t)$, we utilize $\widetilde{\textbf{FFN}}$, FFN with precision (15). It ensures that the output of FFN undergoes a simple rounding operation.

Thus, for two-layer normalization-free Transformer $\textbf{TF} \in \mathcal{TF}^{\mathrm{NF},\texttt{type}}_{(2,H,m)}$ with $\phi_{\texttt{type}}$, the output token $\boldsymbol{x}_t^{(2)}$ of $t$-th input token satisfies:

$$\boldsymbol{x}_t^{(1/2)} = \boldsymbol{x}_t^{(0)} + \boldsymbol{W}_O^{(1)} \sum_{h=1}^{H} \textbf{Attn}_t^{(1,h)}(\boldsymbol{X}^{(0)}),$$

$$\boldsymbol{x}_t^{(1)} = \boldsymbol{x}_t^{(1/2)} + \widetilde{\textbf{FFN}}^{(1)}(\boldsymbol{x}_t^{(1/2)}),$$

$$\boldsymbol{x}_t^{(3/2)} = \boldsymbol{x}_t^{(1)} + \boldsymbol{W}_O^{(2)} \sum_{h=1}^{H} \textbf{Attn}_t^{(2,h)}(\boldsymbol{X}^{(1)}),$$

$$\boldsymbol{x}_t^{(2)} = \textbf{FFN}^{(2)}(\boldsymbol{x}_t^{(3/2)}),$$

where

$$\textbf{Attn}_t^{(l,h)}(\boldsymbol{X}) = \boldsymbol{W}_V^{(l,h)} \sum_{s=0}^{+\infty} \boldsymbol{x}_{t-s} \exp\left( \left\langle \boldsymbol{W}_Q^{(l,h)} \boldsymbol{x}_t, \boldsymbol{W}_K^{(l,h)} \boldsymbol{x}_{t-s} \right\rangle + p^{(l,h)} \phi_{\texttt{type}}(s) \right).$$

This proof can be summarized as the following process:

- **Case** $\texttt{type} = \mathrm{lin}$.

$$\boldsymbol{x}_t^{(0)}$$
Step I. 1-st Attn $\downarrow$
$$\boldsymbol{x}_t^{(1/2)} = \boldsymbol{x}_t^{(0)}$$
Step II. 1-st FFN $\downarrow$
$$\boldsymbol{x}_t^{(1)} = (\boldsymbol{x}_t^\top, \boldsymbol{0}^\top, g_1(\boldsymbol{x}_t), \cdots, g_M(\boldsymbol{x}_t), 1)^\top$$
Step III. 2-st Attn $\downarrow$
$$\boldsymbol{x}_t^{(3/2)} \approx (\boldsymbol{x}_t^\top, \boldsymbol{x}_{t-g_1(\boldsymbol{x}_t)}^\top, \cdots, \boldsymbol{x}_{t-g_M(\boldsymbol{x}_t)}^\top, g_1(\boldsymbol{x}_t), \cdots, g_M(\boldsymbol{x}_t), 1)^\top$$
Step IV. 2-st FFN $\downarrow$
$$\boldsymbol{x}_t^{(2)} \approx \boldsymbol{f}(\boldsymbol{x}_t, \boldsymbol{x}_{t-g_1(\boldsymbol{x}_t)}, \cdots, \boldsymbol{x}_{t-g_M(\boldsymbol{x}_t)})$$

- **Case** $\texttt{type} = \log$.

$$\boldsymbol{x}_t^{(0)}$$
Step I. 1-st Attn $\downarrow$
$$\boldsymbol{x}_t^{(1/2)} = \boldsymbol{x}_t^{(0)}$$
Step II. 1-st FFN $\downarrow$
$$\boldsymbol{x}_t^{(1)} = (\boldsymbol{x}_t^\top, \boldsymbol{0}^\top, \log g_1(\boldsymbol{x}_t), \cdots, \log g_M(\boldsymbol{x}_t), \log 2)^\top$$
Step III. 2-st Attn $\downarrow$
$$\boldsymbol{x}_t^{(3/2)} \approx (\boldsymbol{x}_t^\top, \boldsymbol{x}_{t-g_1(\boldsymbol{x}_t)}^\top, \cdots, \boldsymbol{x}_{t-g_M(\boldsymbol{x}_t)}^\top, \log g_1(\boldsymbol{x}_t), \cdots, \log g_M(\boldsymbol{x}_t), \log 2)^\top$$
Step IV. 2-st FFN $\downarrow$
$$\boldsymbol{x}_t^{(2)} \approx \boldsymbol{f}(\boldsymbol{x}_t, \boldsymbol{x}_{t-g_1(\boldsymbol{x}_t)}, \cdots, \boldsymbol{x}_{t-g_M(\boldsymbol{x}_t)})$$

Now we give the formal proof.

**Step I.** Identity map.

For the first Attn layer, we only need to do the identity map by taking $\boldsymbol{W}_0^{(1)} = \boldsymbol{0}$. Then $\boldsymbol{x}_t^{(1/2)} = \boldsymbol{x}_t^{(0)}$.

**Step II.** Approximate the adaptive memory function by the first FFN layer.

- **Case** `type` = lin. Our main idea is that using the first FFN layer to express $(\boldsymbol{x}_t^\top, \boldsymbol{0}^\top, g_1(\boldsymbol{x}_t), \cdots, g_M(\boldsymbol{x}_t), 1)^\top$ exactly.

First, we consider to approximate the $r$-th memory function $g_r(\boldsymbol{x})$ by standard FFN.

For any $r \in [M]$, by Lemma G.6, there exists a two-layer neural network with $m_r$ neurons

$$f_{(1,r)}^{\text{2NN}}(\boldsymbol{x}) = \sum_{k=1}^{m_r} a_k^{(1,r)} \sigma\left(\boldsymbol{b}_k^{(1,r)\,\top} \boldsymbol{x} + c_k^{(1,r)}\right)$$

defined on $\mathbb{R}^d$ such that

$$\left\| g_r - f_{(1,r)}^{\text{2NN}} \right\|_{L^\infty([-1,1]^D)} \leq \tilde{\mathcal{O}}\left( \frac{\|g_r\|_{\mathcal{B}}}{\sqrt{m_r}} \right).$$

Therefore, if we choose

$$\tilde{\mathcal{O}}\left( \frac{\|g_r\|_{\mathcal{B}}}{\sqrt{m_r}} \right) < \frac{1}{2},$$

the following holds:

$$\left| g_r(\boldsymbol{x}_t) - f_{(1,r)}^{\text{2NN}}(\boldsymbol{x}_t) \right| \leq \left\| g_r - f_{(1,r)}^{\text{2NN}} \right\|_{L^\infty([-1,1]^d)} < \frac{1}{2},$$

Noticing $g_r(\boldsymbol{x}_t) \in \mathbb{N}_+$, we have $\left[ f_{(1,r)}^{\text{2NN}}(\boldsymbol{x}_t) \right] = g_r(\boldsymbol{x}_t)$, which implies:

$$\widetilde{f_{(1,r)}^{\text{2NN}}}(\boldsymbol{x}_t) = \left[ f_{(1,r)}^{\text{2NN}}(\boldsymbol{x}_t) \right] = g_r(\boldsymbol{x}_t).$$

Consequently, in order to construct the form $(\boldsymbol{0}^\top, g_1(\boldsymbol{x}_t), \cdots, g_M(\boldsymbol{x}_t), 1)^\top \in \mathbb{R}^D$, we need to arrange the parameters $a_k^{(1,r)}$, $\boldsymbol{b}_k^{(1,r)}$, and $c_k^{(1,r)}$ ($k \in [m_r], r \in [M]$) appropriately.

Denote $\bar{\boldsymbol{b}}_k^{(1,r)} = (\boldsymbol{b}_k^{(1,r)\,\top}, \boldsymbol{0}^\top)^\top \in \mathbb{R}^D$ for $k \in [m_r], r \in [M]$. Consider the following two-layer neural network with $1 + \sum_{r=1}^M m_r$ neurons defined on $\mathbb{R}^D$:

$$\mathbf{FFN}^{(1)}(\boldsymbol{x}) = \sum_{r=1}^M \sum_{1+\sum_{j=0}^{r-1} m_j \leq k \leq \sum_{j=0}^r m_j} \boldsymbol{e}_{D-M+r-1} a_k^{(1,r)} \sigma\left( \bar{\boldsymbol{b}}_k^{(1,r)\,\top} \boldsymbol{x} + c_k^{(1,r)} \right)$$
$$+ \boldsymbol{e}_D \cdot 1 \cdot \sigma(0 + 1).$$

It is easy to verify that for any $\boldsymbol{x}_t^{(1/2)}$, it holds that

$$\mathbf{FFN}^{(1)}(\boldsymbol{x}_t^{(1/2)}) = \mathbf{FFN}^{(1)}(\boldsymbol{x}_t^{(0)})$$

$$= \sum_{r=1}^M \sum_{1+\sum_{j=0}^{r-1} m_j \leq k \leq \sum_{j=0}^r m_j} \boldsymbol{e}_{D-M+r-1} a_k^{(1,r)} \sigma\left( \bar{\boldsymbol{b}}_k^{(1,r)\,\top} \boldsymbol{x}_t^{(0)} + c_k^{(1,r)} \right) + \boldsymbol{e}_D \cdot 1 \cdot \sigma(0 + 1)$$

$$= \sum_{r=1}^M \sum_{1+\sum_{j=0}^{r-1} m_j \leq k \leq \sum_{j=0}^r m_j} \boldsymbol{e}_{D-M+r-1} a_k^{(1,r)} \sigma\left( \boldsymbol{b}_k^{(1,r)\,\top} \boldsymbol{x}_t + c_k^{(1,r)} \right) + \boldsymbol{e}_D \cdot 1 \cdot \sigma(0 + 1)$$

$$= \sum_{r=1}^M \boldsymbol{e}_{D-M+r-1} f_r^{\text{2NN}}(\boldsymbol{x}_t) + \boldsymbol{e}_D$$

$$= (\boldsymbol{0}_d^\top, f_{(1,1)}^{\text{2NN}}(\boldsymbol{x}_t), \cdots, f_{(1,M)}^{\text{2NN}}(\boldsymbol{x}_t), 1)^\top \in \mathbb{R}^D.$$

Moreover, it satisfies that

$$\widetilde{\mathbf{FFN}}^{(1)}(\boldsymbol{x}_t^{(1/2)}) = \left[ \mathbf{FFN}^{(1)}(\boldsymbol{x}_t^{(1/2)}) \right]$$

$$= (\mathbf{0}_d^\top, \left[f_{(1,1)}^{\mathrm{2NN}}(\boldsymbol{x}_t)\right], \cdots, \left[f_{(1,M)}^{\mathrm{2NN}}(\boldsymbol{x}_t)\right], 1)^\top$$

$$= (\mathbf{0}_d^\top, g_1(\boldsymbol{x}_t), \cdots, g_M(\boldsymbol{x}_t), 1)^\top \in \mathbb{R}^D.$$

Thus, we have achieved our goal in this step:

$$\boldsymbol{x}_t^{(1)} = \boldsymbol{x}_t^{(1/2)} + \widetilde{\mathbf{FFN}}^{(1)}(\boldsymbol{x}_t^{(1/2)}) = (\boldsymbol{x}_t^\top, \mathbf{0}^\top, g_1(\boldsymbol{x}_t), \cdots, g_M(\boldsymbol{x}_t), 1)^\top.$$

- **Case** `type` $=$ `log`.  Our main idea is that using the first FFN layer to express $(\boldsymbol{x}_t^\top, \mathbf{0}^\top, \log g_1(\boldsymbol{x}_t), \cdots, \log g_M(\boldsymbol{x}_t), \log 2)^\top$ exactly.

First, we consider to approximate the $r$-th memory function $\log g_r(\boldsymbol{x})$ by standard FFN.

For any $r \in [M]$, by Lemma G.6, there exists a two-layer neural network with $m_r$ neurons

$$f_{(1,r)}^{\mathrm{2NN}}(\boldsymbol{x}) = \sum_{k=1}^{m_r} a_k^{(1,r)} \sigma\left(\boldsymbol{b}_k^{(1,r)\ \top} \boldsymbol{x} + c_k^{(1,r)}\right)$$

defined on $\mathbb{R}^d$ such that

$$\left\|\log g_r - f_{(1,r)}^{\mathrm{2NN}}\right\|_{L^\infty([-1,1]^D)} \le \tilde{\mathcal{O}}\left(\frac{\|\log g_r\|_{\mathcal{B}}}{\sqrt{m_r}}\right).$$

Therefore, if we choose

$$\tilde{\mathcal{O}}\left(\frac{\|\log g_r\|_{\mathcal{B}}}{\sqrt{m_r}}\right) < \frac{1}{4T_r},$$

the following holds:

$$\left|\log g_r(\boldsymbol{x}_t) - f_{(1,r)}^{\mathrm{2NN}}(\boldsymbol{x}_t)\right| \le \left\|g_r - f_{(1,r)}^{\mathrm{2NN}}\right\|_{L^\infty([-1,1]^d)} < \frac{1}{4T_r},$$

which ensures

$$\left|\exp\left(f_{(1,r)}^{\mathrm{2NN}}(\boldsymbol{x}_t)\right) - g_r(\boldsymbol{x}_t)\right| = \left|\exp\left(f_{(1,r)}^{\mathrm{2NN}}(\boldsymbol{x}_t)\right) - \exp\left(\log\left(g_r(\boldsymbol{x}_t)\right)\right)\right|$$

$$\le \exp\left(\max\left\{f_{(1,r)}^{\mathrm{2NN}}(\boldsymbol{x}_t), \log\left(g_r(\boldsymbol{x}_t)\right)\right\}\right) \left|f_{(1,r)}^{\mathrm{2NN}}(\boldsymbol{x}_t) - \log\left(g_r(\boldsymbol{x}_t)\right)\right|$$

$$\le \exp\left(\log g_r(\boldsymbol{x}_t) + \frac{1}{4}\right) \frac{1}{4T_r}$$

$$\le e^{1/4} \cdot T_r \cdot \frac{1}{4T_r} < \frac{1}{2}.$$

Noticing $g_r(\boldsymbol{x}_t) \in \mathbb{N}_+$, we have $\left[\exp\left(f_{(1,r)}^{\mathrm{2NN}}(\boldsymbol{x}_t)\right)\right] = g_r(\boldsymbol{x}_t)$, which implies:

$$\widetilde{f_{(1,r)}^{\mathrm{2NN}}}(\boldsymbol{x}_t) = \log\left[\exp\left(f_{(1,r)}^{\mathrm{2NN}}\right)\right] = \log g_r(\boldsymbol{x}_t).$$

Consequently, in order to construct the form $(\mathbf{0}^\top, \log g_1(\boldsymbol{x}_t), \cdots, \log g_M(\boldsymbol{x}_t), \log 2)^\top$, we need to arrange the parameters $a_k^{(1,r)}$, $\boldsymbol{b}_k^{(1,r)}$, and $c_k^{(1,r)}$ ($k \in [m_r], r \in [M]$) appropriately.

Denote $\bar{\boldsymbol{b}}_k^{(1,r)} = (\boldsymbol{b}_k^{(1,r)\top}, \mathbf{0}^\top)^\top \in \mathbb{R}^D$ for $k \in [m_r], r \in [M]$. Consider the following two-layer neural network with $1 + \sum_{r=1}^M m_r$ neurons defined on $\mathbb{R}^D$:

$$\mathbf{FFN}^{(1)}(\boldsymbol{x}) = \sum_{r=1}^M \sum_{1+\sum_{j=0}^{r-1} m_j \le k \le \sum_{j=0}^r m_j} \boldsymbol{e}_{D-M+r-1} a_k^{(1,r)} \sigma\left(\bar{\boldsymbol{b}}_k^{(1,r)\ \top} \boldsymbol{x} + c_k^{(1,r)}\right)$$

$$+ \boldsymbol{e}_D \cdot 1 \cdot \sigma(0 + \log 2).$$

It is easy to verify that for any $x_t^{(1/2)}$, it holds that

$$\mathbf{FFN}^{(1)}(x_t^{(1/2)}) = \mathbf{FFN}^{(1)}(x_t^{(0)})$$

$$= \sum_{r=1}^{M} \sum_{1+\sum_{j=0}^{r-1} m_j \leq k \leq \sum_{j=0}^{r} m_j} e_{D-M+r-1} a_k^{(1,r)} \sigma\left(\bar{b}_k^{(1,r)\top} x_t^{(0)} + c_k^{(1,r)}\right) + e_D \cdot 1 \cdot \sigma(0 + \log 2)$$

$$= \sum_{r=1}^{M} \sum_{1+\sum_{j=0}^{r-1} m_j \leq k \leq \sum_{j=0}^{r} m_j} e_{D-M+r-1} a_k^{(1,r)} \sigma\left(b_k^{(1,r)\top} x_t + c_k^{(1,r)}\right) + e_D \cdot 1 \cdot \sigma(0 + \log 2)$$

$$= \sum_{r=1}^{M} e_{D-M+r-1} f_r^{\text{2NN}}(x_t) + e_D \log 2$$

$$= (\mathbf{0}_d^\top, f_{(1,1)}^{\text{2NN}}(x_t), \cdots, f_{(1,M)}^{\text{2NN}}(x_t), \log 2)^\top \in \mathbb{R}^D.$$

Moreover, it satisfies that

$$\widetilde{\mathbf{FFN}}^{(1)}(x_t^{(1/2)}) = \log\left[\exp\left(\mathbf{FFN}^{(1)}(x_t^{(1/2)})\right)\right]$$

$$= (\mathbf{0}_d^\top, \log\left[\exp\left(f_{(1,1)}^{\text{2NN}}(x_t)\right)\right], \cdots, \log\left[\exp\left(f_{(1,M)}^{\text{2NN}}(x_t)\right)\right], \log 2, \mathbf{0}^\top)^\top$$

$$= (\mathbf{0}_d^\top, \log g_1(x_t), \cdots, \log g_M(x_t), \log 2)^\top.$$

Thus, we have achieved our goal in this step:

$$x_t^{(1)} = x_t^{(1/2)} + \widetilde{\mathbf{FFN}}^{(1)}(x_t^{(1/2)}) = (x_t^\top, \mathbf{0}^\top, \log g_1(x_t), \cdots, \log g_M(x_t), \log 2)^\top.$$

As established in the proof above, the width $m$ must satisfy:

$$m \geq 1 + \sum_{r=1}^{M} m_r = \begin{cases} \tilde{\Omega}\left(\sum_{r=1}^{M} \|g_r\|_{\mathcal{B}}^2\right), & \texttt{type} = \texttt{lin} \\ \tilde{\Omega}\left(\sum_{r=1}^{M} \|\log g_r\|_{\mathcal{B}}^2 T_r^2\right), & \texttt{type} = \texttt{log} \end{cases}.$$

**Step III.** Extract the adaptive memories by the second Attn layer.

We consider to use $H_k$ attention heads (from $\sum_{i=1}^{k-1} H_i + 1$-th head to $\sum_{i=1}^{k} H_i$-th head) to extract it, and it satisfies to $\sum_{k=1}^{M} H_k = H$.

For simplicity, we denote the following projection matrices in $\mathbb{R}^{D \times D}$:

$$P^{(k)} := (\mathbf{0}_{d \times kd} \quad I_{d \times d} \quad \mathbf{0}) \in \mathbb{R}^{d \times D}, \quad 1 \leq k \leq M;$$

$$P_\perp^{(k)} := \begin{pmatrix} I_{kd \times kd} & \mathbf{0}_{d \times d} & \mathbf{0} \\ \mathbf{0} & \mathbf{0}_{d \times d} & I_{(D-(k+1)d) \times (D-(k+1)d)} \end{pmatrix} \in \mathbb{R}^{(D-d) \times D}, \quad 1 \leq k \leq M;$$

$$Q^{(M)} := \left(I_{(M+1)d \times (M+1)d} \quad \mathbf{0}\right) \in \mathbb{R}^{(M+1)d \times D}.$$

Now we consider the extraction of $k$-th adaptive memory $x_{t-g_k(x_t)}$ $(1 \leq k \leq M)$.

- **Case** $\texttt{type} = \texttt{lin}$.

  By Lemma F.2, for any rate $n \in \mathbb{N}_+$, there exists a constant $C(n)$ and a function

  $$\phi_k^{\exp}(t; B) = \sum_{\sum_{i=1}^{k-1} H_i + 1 \leq h \leq \sum_{i=1}^{k} H_i} \alpha_h \exp(-\beta_h(t - B))$$

  $$= \sum_{\sum_{i=1}^{k-1} H_i + 1 \leq h \leq \sum_{i=1}^{k} H_i} \alpha_h \exp\left(\beta_h B - \beta_h t\right)$$

such that $\beta_h > 0$ and

$$\sup_{1 \leq B \leq T_k} \left\| \mathbb{I}\{\cdot = B\} - \phi_k^{\exp}(\cdot; B) \right\|_{\ell_1(\mathbb{N})} \leq \frac{C(n)e^{0.01(n+1)T_k}}{H_k^n}.$$

Moreover, Noticing that $1 \leq g_k(\boldsymbol{x}_t) \leq T_k$ holds for any $\boldsymbol{X} = (\boldsymbol{x}_t)_{t \in \mathbb{Z}} \in \mathcal{X}$, the following holds:

$$\sup_{\boldsymbol{X}} \left\| \mathbb{I}\{\cdot = g_k(\boldsymbol{x}_t)\} - \phi_k^{\exp}(\cdot; g_k(\boldsymbol{x}_t)) \right\|_{\ell_1(\mathbb{N})}$$

$$\leq \sup_{1 \leq B \leq T_k} \left\| \mathbb{I}\{\cdot = B\} - \phi_k^{\exp}(\cdot; B) \right\|_{\ell_1(\mathbb{N})} \leq \frac{C(n)e^{0.01(n+1)T_k}}{H_k^n}.$$

Therefore, for these attention heads $(\sum_{i=1}^{k-1} H_i + 1 \leq h \leq \sum_{i=1}^{k} H_i)$, we can choose:

$$p^{(2,h)} = \beta_h, \quad \boldsymbol{W}_O^{(1)} = \boldsymbol{I}_{D \times D}, \quad \boldsymbol{W}_V^{(2,h)} = \alpha_h \boldsymbol{\delta}_{(k+1,1)}^{(d \times d)} \in \mathbb{R}^{D \times D},$$

$$\boldsymbol{W}_Q^{(2,h)} = \sqrt{\beta_h} \boldsymbol{\delta}_{(D-M+k-1,1)}^{(1 \times 1)} \in \mathbb{R}^{D \times D}, \quad \boldsymbol{W}_K^{(2,h)} = \sqrt{\beta_h} \boldsymbol{\delta}_{(D,1)}^{(1 \times 1)} \in \mathbb{R}^{D \times D},$$

where $\boldsymbol{\delta}_{(p_1,p_2)}^{(r \times r)}$ means that: it equals to $\boldsymbol{I}_{r \times r}$ for the $(p_1, p_2)$-th $r \times r$ blocks, and $\boldsymbol{0}_{r \times r}$ for the other $r \times r$ blocks.

Then it is easy to verify:

$$\left\langle \boldsymbol{W}_Q^{(2,h)} \boldsymbol{x}_t^{(1)}, \boldsymbol{W}_K^{(2,h)} \boldsymbol{x}_{t-s}^{(1)} \right\rangle + p^{(2,h)} \phi_{\lin}(s) = -\beta_h \left( s - g_k(\boldsymbol{x}_t) \right),$$

which implies:

$$\sum_{h=\sum_{i=1}^{k-1} H_i + 1}^{\sum_{i=1}^{k} H_i} \mathbf{Attn}_t^{(2,h)}(\boldsymbol{X}^{(1)}) = \sum_{h=\sum_{i=1}^{k-1} H_i + 1}^{\sum_{i=1}^{k} H_i} \alpha_h \sum_{s=0}^{+\infty} e^{-\beta_h(s-g_k(\boldsymbol{x}_t))} \begin{pmatrix} \boldsymbol{0}_{kd} \\ \boldsymbol{x}_{t-s} \\ \boldsymbol{0} \end{pmatrix} \in \mathbb{R}^D,$$

Then it holds that:

$$\boldsymbol{P}^{(k)} \sum_{h=\sum_{i=1}^{k-1} H_i + 1}^{\sum_{i=1}^{k} H_i} \mathbf{Attn}_t^{(2,h)}(\boldsymbol{X}^{(0)}) = \sum_{h=\sum_{i=1}^{k-1} H_i + 1}^{\sum_{i=1}^{k} H_i} \alpha_h \sum_{s=0}^{+\infty} e^{-\beta_h(s-g_k(\boldsymbol{x}_t))} \boldsymbol{x}_{t-s},$$

$$\boldsymbol{P}_\perp^{(k)} \sum_{h=\sum_{i=1}^{k-1} H_i + 1}^{\sum_{i=1}^{k} H_i} \mathbf{Attn}_t^{(2,h)}(\boldsymbol{X}^{(0)}) = \boldsymbol{0},$$

moreover, the following estimate holds:

$$\left\| \sum_{h=\sum_{i=1}^{k-1} H_i + 1}^{\sum_{i=1}^{k} H_i} \boldsymbol{P}^{(k)} \mathbf{Attn}_t^{(2,h)}(\boldsymbol{X}) - \boldsymbol{x}_{t-g_k(\boldsymbol{x}_t)} \right\|_2$$

$$= \left\| \sum_{h=\sum_{i=1}^{k-1} H_i + 1}^{\sum_{i=1}^{k} H_i} \alpha_h \sum_{s=0}^{+\infty} e^{-\beta_h(s-g_k(\boldsymbol{x}_t))} \boldsymbol{x}_{t-s} - \boldsymbol{x}_{t-g_k(\boldsymbol{x}_t)} \right\|_2$$

$$
= \left\| \sum_{s=0}^{+\infty} \left( \sum_{h=\sum_{i=1}^{k-1} H_i + 1}^{\sum_{i=1}^{k} H_i} \alpha_h e^{-\beta_h(s - g_k(\boldsymbol{x}_t))} - \mathbb{I}\{s = g_k(\boldsymbol{x}_t)\} \right) \boldsymbol{x}_{t-s} \right\|_2
$$

$$
\leq \sum_{s=0}^{+\infty} \left| \sum_{h=\sum_{i=1}^{k-1} H_i + 1}^{\sum_{i=1}^{k} H_i} \alpha_h e^{-\beta_h(s - g_k(\boldsymbol{x}_t))} - \mathbb{I}\{s = g_k(\boldsymbol{x}_t)\} \right|
$$

$$
= \left\| \phi_k^{\mathrm{exp}}(\cdot; g_k(\boldsymbol{x}_t)) - \mathbb{I}\{\cdot = g_k(\boldsymbol{x}_t)\} \right\|_{\ell_1(\mathbb{N})} \leq \frac{C(n)e^{0.01(n+1)T_k}}{H_k^n}.
$$

- **Case** `type` $= \log$.

  By Lemma F.5, for any rate $n \in \mathbb{N}_+$, there exists a constant $C(n)$ and a function

$$
\phi_k^{\mathrm{poly}}(t; B) = \sum_{\sum_{i=1}^{k-1} H_i + 1 \leq h \leq \sum_{i=1}^{k} H_i} \alpha_h (t/B)^{-\beta_h}
$$

$$
= \sum_{\sum_{i=1}^{k-1} H_i + 1 \leq h \leq \sum_{i=1}^{k} H_i} \alpha_h \exp\left( \beta_h \log B - \beta_h \log t \right)
$$

  such that $\beta_h > 1$ and

$$
\sup_{1 \leq B \leq T_k} \left\| \mathbb{I}\{\cdot = B\} - \phi_k^{\mathrm{poly}}(\cdot; B) \right\|_{\ell_1(\mathbb{N}_+)} \leq \frac{C(n)T_k^{1.01(n+1)}}{H_k^n}.
$$

  Moreover, Noticing that $1 \leq g_k(\boldsymbol{x}_t) \leq T_k$ holds for any $\boldsymbol{X} = (\boldsymbol{x}_t)_{t \in \mathbb{Z}} \in \mathcal{X}$, the following holds:

$$
\sup_{\boldsymbol{X}} \left\| \mathbb{I}\{\cdot = g_k(\boldsymbol{x}_t)\} - \phi_k^{\mathrm{poly}}(\cdot; g_k(\boldsymbol{x}_t)) \right\|_{\ell_1(\mathbb{N}_+)}
$$

$$
\leq \sup_{1 \leq B \leq T_k} \left\| \mathbb{I}\{\cdot = B\} - \phi_k^{\mathrm{poly}}(\cdot; B) \right\|_{\ell_1(\mathbb{N}_+)} \leq \frac{C(n)T_k^{1.01(n+1)}}{H_k^n}.
$$

  Therefore, for these attention heads ($\sum_{i=1}^{k-1} H_i + 1 \leq h \leq \sum_{i=1}^{k} H_i$), we can choose:

$$
p^{(2,h)} = \beta_h, \quad \boldsymbol{W}_O^{(1)} = \boldsymbol{I}_{D \times D}, \quad \boldsymbol{W}_V^{(2,h)} = \alpha_h \boldsymbol{\delta}_{(k+1,1)}^{(d \times d)} \in \mathbb{R}^{D \times D},
$$

$$
\boldsymbol{W}_Q^{(2,h)} = \sqrt{\beta_h} \boldsymbol{\delta}_{(D-M+k-1,1)}^{(1 \times 1)} \in \mathbb{R}^{D \times D}, \quad \boldsymbol{W}_K^{(2,h)} = \frac{\sqrt{\beta_h}}{\log 2} \boldsymbol{\delta}_{(D,1)}^{(1 \times 1)} \in \mathbb{R}^{D \times D},
$$

  where $\boldsymbol{\delta}_{(p_1, p_2)}^{(r \times r)}$ means that: it equals to $\boldsymbol{I}_{r \times r}$ for the $(p_1, p_2)$-th $r \times r$ blocks, and $\boldsymbol{0}_{r \times r}$ for the other $r \times r$ blocks.

  Then it is easy to verify:

$$
\left\langle \boldsymbol{W}_Q^{(2,h)} \boldsymbol{x}_t^{(1)}, \boldsymbol{W}_K^{(2,h)} \boldsymbol{x}_{t-s}^{(1)} \right\rangle + p^{(2,h)} \phi_{\log}(s) = -\beta_h \log\left( s/g_k(\boldsymbol{x}_t) \right),
$$

  which implies:

$$
\sum_{h=\sum_{i=1}^{k-1} H_i + 1}^{\sum_{i=1}^{k} H_i} \mathbf{Attn}_t^{(2,h)}(\boldsymbol{X}^{(1)}) = \sum_{h=\sum_{i=1}^{k-1} H_i + 1}^{\sum_{i=1}^{k} H_i} \alpha_h \sum_{s=0}^{+\infty} (s/g_k(\boldsymbol{x}_t))^{-\beta_h} \begin{pmatrix} \boldsymbol{0}_{kd} \\ \boldsymbol{x}_{t-s} \\ \boldsymbol{0} \end{pmatrix} \in \mathbb{R}^D,
$$

Then it holds that:

$$\boldsymbol{P}^{(k)} \sum_{h=\sum_{i=1}^{k-1} H_i+1}^{\sum_{i=1}^{k} H_i} \mathbf{Attn}_t^{(2,h)}(\boldsymbol{X}^{(0)}) = \sum_{h=\sum_{i=1}^{k-1} H_i+1}^{\sum_{i=1}^{k} H_i} \alpha_h \sum_{s=0}^{+\infty}(s/g_k(\boldsymbol{x}_t))^{-\beta_h}\boldsymbol{x}_{t-s},$$

$$\boldsymbol{P}_\perp^{(k)} \sum_{h=\sum_{i=1}^{k-1} H_i+1}^{\sum_{i=1}^{k} H_i} \mathbf{Attn}_t^{(2,h)}(\boldsymbol{X}^{(0)}) = \boldsymbol{0},$$

moreover, the following estimate holds:

$$\left\| \sum_{h=\sum_{i=1}^{k-1} H_i+1}^{\sum_{i=1}^{k} H_i} \boldsymbol{P}^{(k)} \mathbf{Attn}_t^{(2,h)}(\boldsymbol{X}) - \boldsymbol{x}_{t-g_k(\boldsymbol{x}_t)} \right\|_2$$

$$= \left\| \sum_{h=\sum_{i=1}^{k-1} H_i+1}^{\sum_{i=1}^{k} H_i} \alpha_h \sum_{s=0}^{+\infty}(s/g_k(\boldsymbol{x}_t))^{-\beta_h}\boldsymbol{x}_{t-s} - \boldsymbol{x}_{t-g_k(\boldsymbol{x}_t)} \right\|_2$$

$$= \left\| \sum_{s=0}^{+\infty} \left( \sum_{h=\sum_{i=1}^{k-1} H_i+1}^{\sum_{i=1}^{k} H_i} \alpha_h (s/g_k(\boldsymbol{x}_t))^{-\beta_h} - \mathbb{I}\{s = g_k(\boldsymbol{x}_t)\} \right) \boldsymbol{x}_{t-s} \right\|_2$$

$$\leq \sum_{s=0}^{+\infty} \left| \sum_{h=\sum_{i=1}^{k-1} H_i+1}^{\sum_{i=1}^{k} H_i} \alpha_h (s/g_k(\boldsymbol{x}_t))^{-\beta_h} - \mathbb{I}\{s = g_k(\boldsymbol{x}_t)\} \right|$$

$$= \left\| \phi_k^{\text{poly}}(\cdot; g_k(\boldsymbol{x}_t)) - \mathbb{I}\{\cdot = g_k(\boldsymbol{x}_t)\} \right\|_{\ell_1(\mathbb{N}_+)} \leq \frac{C(n)T_k^{1.01(n+1)}}{H_k^n}.$$

Then we combine the estimate for all $k \in [M]$ for these two cases. It holds that

$$\left\| \boldsymbol{Q}^{(M)}\boldsymbol{x}_t^{(3/2)} - \begin{pmatrix} \boldsymbol{x}_t \\ \boldsymbol{x}_{t-g_1(\boldsymbol{x}_t)} \\ \vdots \\ \boldsymbol{x}_{t-g_M(\boldsymbol{x}_t)} \end{pmatrix} \right\|_2$$

$$= \left\| \begin{pmatrix} \boldsymbol{x}_t \\ \boldsymbol{0}_{Md} \end{pmatrix} + \sum_{h=1}^{M} \boldsymbol{Q}^{(M)}\mathbf{Attn}_t^{(2,h)}(\boldsymbol{X}^{(1)}) - \begin{pmatrix} \boldsymbol{x}_t \\ \boldsymbol{x}_{t-g_1(\boldsymbol{x}_t)} \\ \vdots \\ \boldsymbol{x}_{t-g_M(\boldsymbol{x}_t)} \end{pmatrix} \right\|_2$$

$$= \left\| \sum_{h=1}^{M} \boldsymbol{Q}^{(M)}\mathbf{Attn}_t^{(2,h)}(\boldsymbol{X}) - \begin{pmatrix} \boldsymbol{0}_d \\ \boldsymbol{x}_{t-g_1(\boldsymbol{x}_t)} \\ \vdots \\ \boldsymbol{x}_{t-g_M(\boldsymbol{x}_t)} \end{pmatrix} \right\|_2$$

$$= \left\| \sum_{k=1}^{M} \left( \sum_{h=\sum_{i=1}^{k-1} H_i+1}^{\sum_{i=1}^{k} H_i} \boldsymbol{Q}^{(M)}\mathbf{Attn}_t^{(2,h)}(\boldsymbol{X}) - \begin{pmatrix} \boldsymbol{0}_{kd} \\ \boldsymbol{x}_{t-g_k(\boldsymbol{x}_t)} \\ \boldsymbol{0} \end{pmatrix} \right) \right\|_2$$

$$\leq \sum_{k=1}^{M} \left\| \sum_{h=\sum_{i=1}^{k-1} H_i+1}^{\sum_{i=1}^{k} H_i} \boldsymbol{Q}^{(M)}\mathbf{Attn}_t^{(2,h)}(\boldsymbol{X}) - \begin{pmatrix} \boldsymbol{0}_{kd} \\ \boldsymbol{x}_{t-g_k(\boldsymbol{x}_t)} \\ \boldsymbol{0} \end{pmatrix} \right\|_2$$

$$= \sum_{k=1}^{M} \left\| \sum_{h=\sum_{i=1}^{k-1} H_i+1}^{\sum_{i=1}^{k} H_i} \boldsymbol{P}^{(k)} \mathbf{Attn}_t^{(2,h)}(\boldsymbol{X}) - \boldsymbol{x}_{t-g_k(\boldsymbol{x}_t)} \right\|_2$$

$$\leq \mathcal{E}_{\text{Attn}}(\texttt{type}) := \begin{cases} \dfrac{C(n)e^{0.01(n+1)T_k}}{H_k^n}, & \texttt{type} = \text{lin} \\ \dfrac{C(n)T_k^{1.01(n+1)}}{H_k^n}, & \texttt{type} = \text{log} \end{cases}.$$

Similar to the proof of Theorem B.1, we choose the head number:

$$H_k = \frac{e^{0.01T_k}}{\sum_{j=1}^{M} e^{0.01T_j}} H, \quad k \in [M], \quad \texttt{type} = \text{lin};$$

$$H_k = \frac{T_k^{1.01}}{\sum_{j=1}^{M} T_j^{1.01}} H, \quad k \in [M], \quad \texttt{type} = \text{log}.$$

Thus, we obtain the final bound in Step III:

$$\mathcal{E}_{\text{Attn}}^{\text{Soft}}(\texttt{type}) \leq \begin{cases} \dfrac{C(n)}{H^n} \left( \sum_{k=1}^{M} e^{0.01T_k} \right)^{n+1}, & \texttt{type} = \text{lin} \\ \dfrac{C(n)}{H^n} \left( \sum_{k=1}^{M} T_k^{1.01} \right)^{n+1}, & \texttt{type} = \text{log} \end{cases}.$$

Furthermore, by choosing $\mathcal{E}_{\text{Attn}}(\texttt{type}) \leq 1$, it holds that

$$\left\| \boldsymbol{Q}^{(M)} \boldsymbol{x}_t^{(3/2)} \right\|_\infty \leq \left\| \boldsymbol{Q}^{(M)} \boldsymbol{x}_t^{(3/2)} - \begin{pmatrix} \boldsymbol{x}_t \\ \boldsymbol{x}_{t-g_1(\boldsymbol{x}_t)} \\ \vdots \\ \boldsymbol{x}_{t-g_M(\boldsymbol{x}_t)} \end{pmatrix} \right\|_\infty + \left\| \begin{pmatrix} \boldsymbol{x}_t \\ \boldsymbol{x}_{t-g_1(\boldsymbol{x}_t)} \\ \vdots \\ \boldsymbol{x}_{t-g_M(\boldsymbol{x}_t)} \end{pmatrix} \right\|_\infty$$

$$\leq \mathcal{E}_{\text{Attn}}(\texttt{type}) + 1 \leq 2.$$

**Step IV.** Approximate the nonlinear function by 2-nd FFN layer.

In this step, we aim to approximate the function $f$ using two-layer network. By Lemma G.6, there exists a two-layer neural network with $m$ neurons

$$f_{(2)}^{\text{2NN}}(\boldsymbol{x}) = \sum_{k=1}^{m} a_k^{(2)} \sigma \left( \boldsymbol{b}_k^{(2)\top} \boldsymbol{x} + c_k^{(2)} \right)$$

defined on $\mathbb{R}^{(M+1)d}$ such that

$$\left\| f - f_{(2)}^{\text{2NN}} \right\|_{L^\infty([-2,2]^{(M+1)d})} \leq \tilde{\mathcal{O}} \left( \frac{\|f\|_{\mathcal{B}}}{\sqrt{m}} \right).$$

Denote $\bar{\boldsymbol{b}}_k^{(2)} = (\boldsymbol{b}_k^{(2)\top}, \boldsymbol{0}^\top)^\top \in \mathbb{R}^D$ for $k \in [m]$. And we consider the following two-layer neural network with $m$ neurons defined on $\mathbb{R}^D$:

$$\text{FFN}^{(2)}(\boldsymbol{x}) := \sum_{k=1}^{m} a_k^{(2)} \sigma \left( \bar{\boldsymbol{b}}_k^{(2)\top} \boldsymbol{x} + c_k^{(2)} \right).$$

It is easy to verify:

$$\mathrm{FFN}^{(2)}\left(\boldsymbol{x}_t^{(3/2)}\right) = f_{(2)}^{2\mathrm{NN}}\left(\boldsymbol{Q}^{(M)}\boldsymbol{x}_t^{(3/2)}\right).$$

Moreover,

$$\mathcal{E}_{\mathrm{FFN}}^{(2)} := \left\|f - \mathrm{FFN}^{(2)}\right\|_{L^\infty([-2,2]^{(M+1)d})} \leq \tilde{\mathcal{O}}\left(\frac{\|f\|_{\mathcal{B}}}{\sqrt{m}}\right).$$

**The final bound.**

For any $t$ and $\|\boldsymbol{X}\| \in \mathcal{X}$, it holds that

$$
\begin{aligned}
&\left\|\mathbf{H}_t(\boldsymbol{X}) - \boldsymbol{x}_t^{(2)}\right\| = \left|f\left(\boldsymbol{x}_t, \boldsymbol{x}_{t-g_1(\boldsymbol{x}_t)}, \cdots \boldsymbol{x}_{t-g_M(\boldsymbol{x}_t)}\right) - \mathrm{FFN}^{(2)}\left(\boldsymbol{x}_t^{(3/2)}\right)\right| \\
&= \left|f\left(\boldsymbol{x}_t, \boldsymbol{x}_{t-t_1}, \cdots \boldsymbol{x}_{t-t_M}\right) - f_{(2)}^{2\mathrm{NN}}\left(\boldsymbol{Q}^{(M)}\boldsymbol{x}_t^{(3/2)}\right)\right| \\
&= \left|f\left(\boldsymbol{x}_t, \boldsymbol{x}_{t-g_1(\boldsymbol{x}_t)}, \cdots \boldsymbol{x}_{t-g_M(\boldsymbol{x}_t)}\right) - f\left(\boldsymbol{Q}^{(M)}\boldsymbol{x}_t^{(3/2)}\right)\right. \\
&\quad \left. + f\left(\boldsymbol{Q}^{(M)}\boldsymbol{x}_t^{(3/2)}\right) - f_{(2)}^{2\mathrm{NN}}\left(\boldsymbol{Q}^{(M)}\boldsymbol{x}_t^{(3/2)}\right)\right| \\
&\leq \left|f\left(\boldsymbol{x}_t, \boldsymbol{x}_{t-t_1}, \cdots \boldsymbol{x}_{t-t_M}\right) - f\left(\boldsymbol{Q}^{(M)}\boldsymbol{x}_t^{(3/2)}\right)\right| + \left|f\left(\boldsymbol{Q}^{(M)}\boldsymbol{x}_t^{(3/2)}\right) - f_{(2)}^{2\mathrm{NN}}\left(\boldsymbol{Q}^{(M)}\boldsymbol{x}_t^{(3/2)}\right)\right| \\
&\leq \|f\|_{\mathrm{Lip}}\left\|\left(\boldsymbol{x}_t^\top, \boldsymbol{x}_{t-t_1}^\top, \cdots \boldsymbol{x}_{t-t_M}^\top\right)^\top - \boldsymbol{Q}^{(M)}\boldsymbol{x}_t^{(3/2)}\right\|_2 + \left\|f - f_{(2)}^{2\mathrm{NN}}\right\|_{L^\infty([-2,2]^{(M+1)D})} \\
&\leq \|f\|_{\mathrm{Lip}} \cdot \mathcal{E}_{\mathrm{Attn}} + \mathcal{E}_{\mathrm{FFN}},
\end{aligned}
$$

where

$$\mathcal{E}_{\mathrm{FFN}} = \tilde{\mathcal{O}}\left(\frac{\|f\|_{\mathcal{B}}}{\sqrt{m}}\right);$$

$$\mathcal{E}_{\mathrm{Attn}}(\mathtt{type}) = \begin{cases} \mathcal{O}\left(\frac{C(n)}{H^n}\left(\sum_{k=1}^{M} e^{0.01T_k}\right)^{n+1}\right), & \mathtt{type} = \mathtt{lin} \\ \mathcal{O}\left(\frac{C(n)}{H^n}\left(\sum_{k=1}^{M} T_k^{1.01}\right)^{n+1}\right), & \mathtt{type} = \mathtt{log} \end{cases}.$$

Moreover, recalling our proof in Step II, we further need the hard condition:

$$m \geq \begin{cases} \tilde{\Omega}\left(\sum_{r=1}^{M} \|g_r\|_{\mathcal{B}}^2\right), & \mathtt{type} = \mathtt{lin} \\ \tilde{\Omega}\left(\sum_{r=1}^{M} \|\log g_r\|_{\mathcal{B}}^2 T_r^2\right), & \mathtt{type} = \mathtt{log} \end{cases}.$$

Due to the arbitrariness of $t$ and $\boldsymbol{X}$, the proof is completed.

$\square$

**Remark C.3.** The **core step** in this proof is **Step III**, where the extraction of the memory functions is achieved through a *a nice interaction between the temporal space (provided by RPE) and the token space (provided by DP)*. Specifically, the memory functions $g_i(\boldsymbol{x}_t)$ (in token space) are mapped into the temporal space $s$, resulting in the form of $\boldsymbol{x}_{s-g_i(\boldsymbol{x}_t)}$ for DP Attn with lin-RPE or $\log(s/g_i(\boldsymbol{x}_t))$ for DP Attn with log-RPE.

## C.2 Proof of Proposition 4.2

For Proposition 16, we denote the following one-layer Attn hypothesis class:

$$\mathcal{ATTN}^{\texttt{type}}_{(1,H)} := \left\{ \mathbf{Attn} : \mathbf{TF} \text{ is a 1-layer, } H\text{-head (normalization-free) Attn with } \texttt{type-RPE} \right\};$$

$$\mathcal{ATTN}^{\mathrm{DPF},\texttt{type}}_{(1,H)} := \left\{ \mathbf{TF} : \mathbf{Attn} \text{ is a 1-layer, } H\text{-head DP-free Attn with } \texttt{type-RPE} \right\}.$$
(16)

**Proposition C.4** (The formal version of Proposition 4.2). *Consider* 1*-layer Attn. Then, there exists a target* $\mathbf{H} \in \mathcal{H}^{adap}_{(1,1)}$ (10) *such that:*

*(A) (Attn with DP) For any* $\epsilon > 0$*, there exists a* 1*-layer Attn* $\mathbf{Attn}^{\mathrm{DP}} \in \mathcal{ATTN}^{\texttt{type}}_{(1,H)}$ *such that*

$$\left\| \mathbf{H} - \mathbf{Attn}^{\mathrm{DP}} \right\| \leq \epsilon.$$

*(B) (Attn without DP) For any* 1*-layer DP-free Attn* $\mathbf{Attn}^{\mathrm{DPF}} \in \mathcal{ATTN}^{\mathrm{DPF},\texttt{type}}_{(1,H)}$*, a uniform lower bound holds:*

$$\left\| \mathbf{H} - \mathbf{Attn}^{\mathrm{DPF}} \right\| \geq \frac{2}{3}.$$

*Proof of Proposition C.4.*
Consider the following target function $\mathrm{H} \in \mathcal{H}^{\mathrm{Adap}}_{1,1}$. Let the input sequence $\boldsymbol{X} \in \mathcal{X} = \{-1, 0, 1\}^{\mathbb{Z}}$, and we consider the target

$$y_t = \mathrm{H}_t(\boldsymbol{X}) := x_{t-g(x_t)},$$

where the adaptive memory is

$$g(x) = \begin{cases} 0, & x = -1 \\ 1, & x = 0 \\ 2, & x = 1 \end{cases} .$$

**Part (A).** The Efficiency of Attn with DP.

First, we choose the embedding dimension $D = 2$, and select simple embedding $\boldsymbol{W}_E = \boldsymbol{b}_E = (1, 0)^{\top} \in \mathbb{R}^{2 \times 1}$. Then for any input sequence $\boldsymbol{X} = (\boldsymbol{x}_t)_{t \in \mathbb{Z}}$, the token after embedding satisfies:

$$x_t^{(0)} = \boldsymbol{W}_E x_t + \boldsymbol{b}_E = (x_t, 1)^{\top}.$$

We consider one-layer normalization-free Self-attention with $\phi_{\exp}$, which has the form:

$$\mathrm{Attn}^{\mathrm{DP}}_t(\boldsymbol{X}) = \boldsymbol{W}_O \sum_{h=1}^{H} \boldsymbol{W}_V^{(1,h)} \sum_{s=0}^{+\infty} \boldsymbol{x}_{t-s}^{(0)} \exp\left( \left\langle \boldsymbol{W}_Q^{(l,h)} \boldsymbol{x}_t^{(0)}, \boldsymbol{W}_K^{(1,h)} \boldsymbol{x}_{t-s}^{(0)} \right\rangle + p^{(1,h)} s \right).$$

By Lemma F.2 (for $n = 1$), there exists a constant $C > 0$ and a function

$$\phi^{\exp}(t; B) = \sum_{h=1}^{H} \alpha_h \exp(-\beta_h(t - B)) = \sum_{h=1}^{H} \alpha_h \exp\left( \beta_h B - \beta_h t \right)$$

such that $\beta_h > 0$ and

$$\sup_{0 \leq B \leq 2} \left\| \mathbb{I}\{\cdot = B\} - \phi^{\exp}(\cdot; B) \right\|_{\ell_1(\mathbb{N})} \leq \frac{C e^{0.01 \cdot 2 \cdot 2}}{H} = \mathcal{O}\left( \frac{1}{H} \right).$$

Moreover, Noticing that $0 \leq g(x_t) \leq 2$ holds for any $\boldsymbol{X} = (x_t)_{t \in \mathbb{Z}} \in \mathcal{X}$, the following holds:

$$\sup_{\boldsymbol{X}} \left\| \mathbb{I}\{\cdot = g(x_t)\} - \phi^{\exp}(\cdot; g(x_t)) \right\|_{\ell_1(\mathbb{N})}$$

$$\leq \sup_{0 \leq B \leq 2} \left\| \mathbb{I}\{\cdot = B\} - \phi^{\exp}(\cdot; B) \right\|_{\ell_1(\mathbb{N})} \leq \mathcal{O}\left(\frac{1}{H}\right).$$

Therefore, for attention heads ($1 \leq h \leq H$), we can choose:

$$p^{(1,h)} = \beta_h, \quad \boldsymbol{W}_O^{(1)} = (1,0)^\top \in \mathbb{R}^{2\times 1}, \quad \boldsymbol{W}_V^{(1,h)} = \alpha_h \boldsymbol{\delta}_{(1,1)}^{(1\times 1)} \in \mathbb{R}^{2\times 2},$$
$$\boldsymbol{W}_Q^{(1,h)} = \sqrt{\beta_h}(1,1)^\top \in \mathbb{R}^{2\times 1}, \quad \boldsymbol{W}_K^{(1,h)} = \sqrt{\beta_h}(0,1)^\top \in \mathbb{R}^{2\times 1},$$

where $\boldsymbol{\delta}_{(p_1,p_2)}^{(r\times r)}$ means that: it equals to $\boldsymbol{I}_{r\times r}$ for the $(p_1, p_2)$-th $r \times r$ blocks, and $\boldsymbol{0}_{r\times r}$ for the other $r \times r$ blocks.

Then it is easy to verify:

$$\left\langle \boldsymbol{W}_Q^{(1,h)} \boldsymbol{x}_t^{(0)}, \boldsymbol{W}_K^{(1,h)} \boldsymbol{x}_{t-s}^{(0)} \right\rangle + p^{(2,h)} s = -p^{(1,h)}\left(s - (x_t + 1)\right) = -p^{(1,h)}\left(s - g(x_t)\right).$$

Thus, the following estimate holds:

$$\left\| \mathrm{Attn}_t^{\mathrm{DP}}(\boldsymbol{X}) - x_{t-g(x_t)} \right\|_2$$

$$= \left\| \sum_{h=1}^{H} \alpha_h \sum_{s=0}^{+\infty} e^{-\beta_h(s-g(x_t))} x_{t-s} - x_{t-g(\boldsymbol{x}_t)} \right\|_2$$

$$= \left\| \sum_{s=0}^{+\infty} \left( \sum_{h=1}^{H} \alpha_h e^{-\beta_h(s-g(x_t))} - \mathbb{I}\{s = g(x_t)\} \right) x_{t-s} \right\|_2$$

$$\leq \sum_{s=0}^{+\infty} \left| \sum_{h=1}^{H} \alpha_h e^{-\beta_h(s-g_k(x_t))} - \mathbb{I}\{s = g(x_t)\} \right|$$

$$= \left\| \phi^{\exp}(\cdot; g(x_t)) - \mathbb{I}\{\cdot = g(x_t)\} \right\|_{\ell_1(\mathbb{N})} \leq \mathcal{O}\left(\frac{1}{H}\right).$$

Due to the arbitrariness of $t$ and $\boldsymbol{X}$, the proof is completed: for any $\epsilon > 0$, we only need to use $H = \Omega(1/\epsilon)$ heads to approximate it.

**Part (B).** The Hardness Result of Attn without DP.

We consider one-layer Dot-product-free Self-attention with $\phi_{\exp}$ or $\phi_{\log}$. For any input $\boldsymbol{X}$, the corresponding output can be written as:

$$\mathrm{Attn}_t^{\mathrm{DPF}}(\boldsymbol{X}) = \sum_{s=0}^{+\infty} \rho_s (W_E x_{t-s} + b_E).$$

For simplicity, we denote:

$$x_t^{(0)} = W_E x_t + b_E,$$

then we have the following estimate:

$$|||\mathrm{H} - \mathrm{Attn}^{\mathrm{DPF}}||| = \sup_t \sup_{\boldsymbol{X}} |\mathrm{H}_t^{\mathrm{DPF}}(\boldsymbol{X}) - \mathrm{Attn}_t(\boldsymbol{X})| \geq \sup_{\boldsymbol{X}} |\mathrm{H}_0(\boldsymbol{X}) - \mathrm{Attn}_0^{\mathrm{DPF}}(\boldsymbol{X})|$$

$$= \sup_{(\cdots, x_{-2}, x_{-1}, x_0)} \left| x_{-g(x_0)} - \sum_{s=0}^{+\infty} \rho_s x_{-s}^{(0)} \right|$$

$$\overset{\text{fixing } x_{-s} = 0 \text{ for } s \geq 3}{\geq} \sup_{(x_{-2}, x_{-1}, x_0)} \left| x_{-g(x_0)} - \sum_{s=0}^{2} \rho_s x_{-s}^{(0)} \right|$$

$$\overset{x_0 \in \{-1, 0, 1\}}{=} \max \left\{ \max_{(x_{-2}, x_{-1})} \left| -1 - \left( \rho_0(-W_E + b_E) + \rho_1 x_{-1}^{(0)} + \rho_2 x_{-2}^{(0)} \right) \right|, \right.$$

$$\max_{(x_{-2}, x_{-1})} \left| x_{-g(0)} - \left( \rho_0 b_E + \rho_1 x_{-1}^{(0)} + \rho_2 x_{-2}^{(0)} \right) \right|,$$

$$\left. \max_{(x_{-2}, x_{-1})} \left| x_{-g(1)} - \left( \rho_0(W_E + b_E) + \rho_1 x_{-1}^{(0)} + \rho_2 x_{-2}^{(0)} \right) \right| \right\}$$

$$= \max_{(x_{-2}, x_{-1})} \max \left\{ \left| x_{-1} - \left( \rho_0 b_E + \rho_1 x_{-1}^{(0)} + \rho_2 x_{-2}^{(0)} \right) \right|, \right.$$

$$\left| -1 - \left( \rho_0(-W_E + b_E) + \rho_1 x_{-1}^{(0)} + \rho_2 x_{-2}^{(0)} \right) \right|,$$

$$\left. \left| x_{-2} - \left( \rho_0(W_E + b_E) + \rho_1 x_{-1}^{(0)} + \rho_2 x_{-2}^{(0)} \right) \right| \right\}$$

$$\overset{\max\{a,b,c\} \geq \frac{1}{3}(a+b+c)}{\geq} \frac{1}{3} \max_{(x_{-2}, x_{-1})} \left( \left| x_{-1} - \left( \rho_0 b_E + \rho_1 x_{-1}^{(0)} + \rho_2 x_{-2}^{(0)} \right) \right| \right.$$

$$+ \left| -1 - \left( \rho_0(-W_E + b_E) + \rho_1 x_{-1}^{(0)} + \rho_2 x_{-2}^{(0)} \right) \right|$$

$$\left. + \left| x_{-2} - \left( \rho_0(W_E + b_E) + \rho_1 x_{-1}^{(0)} + \rho_2 x_{-2}^{(0)} \right) \right| \right)$$

$$\overset{|a|+|b| \geq |a+b|}{\geq} \frac{1}{3} \max_{(x_{-2}, x_{-1})} \left( \left| x_{-1} - \left( \rho_0 b_E + \rho_1 x_{-1}^{(0)} + \rho_2 x_{-2}^{(0)} \right) \right| \right.$$

$$\left. + \left| -1 + x_{-2} - 2 \left( \rho_0 b_E + \rho_1 x_{-1}^{(0)} + \rho_2 x_{-2}^{(0)} \right) \right| \right)$$

$$\geq \frac{1}{3} \max_{(x_{-2}, x_{-1})} \left( \left| x_{-1} - \left( \rho_0 b_E + \rho_1 x_{-1}^{(0)} + \rho_2 x_{-2}^{(0)} \right) \right| \right.$$

$$\left. + \left| \frac{-1 + x_{-2}}{2} - \left( \rho_0 b_E + \rho_1 x_{-1}^{(0)} + \rho_2 x_{-2}^{(0)} \right) \right| \right)$$

$$\overset{|a|+|b| \geq |a-b|}{\geq} \frac{1}{3} \max_{(x_{-2}, x_{-1})} \left| x_{-1} + \frac{1}{2} - \frac{x_{-2}}{2} \right| = \frac{2}{3}.$$

$\square$

### C.3   Proof of Proposition 4.3

In this subsection, we propose a structurally simpler yet effective alternative to traditional Dot-product in Self-attention. This alternative is proposed based on our insights into the role of Attn in facilitating interaction between the temporal-space and the token-space. Specifically, we propose a more direct structure to achieve the interaction $\phi_{\texttt{type}}(s) - \phi_{\texttt{type}}(\boldsymbol{w}^\top \boldsymbol{x}_t)$.

**Definition C.5** (TMX Transformer). We define the TMX ("$t$ minus $\boldsymbol{x}$") Transformer as follows. In standard Transformer (2), we modify the term

$$\left\langle \boldsymbol{W}_Q^{(l,h)} \boldsymbol{x}_t, \boldsymbol{W}_K^{(l,h)} \boldsymbol{x}_{t-s} \right\rangle + p^{(l,h)} \phi_{\texttt{type}}(s)$$

in Multi-head Dot-product Self-attention to the new formulation:

$$p^{(l,h)} \left( \phi_{\texttt{type}}(s) - \phi_{\texttt{type}} \left( \boldsymbol{w}^{(l,h)\,\top} \boldsymbol{x}_t \right) \right),$$

where the parameters $\boldsymbol{w}^{(l,h)} \in \mathbb{R}^{D \times 1}$.

Notice that in TMX Transformer, the revised term requires only $\mathcal{O}(D)$ parameters, much less than $\mathcal{O}(D^2)$ in standard Dot-product Transformer.

Consequently, we define the following TMX Transformer hypothesis class.

$$\mathcal{TF}_{(1,H,m)}^{\mathrm{TMX,type}} := \big\{ \mathbf{TF} : \mathbf{TF} \text{ is a 1-layer, } H\text{-head, } m\text{-width}$$

$$\text{(normalization-free) TMX Transformer with } \mathtt{type}\text{-RPE} \big\}. \tag{17}$$

**Proposition C.6** (The formal version of Proposition 4.3). *Under the same conditions in Theorem 4.1, there exists a two-layer TMX Transformer* $\mathbf{TF} \in \mathcal{TF}_{(2,H,m)}^{\mathrm{TMX,type}}$ *(17) such that it can achieve the same approximation rate as standard Transformer presented in Theorem 4.1.*

*Proof of Proposition C.6.*

It is worth noting that TMX Transformer only replaces $\left\langle \boldsymbol{W}_Q^{(l,h)} \boldsymbol{x}_t, \boldsymbol{W}_K^{(l,h)} \boldsymbol{x}_{t-s} \right\rangle + p^{(l,h)} \phi_{\mathtt{type}}(s)$

in standard Transformer with $p^{(l,h)} \left( \phi_{\mathtt{type}}(s) - \phi_{\mathtt{type}} \left( \boldsymbol{w}^{(l,h) \top} \boldsymbol{x}_t \right) \right)$. Therefore, the proof is highly similar to that of Theorem C.2. We only need to prove that TMX Attn can also achieve **Step I** and **Step III** in the proof of Theorem C.2.

**Step I.** Step I is trivial due to the same use of the residual block.

**Step III. Extract the adaptive memories by the second Attn layer.**

We still consider to use $H_k$ attention heads (from $\sum_{i=1}^{k-1} H_i + 1$-th head to $\sum_{i=1}^k H_i$-th head) to extract it, and it satisfies to $\sum_{k=1}^M H_k = H$.

Now we consider the extraction of $k$-th adaptive memory $\boldsymbol{x}_{t-g_k(\boldsymbol{x}_t)}$ $(1 \le k \le M)$.

- **Case** $\mathtt{type} = \mathrm{lin}$**.**

  For the proof of standard Transformer (the proof of Theorem C.2), for the attention heads $(\sum_{i=1}^{k-1} H_i + 1 \le h \le \sum_{i=1}^k H_i)$, we can construct specific $p^{(2,h)}, \boldsymbol{W}_Q^{(2,h)}, \boldsymbol{W}_K^{(2,h)}, \boldsymbol{W}_V^{(2,h)}$ such that

  $$\left\langle \boldsymbol{W}_Q^{(2,h)} \boldsymbol{x}_t^{(1)}, \boldsymbol{W}_K^{(2,h)} \boldsymbol{x}_{t-s}^{(1)} \right\rangle + p^{(2,h)} \phi_{\mathrm{lin}}(s) = -\beta_h \Big( s - g_k(\boldsymbol{x}_t) \Big).$$

  In this proof, we only need to prove that we can also construct specific $\boldsymbol{w}^{(l,h)}, \boldsymbol{W}_V^{(2,h)}$ such that

  $$p^{(2,h)} \left( \phi_{\mathrm{lin}}(s) - \phi_{\mathrm{lin}} \left( \boldsymbol{w}^{(2,h) \top} \boldsymbol{x}_t^{(1)} \right) \right) = -\beta_h \Big( s - g_k(\boldsymbol{x}_t) \Big).$$

  Recalling the proof of Theorem C.2,

  $$\boldsymbol{x}_t^{(1)} = (\boldsymbol{x}_t^\top, \boldsymbol{0}^\top, g_1(\boldsymbol{x}_t), \cdots, g_M(\boldsymbol{x}_t), 1)^\top \in \mathbb{R}^D.$$

  Therefore, we can choose

  $$p^{(2,h)} = \beta_h, \quad \boldsymbol{w}^{(2,h)} = \boldsymbol{\delta}_{(D-M+k-1,1)}^{(1\times1)} \in \mathbb{R}^D, \quad \boldsymbol{W}_V^{(2,h)} = \alpha_h \boldsymbol{\delta}_{(k+1,1)}^{(d\times d)} \in \mathbb{R}^{D\times D}.$$

  where $\boldsymbol{\delta}_{(p_1,p_2)}^{(r\times r)}$ means that: it equals to $\boldsymbol{I}_{r\times r}$ for the $(p_1, p_2)$-th $r \times r$ blocks, and $\boldsymbol{0}_{r\times r}$ for the other $r \times r$ blocks.

  The the following holds:

  $$p^{(2,h)} \left( \phi_{\mathrm{lin}}(s) - \phi_{\mathrm{lin}} \left( \boldsymbol{w}^{(2,h) \top} \boldsymbol{x}_t^{(1)} \right) \right)$$

$$= -p^{(2,h)} \left( s - \boldsymbol{w}^{(2,h)\,\top} \boldsymbol{x}_t^{(1)} \right) = -\beta_h \left( s - g_k(\boldsymbol{x}_t) \right).$$

- **Case** `type` $= \log$.

  For the proof of standard Transformer (the proof of Theorem C.2), for the attention heads $(\sum_{i=1}^{k-1} H_i + 1 \le h \le \sum_{i=1}^{k} H_i)$, we can construct specific $p^{(2,h)}, \boldsymbol{W}_Q^{(2,h)}, \boldsymbol{W}_K^{(2,h)}, \boldsymbol{W}_V^{(2,h)}$ such that

$$\left\langle \boldsymbol{W}_Q^{(2,h)} \boldsymbol{x}_t^{(1)}, \boldsymbol{W}_K^{(2,h)} \boldsymbol{x}_{t-s}^{(1)} \right\rangle + p^{(2,h)} \phi_{\log}(s) = -\beta_h \log \left( s/g_k(\boldsymbol{x}_t) \right).$$

  In this proof, we only need to prove that we can also construct specific $\boldsymbol{w}^{(l,h)}, \boldsymbol{W}_V^{(2,h)}$ such that

$$p^{(2,h)} \left( \phi_{\log}(s) - \phi_{\log} \left( \boldsymbol{w}^{(2,h)\,\top} \boldsymbol{x}_t^{(1)} \right) \right) = -\beta_h \log \left( s/g_k(\boldsymbol{x}_t) \right).$$

  Recalling the proof of Theorem C.2,

$$\boldsymbol{x}_t^{(1)} = (\boldsymbol{x}_t^\top, \boldsymbol{0}^\top, \log g_1(\boldsymbol{x}_t), \cdots, \log g_M(\boldsymbol{x}_t), \log 2)^\top.$$

  Therefore, we can choose

$$p^{(2,h)} = \beta_h, \quad \boldsymbol{w}^{(2,h)} = \boldsymbol{\delta}_{(D-M+k-1,1)}^{(1\times1)} \in \mathbb{R}^D, \quad \boldsymbol{W}_V^{(2,h)} = \alpha_h \boldsymbol{\delta}_{(k+1,1)}^{(d\times d)} \in \mathbb{R}^{D\times D}.$$

  where $\boldsymbol{\delta}_{(p_1,p_2)}^{(r\times r)}$ means that: it equals to $\boldsymbol{I}_{r\times r}$ for the $(p_1, p_2)$-th $r \times r$ blocks, and $\boldsymbol{0}_{r\times r}$ for the other $r \times r$ blocks.

  The the following holds:

$$p^{(2,h)} \left( \phi_{\log}(s) - \phi_{\log} \left( \boldsymbol{w}^{(2,h)\,\top} \boldsymbol{x}_t^{(1)} \right) \right)$$
$$= -p^{(2,h)} \log \left( s/\left( \boldsymbol{w}^{(2,h)\,\top} \boldsymbol{x}_t^{(1)} \right) \right) = -\beta_h \log \left( s/g_k(\boldsymbol{x}_t) \right).$$

The rest of the proof is exactly the same as the proof of Theorem C.2, and we do not repeat it.

$\square$

# D Proof of Section 4.3

## D.1 Proof of Theorem 4.4

In this subsection, we give the detailed proofs for the general case of $K$-adaptive, long but $M$-sparse memory:

$$\boldsymbol{y}_t = \boldsymbol{f}(\boldsymbol{x}_t, \boldsymbol{x}_{t-t_1}, \cdots, \boldsymbol{x}_{t-t_M}),$$

where the adaptive memories satisfy:

$$t_1 = g_1(\boldsymbol{x}_t);$$
$$t_2 = g_2(\boldsymbol{x}_t, \boldsymbol{x}_{t-t_1});$$
$$\cdots$$
$$t_{K+1} = g_{K+1}(\boldsymbol{x}_t, \boldsymbol{x}_{t-t_1}, \cdots, \boldsymbol{x}_{t-t_K});$$
$$\cdots$$
$$t_{K+2} = g_{K+2}(\boldsymbol{x}_t, \boldsymbol{x}_{t-t_1}, \cdots, \boldsymbol{x}_{t-t_K});$$
$$\cdots$$
$$t_M = g_{K+1}(\boldsymbol{x}_t, \boldsymbol{x}_{t-t_1}, \cdots, \boldsymbol{x}_{t-t_K}),$$

where $1 \le t_k \le T_k$ holds for any $k \in [M]$.

**Theorem D.1** (Restatement of Theorem 4.4). *For any target* $\mathbf{H} \in \mathcal{H}_{(K,M)}^{\mathrm{Adap}}$, *rate* $n \in \mathbb{N}_+$, *and* $H, m \in \mathbb{N}_+$, *there exists an L-layer* $(L = K + 1 + \mathbb{I}\{M \ge K + 1\})$ *Transformer* $\mathbf{TF} \in \mathcal{TF}_{(L,H,m)}^{\mathrm{NF},\mathtt{type}}$ (12) *and a constant* $C(n)$ *such that: if the width satisfies*

$$m \ge \begin{cases} \tilde{\Omega}\Big( \max_{i \in [K]} \vee \sum_{i=K+1}^{M} \|g_i\|_{\mathcal{B}}^2 \Big), & \mathtt{type} = \mathrm{lin}, \\ \tilde{\Omega}\Big( \max_{i \in [K]} \vee \sum_{i=K+1}^{M} \|\log g_i\|_{\mathcal{B}}^2 T_i^2 \Big), & \mathtt{type} = \log \end{cases},$$

*then the following approximation rate holds:*

$$\|\mathbf{H} - \mathbf{TF}\| \le \mathcal{E}_{\mathrm{FFN}} + \mathcal{E}_{\mathrm{Attn}}(\mathtt{type}),$$

*where* $\mathcal{E}_{\mathrm{FFN}} = \tilde{\mathcal{O}}\left( \frac{\|f\|_{\mathcal{B}}}{\sqrt{m}} \right)$ *and*

$$\mathcal{E}_{\mathrm{Attn}}(\mathtt{type}) = \begin{cases} \mathcal{O}\left( \frac{C(n)}{H^n} \sqrt{\sum_{l=1}^{K} e^{0.02(n+1)T_l} + \left(\sum_{l=K+1}^{M} e^{0.01T_l}\right)^{2n+2}} \right), & \mathtt{type} = \mathrm{lin} \\ \mathcal{O}\left( \frac{C(n)}{H^n} \sqrt{\sum_{l=1}^{K} T_l^{2.02(n+1)} + \left(\sum_{l=K+1}^{M} T_l^{1.01}\right)^{2n+2}} \right), & \mathtt{type} = \log \end{cases}.$$

*Proof of Theorem D.1.*
First, we choose the embedding dimension $D = (M + 1)(d + 1)$, and select the same embedding matrix $\boldsymbol{W}_E = (\boldsymbol{I}_d, \boldsymbol{0})^\top \in \mathbb{R}^{D \times d}, \boldsymbol{b}_E = \boldsymbol{0} =\in \mathbb{R}^D$ as the proof of Theorem C.2. Moreover, we still use the network with precision $\widetilde{\mathrm{FFN}}$ defined in Appendix C.1 to tackle the discrete values of memories.

Then for any input sequence $\boldsymbol{X} = (\boldsymbol{x}_t)_{t \in \mathbb{Z}}$, the token after embedding satisfies:

$$\boldsymbol{x}_t^{(0)} = \boldsymbol{W}_E \boldsymbol{x}_t + \boldsymbol{b}_E = (\boldsymbol{x}_t^\top, \boldsymbol{0}^\top)^\top \in \mathbb{R}^D.$$

Thus, for $L$-layer $(L = K + 1 + \mathbb{I}\{M \ge K + 1\})$ normalization-free Transformer $\mathbf{TF} \in \mathcal{TF}_{(L,H,m)}^{\mathrm{NF},\mathtt{type}}$ with $\phi_{\mathtt{type}}$, the output token $\boldsymbol{x}_t^{(K+1)}$ of $t$-th input token satisfies:

$$\boldsymbol{x}_t^{(l-1/2)} = \boldsymbol{x}_t^{(l)} + \boldsymbol{W}_O^{(l)} \sum_{h=1}^{H} \mathbf{Attn}_t^{(l,h)}(\boldsymbol{X}^{(l-1)}), \ 1 \le l \le L + 1$$

$$x_t^{(l)} = x_t^{(l-1/2)} + \widetilde{\textbf{FFN}}^{(l)}(x_t^{(l-1/2)}), \ 1 \le l \le L$$
$$x_t^{(L+1)} = \textbf{FFN}^{(L+1)}(x_t^{(L+1/2)}),$$

where

$$\textbf{Attn}_t^{(l,h)}(X) = W_V^{(l,h)} \sum_{s=0}^{+\infty} x_{t-s} \exp\left( \left\langle W_Q^{(l,h)} x_t, W_K^{(l,h)} x_{t-s} \right\rangle + p^{(l,h)} \phi_{\texttt{type}}(s) \right).$$

Since the proof of this theorem is similar to the proof of Theorem C.2, we mainly discuss the differences.

The proof can be summarized as the following process:

- **Case** $\texttt{type} = \text{lin}$.
    - **Regime** $M \ge K + 1$.
$$x_t^{(0)}$$
      Step 1. 1-st Attn $\downarrow$
$$x_t^{(1/2)} = x_t^{(0)}$$
      Step 2. 1-st FFN $\downarrow$
$$x_t^{(1)} = x_t^{(1/2)} + (\mathbf{0}^\top, t_1, \mathbf{0}_{M-1}^\top, 1)^\top$$
      Step 3. 2-st Attn $\downarrow$
$$x_t^{(3/2)} \approx x_t^{(1)} + (\mathbf{0}_d^\top, x_{t-t_1}^\top, \mathbf{0}^\top)^\top$$
      Step 4. 2-st FFN $\downarrow$
$$x_t^{(2)} = x_t^{(3/2)} + (\mathbf{0}^\top, t_2, \mathbf{0}_{M-1}^\top)^\top$$
      Step 5. 3-st Attn $\downarrow$
$$x_t^{(5/2)} \approx x_t^{(2)} + (\mathbf{0}_{2d}^\top, x_{t-t_2}^\top, \mathbf{0}^\top)^\top$$
$$\cdots$$
      Step $2K + 1$. $K + 1$-st Attn $\downarrow$
$$x_t^{(K+1/2)} \approx x_t^{(K)} + (\mathbf{0}_{Kd}^\top, x_{t-t_K}^\top, \mathbf{0}^\top)^\top$$
      Step $2K + 2$. $K + 1$-st FFN $\downarrow$
$$x_t^{(K+1)} = x_t^{(K+1/2)} + (\mathbf{0}^\top, t_{K+1}, \cdots, t_M, 0)^\top$$
      Step $2K + 3$. $K + 2$-st Attn $\downarrow$
$$x_t^{(K+3/2)} \approx x_t^{(K+1)} + (\mathbf{0}_{(K+1)d}^\top, x_{t-t_{K+1}}^\top, \cdots, x_{t-t_M}^\top, 0)^\top$$
      Step $2K + 4$. $K + 2$-st FFN $\downarrow$
$$x_t^{(K+2)} \approx f(x_t, x_{t-t_1}, \cdots, x_{t-t_M})$$
    - **Regime** $M = K$.
$$x_t^{(0)}$$
      Step 1. 1-st Attn $\downarrow$
$$x_t^{(1/2)} = x_t^{(0)}$$
      Step 2. 1-st FFN $\downarrow$
$$x_t^{(1)} = x_t^{(1/2)} + (\mathbf{0}^\top, t_1, \mathbf{0}_{M-1}^\top, 1)^\top$$
      Step 3. 2-st Attn $\downarrow$
$$x_t^{(3/2)} \approx x_t^{(1)} + (\mathbf{0}_d^\top, x_{t-t_1}^\top, \mathbf{0}^\top)^\top$$

Step 4. 2-st FFN $\downarrow$

$$\boldsymbol{x}_t^{(2)} = \boldsymbol{x}_t^{(3/2)} + (\mathbf{0}^\top, t_2, \mathbf{0}_{M-1}^\top)^\top$$

Step 5. 3-st Attn $\downarrow$

$$\boldsymbol{x}_t^{(5/2)} \approx \boldsymbol{x}_t^{(2)} + (\mathbf{0}_{2d}^\top, \boldsymbol{x}_{t-t_2}^\top, \mathbf{0}^\top)^\top$$

$$\cdots$$

Step $2K+1$. $K+1$-st Attn $\downarrow$

$$\boldsymbol{x}_t^{(K+1/2)} \approx \boldsymbol{x}_t^{(K)} + (\mathbf{0}_{Kd}^\top, \boldsymbol{x}_{t-t_K}^\top, \mathbf{0}^\top)^\top$$

Step $2K+2$. $K+1$-st FFN $\downarrow$

$$\boldsymbol{x}_t^{(K+1)} \approx \boldsymbol{f}(\boldsymbol{x}_t, \boldsymbol{x}_{t-t_1}, \cdots, \boldsymbol{x}_{t-t_M})$$

- **Case** $\texttt{type} = \log$.

  - **Regime** $M \geq K+1$.

$$\boldsymbol{x}_t^{(0)}$$

Step 1. 1-st Attn $\downarrow$

$$\boldsymbol{x}_t^{(1/2)} = \boldsymbol{x}_t^{(0)}$$

Step 2. 1-st FFN $\downarrow$

$$\boldsymbol{x}_t^{(1)} = \boldsymbol{x}_t^{(1/2)} + (\mathbf{0}^\top, \log t_1, \mathbf{0}_{M-1}^\top, \log 2)^\top$$

Step 3. 2-st Attn $\downarrow$

$$\boldsymbol{x}_t^{(3/2)} \approx \boldsymbol{x}_t^{(1)} + (\mathbf{0}_d^\top, \boldsymbol{x}_{t-t_1}^\top, \mathbf{0}^\top)^\top$$

Step 4. 2-st FFN $\downarrow$

$$\boldsymbol{x}_t^{(2)} = \boldsymbol{x}_t^{(3/2)} + (\mathbf{0}^\top, \log t_2, \mathbf{0}_{M-1}^\top)^\top$$

Step 5. 3-st Attn $\downarrow$

$$\boldsymbol{x}_t^{(5/2)} \approx \boldsymbol{x}_t^{(2)} + (\mathbf{0}_{2d}^\top, \boldsymbol{x}_{t-t_2}^\top, \mathbf{0}^\top)^\top$$

$$\cdots$$

Step $2K+1$. $K+1$-st Attn $\downarrow$

$$\boldsymbol{x}_t^{(K+1/2)} \approx \boldsymbol{x}_t^{(K)} + (\mathbf{0}_{Kd}^\top, \boldsymbol{x}_{t-\log t_K}^\top, \mathbf{0}^\top)^\top$$

Step $2K+2$. $K+1$-st FFN $\downarrow$

$$\boldsymbol{x}_t^{(K+1)} = \boldsymbol{x}_t^{(K+1/2)} + (\mathbf{0}^\top, \log t_{K+1}, \cdots, \log t_M, 0)^\top$$

Step $2K+3$. $K+2$-st Attn $\downarrow$

$$\boldsymbol{x}_t^{(K+3/2)} \approx \boldsymbol{x}_t^{(K+1)} + (\mathbf{0}_{(K+1)d}^\top, \boldsymbol{x}_{t-t_{K+1}}^\top, \cdots, \boldsymbol{x}_{t-t_M}^\top, 0)^\top$$

Step $2K+4$. $K+2$-st FFN $\downarrow$

$$\boldsymbol{x}_t^{(K+2)} \approx \boldsymbol{f}(\boldsymbol{x}_t, \boldsymbol{x}_{t-t_1}, \cdots, \boldsymbol{x}_{t-t_M})$$

  - **Regime** $M = K$.

$$\boldsymbol{x}_t^{(0)}$$

Step 1. 1-st Attn $\downarrow$

$$\boldsymbol{x}_t^{(1/2)} = \boldsymbol{x}_t^{(0)}$$

Step 2. 1-st FFN $\downarrow$

$$\boldsymbol{x}_t^{(1)} = \boldsymbol{x}_t^{(1/2)} + (\mathbf{0}^\top, \log t_1, \mathbf{0}_{M-1}^\top, \log 2)^\top$$

Step 3. 2-st Attn $\downarrow$

$$\boldsymbol{x}_t^{(3/2)} \approx \boldsymbol{x}_t^{(1)} + (\mathbf{0}_d^\top, \boldsymbol{x}_{t-t_1}^\top, \mathbf{0}^\top)^\top$$

Step 4. 2-st FFN $\downarrow$

$$x_t^{(2)} = x_t^{(3/2)} + (\mathbf{0}^\top, \log t_2, \mathbf{0}_{M-1}^\top)^\top$$

Step 5. 3-st Attn $\downarrow$

$$x_t^{(5/2)} \approx x_t^{(2)} + (\mathbf{0}_{2d}^\top, x_{t-t_2}^\top, \mathbf{0}^\top)^\top$$

$$\cdots$$

Step $2K+1$. $K+1$-st Attn $\downarrow$

$$x_t^{(K+1/2)} \approx x_t^{(K)} + (\mathbf{0}_{Kd}^\top, x_{t-t_K}^\top, \mathbf{0}^\top)^\top$$

Step $2K+2$. $K+1$-st FFN $\downarrow$

$$x_t^{(K+1)} \approx f(x_t, x_{t-t_1}, \cdots, x_{t-t_M})$$

For simplicity, we denote the following projection matrices:

$$P^{(k)} := \begin{pmatrix} \mathbf{0}_{d \times kd} & I_{d \times d} & \mathbf{0} \end{pmatrix} \in \mathbb{R}^{d \times D}, \quad 1 \le k \le M;$$

$$P_\perp^{(k)} := \begin{pmatrix} I_{kd \times kd} & \mathbf{0}_{d \times d} & \mathbf{0} \\ \mathbf{0} & \mathbf{0}_{d \times d} & I_{(D-(k+1)d) \times (D-(k+1)d)} \end{pmatrix} \in \mathbb{R}^{(D-d) \times D}, \quad 1 \le k \le M;$$

$$Q^{(k)} := \begin{pmatrix} I_{(k+1)d \times (k+1)d} & \mathbf{0} \end{pmatrix} \in \mathbb{R}^{(k+1)d \times D}, \quad 1 \le k \le M;$$

$$Q_\perp^{(k)} := \begin{pmatrix} \mathbf{0} & I_{(D-(k+1)d) \times (D-(k+1)d)} \end{pmatrix} \in \mathbb{R}^{(D-(k+1)d) \times D}, \quad 1 \le k \le M;$$

$$R := \begin{pmatrix} \mathbf{0}_{(M-K)d \times (K+1)d} & I_{(M-K)d \times (M-K)d} & \mathbf{0} \end{pmatrix} \in \mathbb{R}^{(M-K)d \times D};$$

$$R_\perp := \begin{pmatrix} I_{(K+1)d \times (K+1)d} & \mathbf{0} & \mathbf{0} \\ \mathbf{0} & \mathbf{0} & I_{(M+1) \times (M+1)} \end{pmatrix} \in \mathbb{R}^{(D-(M-K)d) \times D}.$$

**Step 1** is trivial due to the use of the residual block.

**Step 2.** In the same way as Step II in the proof of Theorem C.2, we obtain the conclusion in this step: If the width of FFN satisfies

$$m \ge \begin{cases} \tilde{\Omega}\left(\|g_1\|_\mathcal{B}^2\right), & \texttt{type} = \texttt{lin} \\ \tilde{\Omega}\left(\|\log g_1\|_\mathcal{B}^2 T_1^2\right), & \texttt{type} = \texttt{log} \end{cases},$$

then the following holds:

$$x_t^{(1)} = x_t^{(1/2)} + (\mathbf{0}^\top, t_1, \mathbf{0}_M^\top, 1)^\top.$$

Thus, **(E1)** holds for $l = 2$.

**Step 3 $\sim$ Step $2K+1$.**

- **Case** $\texttt{type} = \texttt{lin}$.

  - **FFN layers.**
    We use $l$-th ($2 \le l \le K$) FFN layer to express $l$-th memory $t_l$ exactly.
    By Lemma G.6, there exists a two-layer neural network with $m$ neurons defined on $\mathbb{R}^{ld}$

    $$f_{(l)}^{\text{2NN}}(x) = \sum_{k=1}^m a_k^{(l)} \sigma(b_k^{(l)\top} x + c_k^{(l)})$$

    such that

    $$\left\|g_l - f_{(l)}^{\text{2NN}}\right\|_{L^\infty([-2,2]^{ld})} \le \tilde{\mathcal{O}}\left(\frac{\|g_l\|_\mathcal{B}}{\sqrt{m}}\right).$$

    For $l$-th FFN layer, we only need to arrange the parameters $a_k^{(l)}$, $b_k^{(l)}$, and $c_k^{(l)}$ ($k \in [m], 2 \le l \le M$).

Denote $\bar{\boldsymbol{b}}_k^{(l)} = (\boldsymbol{b}_k^{(l)\top}, \boldsymbol{0}^\top)^\top \in \mathbb{R}^D$ for $k \in [m], 2 \le l \le K - 1$. Consider the following two-layer neural network with $m$ neurons defined on $\mathbb{R}^D$:

$$\mathbf{FFN}^{(l)}(\boldsymbol{x}) = \sum_{k=1}^m \boldsymbol{e}_{D-M+l-1} a_k^{(l)} \sigma \left( \bar{\boldsymbol{b}}_k^{(l)\top} \boldsymbol{x} + c_k^{(l)} \right).$$

It is easy to verify

$$\mathbf{FFN}^{(l)}(\boldsymbol{x}) = \left( \boldsymbol{0}^\top, f_{(l)}^{2\mathrm{NN}} \left( \boldsymbol{Q}^{(l)} \boldsymbol{x} \right), \boldsymbol{0}_{D-M+l-1}^\top \right)^\top \in \mathbb{R}^D, \quad \forall \boldsymbol{x} \in \mathbb{R}^D.$$

Notice that if the width $m$ satisfies

$$\tilde{\mathcal{O}} \left( \frac{\|g_l\|_{\mathcal{B}}}{\sqrt{m}} \right) < \frac{1}{4},$$

and the input $\boldsymbol{x}_t^{(l-1/2)}$ satisfies

$$\|g_l\|_{\mathrm{Lip}} \cdot \left\| (\boldsymbol{x}_t^\top, \cdots, \boldsymbol{x}_{t-t_{l-1}}^\top)^\top - \boldsymbol{Q}^{(l-1)} \boldsymbol{x}_t^{(l-1/2)} \right\|_2 \le \frac{1}{4}$$

the following holds:

$$\left| g_l(\boldsymbol{x}_t, \cdots, \boldsymbol{x}_{t-t_{l-1}}) - f_{(l)}^{2\mathrm{NN}} \left( \boldsymbol{Q}^{(l)} \boldsymbol{x}_t^{(l-1/2)} \right) \right|$$
$$\le \left| g_l(\boldsymbol{x}_t, \cdots, \boldsymbol{x}_{t-t_{l-1}}) - g_l(\boldsymbol{Q}^{(l-1)} \boldsymbol{x}_t^{(l-1/2)}) \right|$$
$$\quad + \left| g_l(\boldsymbol{Q}^{(l-1)} \boldsymbol{x}_t^{(l-1/2)}) - f_{(l)}^{2\mathrm{NN}}(\boldsymbol{Q}^{(l-1)} \boldsymbol{x}_t^{(l-1/2)}) \right|$$
$$\le \|g_l\|_{\mathrm{Lip}} \left\| (\boldsymbol{x}_t^\top, \cdots, \boldsymbol{x}_{t-t_{l-1}}^\top)^\top - \boldsymbol{Q}^{(l-1)} \boldsymbol{x}_t^{(l-1/2)} \right\|_2 + \left\| g_l - f_{(l)}^{2\mathrm{NN}} \right\|_{L^\infty}$$
$$< \frac{1}{4} + \frac{1}{4} = \frac{1}{2},$$

Noticing $t_l = g_l(\boldsymbol{x}_t, \cdots, \boldsymbol{x}_{t-t_{l-1}}) \in \mathbb{N}_+$, we have

$$\widetilde{f_{(l)}^{2\mathrm{NN}}}(\boldsymbol{Q}^{(l-1)} \boldsymbol{x}_t^{(l-1/2)}) = t_l.$$

Thus, it holds that:

$$\widetilde{\mathbf{FFN}}^{(l)}(\boldsymbol{x}_t^{(l-1/2)}) = (\boldsymbol{0}^\top, t_l, \boldsymbol{0}_{D-M+l-1}^\top)^\top.$$

– **Attn layers.**
By Lemma F.2, for any rate $n \in \mathbb{N}_+$, there exists a constant $C(n)$ and $K$ functions

$$\phi_l^{\exp}(t; B) = \sum_{h=1}^H \alpha_{l,h} \exp(-\beta_{l,h}(t - B))$$
$$= \sum_{h=1}^H \alpha_{l,h} \exp \left( \beta_{l,h} B - \beta_{l,h} t \right), \quad 1 \le l \le K$$

such that $\beta_{l,h} > 0$ and

$$\sup_{1 \le B \le T_l} \|\mathbb{I}\{\cdot = B\} - \phi_l^{\exp}(\cdot; B)\|_{\ell_1(\mathbb{N})} \le \frac{C(n) e^{0.01(n+1)T_l}}{H^n}, \ 1 \le l \le K.$$

Moreover, Noticing that $1 \le g_l(\cdot) \le T_l$ holds for any $\boldsymbol{X} = (\boldsymbol{x}_t)_{t \in \mathbb{Z}} \in \mathcal{X}$ and $2 \le l \le K$, the following holds:

$$\sup_{\boldsymbol{X}} \|\mathbb{I}\{\cdot = t_l\} - \phi_l^{\exp}(\cdot; t_l)\|_{\ell_1(\mathbb{N})}$$
$$\le \sup_{1 \le B \le T_l} \|\mathbb{I}\{\cdot = B\} - \phi_l^{\exp}(\cdot; B)\|_{\ell_1(\mathbb{N})} \le \frac{C(n) e^{0.01(n+1)T_l}}{H^n}.$$

Therefore, for the attention heads $h$ ($h \in [H]$) in each layer $l$ ($1 \le l \le K$), we can choose:

$$p^{(l+1,h)} = \beta_{l,h}, \quad \boldsymbol{W}_O^{(l+1)} = \boldsymbol{I}, \quad \boldsymbol{W}_V^{(l+1,h)} = \alpha_{l,h}\boldsymbol{\delta}_{(l+1,1)}^{(d \times d)} \in \mathbb{R}^{D \times D},$$

$$\boldsymbol{W}_Q^{(l+1,h)} = \sqrt{\beta_{l,h}}\boldsymbol{\delta}_{(D-M+l,1)}^{(1 \times 1)} \in \mathbb{R}^{D \times (D/H)},$$

$$\boldsymbol{W}_K^{(l+1,h)} = \sqrt{\beta_{l,h}}\boldsymbol{\delta}_{(D,1)}^{(1 \times 1)} \in \mathbb{R}^{D \times (D/H)},$$

where $\boldsymbol{\delta}_{(p_1, p_2)}^{(r \times r)}$ means that: it equals to $\boldsymbol{I}_{r \times r}$ for the $(p_1, p_2)$-th $r \times r$ blocks, and $\boldsymbol{0}_{r \times r}$ for the other $r \times r$ blocks.

- **Case** `type = log`**.**

    - **FFN layers.**
      We use $l$-th ($2 \le l \le K$) FFN layer to express $l$-th memory $t_l$ exactly.
      By Lemma G.6, there exists a two-layer neural network with $m$ neurons defined on $\mathbb{R}^{ld}$

      $$f_{(l)}^{2\mathrm{NN}}(\boldsymbol{x}) = \sum_{k=1}^{m} a_k^{(l)} \sigma(\boldsymbol{b}_k^{(l)\top}\boldsymbol{x} + c_k^{(l)})$$

      such that

      $$\left\| \log g_l - f_{(l)}^{2\mathrm{NN}} \right\|_{L^\infty} \le \tilde{\mathcal{O}}\left( \frac{\|\log g_l\|_{\mathcal{B}}}{\sqrt{m}} \right).$$

      For $l$-th FFN layer, we only need to arrange the parameters $a_k^{(l)}$, $\boldsymbol{b}_k^{(l)}$, and $c_k^{(l)}$ ($k \in [m], 2 \le l \le M$).
      Denote $\bar{\boldsymbol{b}}_k^{(l)} = (\boldsymbol{b}_k^{(l)\top}, \boldsymbol{0}^\top)^\top \in \mathbb{R}^D$ for $k \in [m], 2 \le l \le K - 1$. We consider the following $l$-th layer 2NN with $m$ neurons defined on $\mathbb{R}^D$:

      $$\mathbf{FFN}^{(l)}(\boldsymbol{x}) = \sum_{k=1}^{m} \boldsymbol{e}_{D-M+l-1} a_k^{(r)} \sigma\left( \bar{\boldsymbol{b}}_k^{(r)\top}\boldsymbol{x} + c_k^{(r)} \right).$$

      It is easy to verify

      $$\mathbf{FFN}^{(l)}(\boldsymbol{x}) = \left( \boldsymbol{0}^\top, f_{(l)}^{2\mathrm{NN}}\left( \boldsymbol{Q}^{(l)}\boldsymbol{x} \right), \boldsymbol{0}_{D-M+l-1}^\top \right)^\top \in \mathbb{R}^D, \quad \forall \boldsymbol{x} \in \mathbb{R}^D.$$

      Notice that if the width $m$ satisfies

      $$\tilde{\mathcal{O}}\left( \frac{\|\log g_l\|_{\mathcal{B}}}{\sqrt{m}} \right) < \frac{1}{8T_l},$$

      and the input $\boldsymbol{x}_t^{(l-1/2)}$ satisfies

      $$\|\log g_l\|_{\mathrm{Lip}} \cdot \left\| (\boldsymbol{x}_t^\top, \cdots, \boldsymbol{x}_{t-t_{l-1}}^\top)^\top - \boldsymbol{Q}^{(l)}\boldsymbol{x}_t^{(l-1/2)} \right\|_2 \le \frac{1}{8T_l},$$

      the following holds:

      $$\left| \log g_l(\boldsymbol{x}_t, \cdots, \boldsymbol{x}_{t-t_{l-1}}) - f_{(l)}^{2\mathrm{NN}}(\boldsymbol{Q}^{(l)}\boldsymbol{x}_t^{(l-1/2)}) \right|$$

      $$\le \left| \log g_l(\boldsymbol{x}_t, \cdots, \boldsymbol{x}_{t-t_{l-1}}) - \log g_l(\boldsymbol{Q}^{(l)}\boldsymbol{x}_t^{(l-1/2)}) \right|$$

      $$+ \left| g_l(\boldsymbol{Q}^{(l)}\boldsymbol{x}_t^{(l-1/2)}) - f_{(l)}^{2\mathrm{NN}}(\boldsymbol{Q}^{(l)}\boldsymbol{x}_t^{(l-1/2)}) \right|$$

$$\leq \|\log g_l\|_{\mathrm{Lip}} \left\| (\boldsymbol{x}_t^\top, \cdots, \boldsymbol{x}_{t-t_{l-1}}^\top)^\top - \boldsymbol{Q}^{(l)} \boldsymbol{x}_t^{(l-1/2)} \right\|_2 + \left\| \log g_l - f_{(l)}^{2\mathrm{NN}} \right\|_{L^\infty}$$

$$< \frac{1}{8T_l} + \frac{1}{8T_l} = \frac{1}{4T_l},$$

which ensures

$$\left| \exp\left( f_{(l)}^{2\mathrm{NN}}(\boldsymbol{Q}^{(l)} \boldsymbol{x}_t^{(l-1/2)}) \right) - g_l(\boldsymbol{x}_t, \cdots, \boldsymbol{x}_{t-t_{l-1}}) \right|$$

$$= \left| \exp\left( f_{(l)}^{2\mathrm{NN}}(\boldsymbol{Q}^{(l)} \boldsymbol{x}_t^{(l-1/2)}) \right) - \exp\left( \log\left( g_l(\boldsymbol{x}_t, \cdots, \boldsymbol{x}_{t-t_{l-1}}) \right) \right) \right|$$

$$\leq \exp\left( \max\left\{ f_{(l)}^{2\mathrm{NN}}(\boldsymbol{Q}^{(l)} \boldsymbol{x}_t^{(l-1/2)}), \log\left( g_l(\boldsymbol{x}_t, \cdots, \boldsymbol{x}_{t-t_{l-1}}) \right) \right\} \right)$$

$$\cdot \left| f_{(l)}^{2\mathrm{NN}}(\boldsymbol{Q}^{(l)} \boldsymbol{x}_t^{(l-1/2)}) - \log\left( g_l(\boldsymbol{x}_t, \cdots, \boldsymbol{x}_{t-t_{l-1}}) \right) \right|$$

$$\leq \exp\left( \log\left( g_l(\boldsymbol{x}_t, \cdots, \boldsymbol{x}_{t-t_{l-1}}) \right) + \frac{1}{8} \right) \frac{1}{4T_r}$$

$$\leq e^{1/8} \cdot T_r \cdot \frac{1}{4T_r} < \frac{1}{2}.$$

Noticing $t_l = g_l(\boldsymbol{x}_t, \cdots, \boldsymbol{x}_{t-t_{l-1}}) \in \mathbb{N}_+$, we have

$$\widetilde{f_{(l)}^{2\mathrm{NN}}}(\boldsymbol{Q}_l \boldsymbol{x}_t^{(l-1/2)}) = \log\left( \exp\left[ \mathrm{FFN}^{(l)}(\boldsymbol{x}_t, \cdots, \boldsymbol{x}_{t-t_{l-1}}) \right] \right) = \log t_l.$$

Thus, it holds that:

$$\widetilde{\mathbf{FFN}}^{(l)}(\boldsymbol{x}_t^{(l-1/2)}) = (\boldsymbol{0}^\top, \log t_l, \boldsymbol{0}_{D-M+l-1}^\top)^\top.$$

– **Attn layers.**
By Lemma F.5, for any rate $n \in \mathbb{N}_+$, there exists a constant $C(n)$ and $K$ functions

$$\phi_l^{\mathrm{poly}}(t; B) = \sum_{h=1} \alpha_{l,h}(t/B)^{-\beta_{l,h}}$$

$$= \sum_{h=1}^H \alpha_{l,h} \exp\left( -\beta_{l,h} \log(t/B) \right), \quad 1 \leq l \leq K$$

such that $\beta_{l,h} > 0$ and

$$\sup_{1 \leq B \leq T_l} \left\| \mathbb{I}\{\cdot = B\} - \phi_l^{\mathrm{poly}}(\cdot; B) \right\|_{\ell_1(\mathbb{N}_+)} \leq \frac{C(n)T_l^{1.01(n+1)}}{H^n}, \quad 1 \leq l \leq K.$$

Moreover, Noticing that $1 \leq g_l(\cdot) \leq T_l$ holds for any $\boldsymbol{X} = (\boldsymbol{x}_t)_{t\in\mathbb{Z}} \in \mathcal{X}$ and $1 \leq l \leq K$, the following holds:

$$\sup_{\boldsymbol{X}} \left\| \mathbb{I}\{\cdot = t_l\} - \phi_l^{\mathrm{poly}}(\cdot; t_l) \right\|_{\ell_1(\mathbb{N}_+)}$$

$$\leq \sup_{1 \leq B \leq T_l} \left\| \mathbb{I}\{\cdot = B\} - \phi_l^{\mathrm{poly}}(\cdot; B) \right\|_{\ell_1(\mathbb{N}_+)} \leq \frac{C(n)T_l^{1.01(n+1)}}{H^n}.$$

Therefore, for the attention heads $h$ ($h \in [H]$) in each layer $l$ ($1 \leq l \leq K$), we can choose:

$$p^{(l+1,h)} = \beta_{l,h}, \quad \boldsymbol{W}_O^{(l+1)} = \boldsymbol{I}, \quad \boldsymbol{W}_V^{(l+1,h)} = \alpha_{l,h}\boldsymbol{\delta}_{(l+1,1)}^{(d\times d)} \in \mathbb{R}^{D\times D},$$

$$\boldsymbol{W}_Q^{(l+1,h)} = \sqrt{\beta_{l,h}}\boldsymbol{\delta}_{(D-M+l,1)}^{(1\times 1)} \in \mathbb{R}^{D\times(D/H)},$$

$$\boldsymbol{W}_K^{(l+1,h)} = \sqrt{\beta_{l,h}}\boldsymbol{\delta}_{(D,1)}^{(1\times 1)} \in \mathbb{R}^{D\times(D/H)},$$

where $\boldsymbol{\delta}_{(p_1,p_2)}^{(r\times r)}$ means that: it equals to $\boldsymbol{I}_{r\times r}$ for the $(p_1, p_2)$-th $r \times r$ blocks, and $\boldsymbol{0}_{r\times r}$ for the other $r \times r$ blocks.

Similar to the estimate in Step II and Step III in the proof of Theorem C.2, it is easy to prove the following estimates by induction.

If the width satisfies

$$m \geq \begin{cases} \tilde{\Omega}\left(\max_{l\in[K]}\|g_l\|_{\mathcal{B}}^2\right) = \tilde{\Omega}\left(\|g_K\|_{\mathcal{B}}^2\right), & \texttt{type} = \texttt{lin} \\ \tilde{\Omega}\left(\max_{l\in[K]}\|\log g_l\|_{\mathcal{B}}^2 T_l^2\right) = \tilde{\Omega}\left(\|\log g_K\|_{\mathcal{B}}^2 T_K^2\right), & \texttt{type} = \texttt{log} \end{cases},$$

and the head number satisfies

$$\begin{cases} \frac{C(n)}{H^n}\sqrt{\sum_{l=1}^{K-1} e^{0.02(n+1)T_l}} \leq \frac{1}{4\max_{l\in[K-1]}\|g_l\|_{\mathrm{Lip}}}, & \texttt{type} = \texttt{lin} \\ \frac{C(n)}{H^n}\sqrt{\sum_{l=1}^{K-1} T_l^{2.02(n+1)}} \leq \frac{1}{4\max_{l\in[K-1]}\|\log g_l\|_{\mathrm{Lip}}}, & \texttt{type} = \texttt{log} \end{cases},$$

then the following estimates hold:

- **(E1)** for any $2 \leq l \leq K$,

$$\boldsymbol{x}_t^{(l)} = \boldsymbol{x}_t^{(l-1/2)} + (\boldsymbol{0}^\top, t_l, \boldsymbol{0}_{M-l+1}^\top)^\top;$$

- **(E2)** for any $1 \leq l \leq K$,
$$\boldsymbol{P}_\perp^{(l)} \boldsymbol{x}_t^{(l+1/2)} = \boldsymbol{P}_\perp^{(l)} \boldsymbol{x}_t^{(l)};$$

- **(E3)** for any $1 \leq l \leq K$,

$$\left\|\boldsymbol{P}^{(l)}\left(\boldsymbol{x}_t^{(l+1/2)} - \left(\boldsymbol{x}_t^{(l)} + (\boldsymbol{0}_{ld}^\top, \boldsymbol{x}_{t-t_l}^\top, \boldsymbol{0}^\top)^\top\right)\right)\right\|_2 \leq \begin{cases} \frac{C(n)e^{0.01(n+1)T_l}}{H^n}, & \texttt{type} = \texttt{lin} \\ \frac{C(n)T_l^{1.01(n+1)}}{H^n}, & \texttt{type} = \texttt{log} \end{cases};$$

$$\left\|\boldsymbol{Q}^{(l)}\left(\boldsymbol{x}_t^{(l+1/2)} - \boldsymbol{x}_t^{(0)}\right)\right\|_2 \leq \begin{cases} \frac{C(n)}{H^n}\sqrt{\sum_{j=1}^l e^{0.02(n+1)T_l}}, & \texttt{type} = \texttt{lin} \\ \frac{C(n)}{H^n}\sqrt{\sum_{j=1}^l T_j^{2.02(n+1)}}, & \texttt{type} = \texttt{log} \end{cases}.$$

**The Remained Steps.**

- **Regime $M \geq K + 1$.**

  Step $2K + 2$ and $2K + 3$.

  In the similar way as **Step 3** $\sim$ **Step $2K - 1$** in this proof and **Step II, Step III** in the proof of Theorem C.2, it is easy to verify the following estimate.

  If the width satisifes

$$m \geq \begin{cases} \tilde{\Omega}\left(\sum_{l=K+1}^M \|g_l\|_{\mathcal{B}}^2\right), & \texttt{type} = \texttt{lin} \\ \tilde{\Omega}\left(\sum_{l=K+1}^M \|\log g_l\|_{\mathcal{B}}^2 T_l^2\right), & \texttt{type} = \texttt{log} \end{cases},$$

  and the head number satisfies

$$\begin{cases} \frac{C(n)}{H^n}\sqrt{\sum_{l=1}^K e^{0.02(n+1)T_l}} \leq \frac{1}{4\max_{l\in[K]}\|g_l\|_{\mathrm{Lip}}}, & \texttt{type} = \texttt{lin} \\ \frac{C(n)}{H^n}\sqrt{\sum_{l=1}^K T_l^{2.02(n+1)}} \leq \frac{1}{4\max_{l\in[K]}\|\log g_l\|_{\mathrm{Lip}}}, & \texttt{type} = \texttt{log} \end{cases},$$

  then the following estimates hold:

$$\boldsymbol{x}_t^{(K+1)} = \boldsymbol{x}_t^{(K+1/2)} + (\boldsymbol{0}^\top, t_{K+1}, \cdots, t_M, 0)^\top;$$

$$\boldsymbol{R}_\perp \boldsymbol{x}_t^{(K+3/2)} = \boldsymbol{R}_\perp \boldsymbol{x}_t^{(K+1)};$$

$$\left\| \boldsymbol{R} \left( \boldsymbol{x}_t^{(K+3/2)} - \left( \boldsymbol{x}_t^{(K+1)} + (\boldsymbol{0}_{(K+1)D}^\top, \boldsymbol{x}_{t-t_{K+1}}^\top, \cdots, \boldsymbol{x}_{t-t_M}^\top, \boldsymbol{0}^\top)^\top \right) \right) \right\|_2$$
$$\leq \begin{cases} C(n) \left( \dfrac{\sum_{l=K+1}^M e^{0.01 T_l}}{H^{\frac{n}{n+1}}} \right)^{n+1}, & \texttt{type} = \text{lin} \\[4mm] C(n) \left( \dfrac{\sum_{l=K+1}^M T_l^{1.01}}{H^{\frac{n}{n+1}}} \right)^{n+1}, & \texttt{type} = \log \end{cases},$$

$$\left\| \boldsymbol{Q}^{(M)} \left( \boldsymbol{x}_t^{(K+3/2)} - \boldsymbol{x}_t^{(0)} \right) \right\|_2$$
$$\leq \begin{cases} \dfrac{C(n)}{H^n} \sqrt{\sum_{l=1}^K e^{0.02(n+1)T_l} + \left( \sum_{l=K+1}^M e^{0.01 T_l} \right)^{2n+2}}, & \texttt{type} = \text{lin} \\[4mm] \dfrac{C(n)}{H^n} \sqrt{\sum_{l=1}^K T_l^{2.02(n+1)} + \left( \sum_{l=K+1}^M T_l^{1.01} \right)^{2n+2}}, & \texttt{type} = \log \end{cases}.$$

#### Step $2K + 4$ and the final bound.

In the same way as Step IV in the proof of Theorem C.2, there exists **FFN**, such that the following estimate holds for any $t$ and $\boldsymbol{X}$:

$$\left\| \boldsymbol{H}_t(\boldsymbol{X}) - \boldsymbol{x}_t^{(K+2)} \right\| \leq \|f\|_{\text{Lip}} \cdot \left\| \boldsymbol{Q}^{(M)} \left( \boldsymbol{x}_t^{(K+3/2)} - \boldsymbol{x}_t^{(0)} \right) \right\|_2 + \mathcal{E}_{\text{FFN}}$$
$$= \|f\|_{\text{Lip}} \cdot \mathcal{E}_{\text{Attn}}(\texttt{type}) + \mathcal{E}_{\text{FFN}},$$

where

$$\mathcal{E}_{\text{FFN}} = \mathcal{O}\left( \frac{\|f\|_{\mathcal{B}}}{\sqrt{m}} \right),$$

$$\mathcal{E}_{\text{Attn}}(\texttt{type}) = \left\| \boldsymbol{Q}^{(M)} \left( \boldsymbol{x}_t^{(K+3/2)} - \boldsymbol{x}_t^{(0)} \right) \right\|_2$$
$$= \begin{cases} \dfrac{C(n)}{H^n} \sqrt{\sum_{l=1}^K e^{0.02(n+1)T_l} + \left( \sum_{l=K+1}^M e^{0.01 T_l} \right)^{2n+2}}, & \texttt{type} = \text{lin} \\[4mm] \dfrac{C(n)}{H^n} \sqrt{\sum_{l=1}^K T_l^{2.02(n+1)} + \left( \sum_{l=K+1}^M T_l^{1.01} \right)^{2n+2}}, & \texttt{type} = \log \end{cases}.$$

Recalling our analysis, we need the head number satisfies

$$\begin{cases} \dfrac{C(n)}{H^n} \sqrt{\sum_{l=1}^K e^{0.02(n+1)T_l}} \leq \dfrac{1}{4 \max\limits_{l \in [K]} \|g_l\|_{\text{Lip}}}, & \texttt{type} = \text{lin} \\[4mm] \dfrac{C(n)}{H^n} \sqrt{\sum_{l=1}^K T_l^{2.02(n+1)}} \leq \dfrac{1}{4 \max\limits_{l \in [K]} \|\log g_l\|_{\text{Lip}}}, & \texttt{type} = \log \end{cases}.$$

Due to

$$\begin{cases} \dfrac{C(n)}{H^n} \sqrt{\sum_{l=1}^K e^{0.02(n+1)T_l}} \leq \mathcal{E}_{\text{Attn}}(\texttt{type}), & \texttt{type} = \text{lin} \\[4mm] \dfrac{C(n)}{H^n} \sqrt{\sum_{l=1}^K T_l^{2.02(n+1)}} \leq \mathcal{E}_{\text{Attn}}(\texttt{type}), & \texttt{type} = \log \end{cases},$$

when we is large enough, this condition holds naturally and do not affect the approximation rate.

Moreover, we need the following condition on the width:

$$m \geq \begin{cases} \tilde{\Omega} \left( \max_{l \in [K]} \vee \sum_{l=K+1}^{M} \|g_l\|_{\mathcal{B}}^2 \right), & \texttt{type} = \texttt{lin} \\ \tilde{\Omega} \left( \max_{l \in [K]} \vee \sum_{l=K+1}^{M} \|\log g_l\|_{\mathcal{B}}^2 T_l^2 \right), & \texttt{type} = \texttt{log} \end{cases},$$

- **Regime $M = K$.**

  Step $2K + 2$ **and the final bound.**

  In the same way as Step IV in the proof of Theorem C.2, there exists **FFN**, such that the following estimate holds for any $t$ and $\boldsymbol{X}$:

$$\left\| \mathbf{H}_t(\boldsymbol{X}) - \boldsymbol{x}_t^{(K+1)} \right\| \leq \|f\|_{\mathrm{Lip}} \cdot \left\| \boldsymbol{Q}^{(M)} \left( \boldsymbol{x}_t^{(K+1/2)} - \boldsymbol{x}_t^{(0)} \right) \right\|_2 + \mathcal{E}_{\mathrm{FFN}}$$
$$= \|f\|_{\mathrm{Lip}} \cdot \mathcal{E}_{\mathrm{Attn}}(\texttt{type}) + \mathcal{E}_{\mathrm{FFN}},$$

  where

$$\mathcal{E}_{\mathrm{FFN}} = \mathcal{O} \left( \frac{\|f\|_{\mathcal{B}}}{\sqrt{m}} \right),$$

$$\mathcal{E}_{\mathrm{Attn}}(\texttt{type}) = \left\| \boldsymbol{Q}_M \left( \boldsymbol{x}_t^{(K+1/2)} - \boldsymbol{x}_t^{(0)} \right) \right\|_2$$
$$= \begin{cases} \frac{C(n)}{H^n} \sqrt{\sum_{l=1}^{K} e^{0.02(n+1)T_l}}, & \texttt{type} = \texttt{lin} \\ \frac{C(n)}{H^n} \sqrt{\sum_{l=1}^{K} T_l^{2.02(n+1)}}, & \texttt{type} = \texttt{log} \end{cases}.$$

  Recalling our analysis, we need the head number satisfies

$$\begin{cases} \frac{C(n)}{H^n} \sqrt{\sum_{l=1}^{K-1} e^{0.02(n+1)T_l}} \leq \frac{1}{4 \max_{l \in [K-1]} \|g_l\|_{\mathrm{Lip}}}, & \texttt{type} = \texttt{lin} \\ \frac{C(n)}{H^n} \sqrt{\sum_{l=1}^{K-1} T_l^{2.02(n+1)}} \leq \frac{1}{4 \max_{l \in [K-1]} \|\log g_l\|_{\mathrm{Lip}}}, & \texttt{type} = \texttt{log} \end{cases}.$$

  Due to

$$\begin{cases} \frac{C(n)}{H^n} \sqrt{\sum_{l=1}^{K-1} e^{0.02(n+1)T_l}} \leq \mathcal{E}_{\mathrm{Attn}}(\texttt{type}), & \texttt{type} = \texttt{lin} \\ \frac{C(n)}{H^n} \sqrt{\sum_{l=1}^{K-1} T_l^{2.02(n+1)}} \leq \mathcal{E}_{\mathrm{Attn}}(\texttt{type}), & \texttt{type} = \texttt{log} \end{cases},$$

  when $H$ is large enough, this condition holds naturally and do not affect the approximation rate.

  Moreover, we need the following condition on the width:

$$m \geq \begin{cases} \tilde{\Omega} \left( \max_{l \in [K]} \|g_l\|_{\mathcal{B}}^2 \right), & \texttt{type} = \texttt{lin} \\ \tilde{\Omega} \left( \max_{l \in [K]} \|\log g_l\|_{\mathcal{B}}^2 T_l^2 \right), & \texttt{type} = \texttt{log} \end{cases},$$

Combining these two regimes, we complete our proof.

$$\square$$

## D.2 Proof of Proposition 4.5

*Proof of Proposition 4.5.*
This proposition is a direct corollary of Theorem 4.4. It can be seen as a special case of $M = K$ in Theorem 4.4.

Therefore, under the same conditions, there exists a $K + 1$-layer Transformer $\mathbf{TF} \in \mathcal{TF}_{(K+1,H,m)}^{\text{NF,type}}$ (12) and a constant $C(n)$ such that: if the width satisfies

$$m \geq \begin{cases} \tilde{\Omega}\Big( \max_{i \in [K]} \|g_i\|_{\mathcal{B}}^2 \Big), & \texttt{type} = \text{lin}, \\ \tilde{\Omega}\Big( \max_{i \in [K]} \|\log g_i\|_{\mathcal{B}}^2 T_i^2 \Big), & \texttt{type} = \log \end{cases},$$

then the following approximation rate holds:

$$\|\mathbf{H} - \mathbf{TF}\| \leq \mathcal{E}_{\text{FFN}} + \|f\|_{\text{Lip}} \mathcal{E}_{\text{Attn}}(\texttt{type}),$$

where $\mathcal{E}_{\text{FFN}} = \tilde{\mathcal{O}}\left( \frac{\|f\|_{\mathcal{B}}}{\sqrt{m}} \right)$ and

$$\mathcal{E}_{\text{Attn}}(\texttt{type}) = \begin{cases} \mathcal{O}\left( \frac{C(n)}{H^n} \sqrt{\sum_{l=1}^K e^{0.02(n+1)T_l}} \right), & \texttt{type} = \text{lin} \\ \mathcal{O}\left( \frac{C(n)}{H^n} \sqrt{\sum_{l=1}^K T_l^{2.02(n+1)}} \right), & \texttt{type} = \log \end{cases}.$$

$\square$

**Comparison between Proposition 4.5 and Theorem 4.1.**

We compare 2-layer Transformer and $M + 1$-layer Transformer regarding the requirement of the number of heads and width.

- *The required width of FFN layers.*
  - For 2-layer Transformer, the required width of FFN layers $m_{\text{need}}^{(2)}$ is proportionally linked to the **sum** of all the memory functions' complexity:

  $$m_{\text{need}}^{(2)} = \begin{cases} \tilde{\Omega}\Big( \sum_{i \in [K]} \|g_i\|_{\mathcal{B}}^2 \Big), & \texttt{type} = \text{lin}, \\ \tilde{\Omega}\Big( \sum_{i \in [K]} \|\log g_i\|_{\mathcal{B}}^2 T_i^2 \Big), & \texttt{type} = \log \end{cases}.$$

  - For $M + 1$-layer Transformer, the required width of FFN layers $m_{\text{need}}^{(M+1)}$ correlates with the **maximum** complexity of the memory functions:

  $$m_{\text{need}}^{(M+1)} = \begin{cases} \tilde{\Omega}\Big( \max_{i \in [K]} \|g_i\|_{\mathcal{B}}^2 \Big), & \texttt{type} = \text{lin}, \\ \tilde{\Omega}\Big( \max_{i \in [K]} \|\log g_i\|_{\mathcal{B}}^2 T_i^2 \Big), & \texttt{type} = \log \end{cases}.$$

  It is easy to see that:

  $$\frac{m_{\text{need}}^{(M+1)}}{m_{\text{need}}^{(2)}} = \frac{\max\{a_1, \cdots, a_M\}}{\sum_{k=1}^M a_k},$$

  $$\max\{a_1, \cdots, a_M\} \leq \sum_{k=1}^M a_k$$

- *The required number of Attn heads.* To achieve the same $\mathcal{E}_{\text{Attn}}(\texttt{type}) = \epsilon$,

- for 2-layer Transformer, the required number of Attn heads $H_{\text{need}}^{(2)}$ satisfies:

$$\epsilon = \begin{cases} \mathcal{O}\left( \frac{C(n)}{\left(H_{\text{need}}^{(2)}\right)^n} \sqrt{\sum_{l=1}^{K} e^{0.02(n+1)T_l}} \right), & \texttt{type} = \text{lin} \\ \mathcal{O}\left( \frac{C(n)}{\left(H_{\text{need}}^{(2)}\right)^n} \sqrt{\sum_{l=1}^{K} T_l^{2.02(n+1)}} \right), & \texttt{type} = \log \end{cases} .$$

- for $M+1$-layer Transformer, the required number of Attn heads $H_{\text{need}}^{(M+1)}$ satisfies:

$$\epsilon = \begin{cases} \mathcal{O}\left( \frac{C(n)}{\left(H_{\text{need}}^{(M+1)}\right)^n} \left(\sum_{i=1}^{M} e^{0.01T_i}\right)^{n+1} \right), & \texttt{type} = \text{lin} \\ \mathcal{O}\left( \frac{C(n)}{\left(H_{\text{need}}^{(M+1)}\right)^n} \left(\sum_{i=1}^{M} T_i^{1.01}\right)^{n+1} \right), & \texttt{type} = \log \end{cases} .$$

It is easy to see that:

$$\left( \frac{H_{\text{need}}^{(M+1)}}{H_{\text{need}}^{(2)}} \right)^{2n} = \frac{b_1^2 + \cdots + b_M^2}{(b_1 + \cdots b_M)^2},$$

$$b_1^2 + \cdots + b_M^2 \leq (b_1 + \cdots b_M)^2.$$

This finding suggests that increased depth can significantly reduce the demands on the number of heads and the width. The underlying reason is that deep networks can distribute memories across different layers for processing, with each layer focusing on approximating only a single memory function.

# E  Proof of Section 5

## E.1  Proof of Theorem 5.1

In this subsection, we give the detailed proofs of the warm-up case of (fixed) essentially sparse memories as follows:

$$y_t = f\left(\left(\boldsymbol{X} * \rho_1\right)(t), \cdots, \left(\boldsymbol{X} * \rho_M\right)(t)\right),$$

where $\rho_1(\cdot), \cdots, \rho_M(\cdot) \in \ell^1(\mathbb{N})$ serve as memory kernels, and $(\boldsymbol{X} * \rho_k)(t) = \sum_{s=0}^{+\infty} \boldsymbol{x}_{t-s} \rho_k(s)$ denotes the convolution of the inputs with kernel $\rho_k$.

**Theorem E.1** (Restatement of Theorem 5.1)**.**
*(A) Consider $\mathcal{H}^{\mathrm{Ess}}$ (14) with exponentially decayed memory kernels, i.e., there exists $\beta > 0$ such that $\rho_1(t), \cdots, \rho_M(t) = \mathcal{O}(e^{-\beta t})$. Then for any target $\mathbf{H} \in \mathcal{H}^{\mathrm{Ess}}$, rate $n \in [\lfloor 99\beta \rfloor]$, and $H, m \in \mathbb{N}_+$, there exists a 1-layer DP-free Transformer $\mathbf{TF} \in \mathcal{TF}_{(1,H,m)}^{\mathrm{DPF,exp}}$ (7) and a constant $C(n)$ such that*

$$\|\mathbf{H} - \mathbf{TF}\| \leq \mathcal{E}_{\mathrm{FFN}} + \|f\|_{\mathrm{Lip}} \mathcal{E}_{\mathrm{Attn}}(\mathtt{type});$$

*(B) Consider $\mathcal{H}^{\mathrm{Ess}}$ (14) with polynomially decayed memory kernels, i.e., there exists $\beta > 1$ such that $\rho_1(t), \cdots, \rho_M(t) = \mathcal{O}(t^{-\beta})$. Then for any target $\mathbf{H} \in \mathcal{H}^{\mathrm{Ess}}$, rate $n \in [\lfloor 0.99\beta \rfloor - 1]$, and $H, m \in \mathbb{N}_+$, there exists a 1-layer DP-free Transformer $\mathbf{TF} \in \mathcal{TF}_{(1,H,m)}^{\mathrm{DPF,poly}}$ (7) and a constant $C(n)$ such that*

$$\|\mathbf{H} - \mathbf{TF}\| \leq \mathcal{E}_{\mathrm{FFN}} + \|f\|_{\mathrm{Lip}} \mathcal{E}_{\mathrm{Attn}}(\mathtt{type});$$

*where $\mathcal{E}_{\mathrm{FFN}} = \tilde{\mathcal{O}}\left(\frac{\|f\|_{\mathcal{B}}}{\sqrt{m}}\right)$ and*

$$\mathcal{E}_{\mathrm{Attn}}(\mathtt{type}) = \mathcal{O}\left(\frac{C(n)M^{n+1}}{H^n}\right).$$

*Proof of Theorem E.1.*
The proof of this theorem is highly similar to the proof of Theorem B.1. The only difference is that the Attn layer needs to be used to approximate general memory kernel $\rho_k(\cdot)$ instead of simple $\mathbb{I}\{\cdot = T_k\}$. But for the completeness of the proof in this section, we still provide the detailed proof.

First, we choose the embedding dimension $D = Md$, and select the simple embedding $\boldsymbol{W}_E = (\boldsymbol{I}_{d \times d}, \boldsymbol{0})^\top \in \mathbb{R}^{D \times d}, \boldsymbol{b}_E = \boldsymbol{0} \in \mathbb{R}^D$.

For any input sequence $\boldsymbol{X} = (\boldsymbol{x}_t)_{t \in \mathbb{Z}}$, the token after embedding satisfies:

$$\boldsymbol{x}_t^E = \boldsymbol{W}_E \boldsymbol{x}_t + \boldsymbol{b}_E = (\boldsymbol{x}_t^\top, \boldsymbol{0}^\top)^\top \in \mathbb{R}^D.$$

Then for one-layer Dot-product-free Transformer $\mathbf{TF} \in \mathcal{TF}_{(1,H,m)}^{\mathrm{DPF,type}}$ without residual blocks, the output token $\mathbf{TF}_t(\boldsymbol{X})$ of $t$-th input token $\boldsymbol{x}_t$ satisfies:

$$\boldsymbol{x}_t^{(1/2)} = \boldsymbol{W}_O^{(1)} \sum_{h=1}^H \mathbf{Attn}_t^{(1,h)}(\boldsymbol{X}^{(0)}),$$

$$\boldsymbol{x}_t^{(1)} = \mathbf{FFN}^{(1)}(\boldsymbol{x}_t^{(1/2)})$$

where

$$\mathbf{Attn}_t^{(1,h)}(\boldsymbol{X}) = \boldsymbol{W}_V^{(1,h)} \sum_{s=0}^{+\infty} \frac{\boldsymbol{x}_{t-s} \exp\left(p^{(1,h)} \phi_{\mathtt{type}}(s)\right)}{\sum_{j=0}^{+\infty} \exp\left(p^{(1,h)} \phi_{\mathtt{type}}(j)\right)}.$$

This proof can be summarized as the following process:

$$\cdots \quad \boldsymbol{x}_t^E \quad \cdots$$

Step I. Attn layer $\downarrow$

$$\cdots \quad \boldsymbol{x}_t^{(1/2)} \approx ((\boldsymbol{X} * \rho_1)(t), \cdots, (\boldsymbol{X} * \rho_M)(t))^\top \quad \cdots$$

Step II. FFN layer $\downarrow$

$$\cdots \quad \boldsymbol{x}_t^{(1)} \approx \boldsymbol{f} \left( (\boldsymbol{X} * \rho_1)(t), \cdots, (\boldsymbol{X} * \rho_M)(t) \right) \quad \cdots$$

Now we give the formal proof.

**Step I.** Extract the memory locations by (Dot-product-free) Attn layer.

We consider to use $H_k$ attention heads (from $\sum_{i=1}^{k-1} H_i + 1$-th head to $\sum_{i=1}^{k} H_i$-th head) to extract it, and it satisfies to $\sum_{k=1}^{M} H_k = H$.

$$\boldsymbol{P}^{(k)} := \begin{pmatrix} \boldsymbol{0}_{d \times (k-1)d} & \boldsymbol{I}_{d \times d} & \boldsymbol{0} \end{pmatrix} \in \mathbb{R}^{d \times D}, \quad 1 \leq k \leq M.$$

$$\boldsymbol{P}_\perp^{(k)} := \begin{pmatrix} \boldsymbol{I}_{(k-1)d \times (k-1)d} & \boldsymbol{0}_{d \times d} & \boldsymbol{0} \\ \boldsymbol{0} & \boldsymbol{0}_{d \times d} & \boldsymbol{I}_{(M-k-1)d \times (M-k-1)d} \end{pmatrix} \in \mathbb{R}^{(M-1)d \times D}, \quad 1 \leq k \leq M.$$

Now we consider the extraction of $k$-th memory $(\boldsymbol{X} * \rho_k)(t)$ $(1 \leq k \leq M)$.

- **Case (A). Approximating exponentially decayed memories by** $\texttt{type} = \text{lin}$**.**

  Because there exists $\beta > 0$ such that $\rho_k(t) = \mathcal{O}(e^{-\beta t})$, by Lemma F.3, for any $n \in [\lfloor 99\beta \rfloor]$ and $m \in \mathbb{N}_+$, there exists an absolute constant $C(n)$ only depending on $n$ and a function

  $$\phi_k^{\exp}(t) = \sum_{\sum_{i=1}^{k-1} H_i + 1 \leq h \leq \sum_{i=1}^{k} H_i} \alpha_h e^{-\beta_h t}$$

  such that $\beta_h > 0$ and

  $$\|\rho_k(\cdot) - \phi_k^{\exp}(\cdot)\|_{\ell_1(\mathbb{N})} = \sum_{s=0}^{+\infty} |\rho_k(s) - \phi_k^{\exp}(s)| \leq \frac{C(n)}{m^n}.$$

  Therefore, for these attention heads $(\sum_{i=1}^{k-1} H_i + 1 \leq h \leq \sum_{i=1}^{k} H_i)$, we can choose

  $$p^{(1,h)} = \beta_h, \quad \boldsymbol{W}_V^{(1,h)} = \alpha_h \left( \sum_{j=0}^{+\infty} \exp(-\beta_h j) \right) \boldsymbol{\delta}_{(k,1)}^{d \times d},$$

  where $\boldsymbol{\delta}^{(k,1)} \in \mathbb{R}^{D \times D}$ means that: it equals to $\boldsymbol{I}_{d \times d}$ for the $(k,1)$-th $d \times d$ blocks, and $\boldsymbol{0}_{d \times d}$ for the other $d \times d$ blocks.

  Then it holds that:

  $$\sum_{h=\sum_{i=1}^{k-1} H_i+1}^{\sum_{i=1}^{k} H_i} \mathbf{Attn}_t^{(1,h)}(\boldsymbol{X}^{(0)}) = \sum_{h=\sum_{i=1}^{k-1} H_i+1}^{\sum_{i=1}^{k} H_i} \alpha_h \sum_{s=0}^{+\infty} e^{-\beta_h s} \begin{pmatrix} \boldsymbol{0}_{(k-1)d} \\ \boldsymbol{x}_{t-s} \\ \boldsymbol{0} \end{pmatrix} \in \mathbb{R}^D,$$

  This implies:

  $$\boldsymbol{P}^{(k)} \sum_{h=\sum_{i=1}^{k-1} H_i+1}^{\sum_{i=1}^{k} H_i} \mathbf{Attn}_t^{(1,h)}(\boldsymbol{X}^{(0)}) = \sum_{h=\sum_{i=1}^{k-1} H_i+1}^{\sum_{i=1}^{k} H_i} \alpha_h \sum_{s=0}^{+\infty} e^{-\beta_h s} \boldsymbol{x}_{t-s},$$

$$\boldsymbol{P}_{\perp}^{(k)} \sum_{h=\sum_{i=1}^{k-1} H_i+1}^{\sum_{i=1}^{k} H_i} \mathbf{Attn}_t^{(1,h)}(\boldsymbol{X}^{(0)}) = \mathbf{0},$$

moreover, the following estimate holds:

$$\left\| \sum_{h=\sum_{i=1}^{k-1} H_i+1}^{\sum_{i=1}^{k} H_i} \boldsymbol{P}^{(k)} \mathbf{Attn}_t^{(1,h)}(\boldsymbol{X}^{(0)}) - (\boldsymbol{X} * \rho_k)(t) \right\|_2$$

$$= \left\| \sum_{h=\sum_{i=1}^{k-1} H_i+1}^{\sum_{i=1}^{k} H_i} \alpha_h \sum_{s=0}^{+\infty} e^{-\beta_h s} \boldsymbol{x}_{t-s} - \sum_{s=0}^{+\infty} \boldsymbol{x}_{t-s} \rho_k(s) \right\|_2$$

$$= \left\| \sum_{s=0}^{+\infty} \left( \sum_{h=\sum_{i=1}^{k-1} H_i+1}^{\sum_{i=1}^{k} H_i} \alpha_h e^{-\beta_h s} - \mathbb{I}\{s = T_k\} \right) \boldsymbol{x}_{t-s} \right\|_2$$

$$\leq \sum_{s=0}^{+\infty} \left| \sum_{h=\sum_{i=1}^{k-1} H_i+1}^{\sum_{i=1}^{k} H_i} \alpha_h e^{-\beta_h s} - \rho_k(s) \right|$$

$$= \left\| \phi_k^{\exp}(\cdot) - \rho_k(\cdot) \right\|_{\ell_1(\mathbb{N})} \leq \frac{C(n)}{H_k^n}.$$

- **Case (B). Approximating polynomially decayed memories by** $\texttt{type} = \log$**.**

  Because there exists $\beta > 0$ such that $\rho_k(t) = \mathcal{O}(t^{-\beta})$, by Lemma F.6, for any $n \in [\lfloor 0.99\beta \rfloor - 1]$ and $m \in \mathbb{N}_+$, there exists an absolute constant $C(n)$ only depending on $n$ and a function

  $$\phi_k^{\text{poly}}(t) = \sum_{\sum_{i=1}^{k-1} H_i+1 \leq h \leq \sum_{i=1}^{k} H_i} \alpha_h t^{-\beta_h}$$

  such that $\beta_h > 1$ and

  $$\left\| \rho_k(\cdot) - \phi_k^{\text{poly}}(\cdot) \right\|_{\ell_1(\mathbb{N}_+)} = \sum_{s=1}^{+\infty} \left| \rho_k(s) - \phi_k^{\text{poly}}(s) \right| \leq \frac{C(n)}{m^n}.$$

  Therefore, for these attention heads ($\sum_{i=1}^{k-1} H_i + 1 \leq h \leq \sum_{i=1}^{k} H_i$), we can choose

  $$p^{(1,h)} = \beta_h, \quad \boldsymbol{W}_V^{(1,h)} = \alpha_h \left( \sum_{j=1}^{+\infty} j^{-\beta_h} \right) \boldsymbol{\delta}^{(k,1)},$$

  where $\boldsymbol{\delta}^{(k,1)} \in \mathbb{R}^{D \times D}$ means that: it equals to $\boldsymbol{I}_{d \times d}$ for the $(k,1)$-th $d \times d$ blocks, and $\boldsymbol{0}_{d \times d}$ for the other $d \times d$ blocks.

  Then it holds that:

  $$\sum_{h=\sum_{i=1}^{k-1} H_i+1}^{\sum_{i=1}^{k} H_i} \mathbf{Attn}_t^{(1,h)}(\boldsymbol{X}^{(0)}) = \sum_{h=\sum_{i=1}^{k-1} H_i+1}^{\sum_{i=1}^{k} H_i} \alpha_h \sum_{s=1}^{+\infty} s^{-\beta_h} \begin{pmatrix} \boldsymbol{0}_{(k-1)d} \\ \boldsymbol{x}_{t-s} \\ \boldsymbol{0} \end{pmatrix},$$

  This implies:

  $$\boldsymbol{P}^{(k)} \sum_{h=\sum_{i=1}^{k-1} H_i+1}^{\sum_{i=1}^{k} H_i} \mathbf{Attn}_t^{(1,h)}(\boldsymbol{X}^{(0)}) = \sum_{h=\sum_{i=1}^{k-1} H_i+1}^{\sum_{i=1}^{k} H_i} \alpha_h \sum_{s=1}^{+\infty} s^{-\beta_h} \boldsymbol{x}_{t-s},$$

$$\boldsymbol{P}_{\perp}^{(k)} \sum_{h=\sum_{i=1}^{k-1} H_i+1}^{\sum_{i=1}^{k} H_i} \mathbf{Attn}_t^{(1,h)}(\boldsymbol{X}^{(0)}) = \mathbf{0},$$

$$
\begin{aligned}
&\left\| \sum_{h=\sum_{i=1}^{k-1} H_i+1}^{\sum_{i=1}^{k} H_i} \boldsymbol{P}^{(k)} \mathbf{Attn}_t^{(1,h)}(\boldsymbol{X}^{(0)}) - (\boldsymbol{X} * \rho_k)(t) \right\|_2 \\
&= \left\| \sum_{h=\sum_{i=1}^{k-1} H_i+1}^{\sum_{i=1}^{k} H_i} \alpha_h \sum_{s=1}^{+\infty} s^{-\beta_h} \boldsymbol{x}_{t-s} - \sum_{s=0}^{+\infty} \boldsymbol{x}_{t-s} \rho_k(s) \right\|_2 \\
&= \left\| \sum_{s=1}^{+\infty} \left( \sum_{h=\sum_{i=1}^{k-1} H_i+1}^{\sum_{i=1}^{k} H_i} \alpha_h s^{-\beta_h} - \rho_k(s) \right) \boldsymbol{x}_{t-s} \right\|_2 \\
&\leq \sum_{s=1}^{+\infty} \left| \sum_{h=\sum_{i=1}^{k-1} H_i+1}^{\sum_{i=1}^{k} H_i} \alpha_h s^{-\beta_h} - \rho_k(s) \right| \\
&= \left\| \phi_{H_k}^{\mathrm{poly}}(\cdot) - \rho_k(\cdot) \right\|_{\ell_1(\mathbb{N}_+)} \leq \frac{C(n)}{H_k^n}.
\end{aligned}
$$

Then we combine the estimate for all $k \in [M]$ for these two cases. By choose $\boldsymbol{W}_O = \boldsymbol{I}_D$, we have:

$$
\begin{aligned}
&\left\| \boldsymbol{x}_t^{(1/2)} - \begin{pmatrix} (\boldsymbol{X} * \rho_1)(t) \\ \vdots \\ (\boldsymbol{X} * \rho_M)(t) \end{pmatrix} \right\|_2 \\
&= \left\| \sum_{k=1}^{M} \left( \sum_{h=\sum_{i=1}^{k-1} H_i+1}^{\sum_{i=1}^{k} H_i} \mathbf{Attn}_t^{(1,h)}(\boldsymbol{X}) - \begin{pmatrix} \mathbf{0}_{(k-1)d} \\ (\boldsymbol{X} * \rho_k)(t) \\ \mathbf{0}_d \end{pmatrix} \right) \right\|_2 \\
&\leq \sum_{k=1}^{M} \left\| \sum_{h=\sum_{i=1}^{k-1} H_i+1}^{\sum_{i=1}^{k} H_i} \mathbf{Attn}_t^{(1,h)}(\boldsymbol{X}) - \begin{pmatrix} \mathbf{0}_{(k-1)d} \\ (\boldsymbol{X} * \rho_k)(t) \\ \mathbf{0}_d \end{pmatrix} \right\|_2 \\
&= \sum_{k=1}^{M} \left\| \sum_{h=\sum_{i=1}^{k-1} H_i+1}^{\sum_{i=1}^{k} H_i} \boldsymbol{P}^{(k)} \mathbf{Attn}_t^{(1,h)}(\boldsymbol{X}^{(0)}) - (\boldsymbol{X} * \rho_k)(t) \right\|_2 \\
&\leq \mathcal{E}_{\mathrm{Attn}} := \sum_{k=1}^{M} \frac{C(n)}{H_k^n}, \quad \text{for both } \textbf{Case (A)} \text{ and } \textbf{Case (B)}.
\end{aligned}
$$

Consequently, one detail is to assign the head number $\{H_k\}_{k=1}^{M}$ such that the error's sum $\mathcal{E}_{\mathrm{Attn}}(\texttt{type})$ is as small as possible. Here, we simply choose the same $H_k$:

$$H_k = \frac{H}{M}, \quad k \in [M].$$

Thus, we obtain the bound in Step I:

$$\mathcal{E}_{\mathrm{Attn}} = \sum_{k=1}^{M} \frac{C(n)}{H_k^n} = \frac{C(n) M^{n+1}}{H^n}.$$

Furthermore, by choosing $\mathcal{E}_{\text{Attn}} \leq 1$, it holds that

$$\left\| \boldsymbol{x}_t^{(1/2)} \right\|_\infty \leq \left\| \boldsymbol{x}_t^{(1/2)} - \begin{pmatrix} (\boldsymbol{X} * \rho_1)(t) \\ \vdots \\ (\boldsymbol{X} * \rho_M)(t) \end{pmatrix} \right\|_\infty + \left\| \begin{pmatrix} (\boldsymbol{X} * \rho_1)(t) \\ \vdots \\ (\boldsymbol{X} * \rho_M)(t) \end{pmatrix} \right\|_\infty \leq \mathcal{E}_{\text{Attn}} + 1 \leq 2.$$

**Step II.** Approximate the readout function by FFN layer.

In this step, we aim to approximate the function $f$ using two-layer network. By Lemma G.6, there exists a two-layer neural network with $m$ neurons defined on $\mathbb{R}^D$

$$\text{FFN}^{(1)}(\boldsymbol{y}) = \sum_{k=1}^m a_k \sigma(\boldsymbol{b}_k^\top \boldsymbol{y} + c_k)$$

such that

$$\mathcal{E}_{\text{FFN}} := \left\| \text{FFN}^{(1)} - f \right\|_{L^\infty([-2,2]^D)} \leq \tilde{\mathcal{O}}\left( \frac{\|f\|_\mathcal{B}}{\sqrt{m}} \right).$$

**The final bound.**

For any $t$ and $\boldsymbol{X} \in \mathcal{X}$, it holds that

$$\begin{aligned}
\left\| \mathbf{H}_t(\boldsymbol{X}) - \boldsymbol{x}_t^{(1)} \right\| &= \left| f((\boldsymbol{X} * \rho_1)(t), \cdots, (\boldsymbol{X} * \rho_M)(t)) - \text{FFN}^{(1)}\left( \boldsymbol{x}_t^{(1/2)} \right) \right| \\
&= \left| f((\boldsymbol{X} * \rho_1)(t), \cdots, (\boldsymbol{X} * \rho_M)(t)) - f\left( \boldsymbol{x}_t^{(1/2)} \right) + f\left( \boldsymbol{x}_t^{(1/2)} \right) - \text{FFN}^{(1)}\left( \boldsymbol{x}_t^{(1/2)} \right) \right| \\
&\leq \left| f((\boldsymbol{X} * \rho_1)(t), \cdots, (\boldsymbol{X} * \rho_M)(t)) - f\left( \boldsymbol{x}_t^{(1/2)} \right) \right| + \left| f\left( \boldsymbol{x}_t^{(1/2)} \right) - \text{FFN}^{(1)}\left( \boldsymbol{x}_t^{(1/2)} \right) \right| \\
&\leq \|f\|_{\text{Lip}} \left\| ((\boldsymbol{X} * \rho_1)(t)^\top, \cdots, (\boldsymbol{X} * \rho_M)(t)^\top)^\top - \boldsymbol{x}_t^{(1/2)} \right\|_2 + \left\| f - \text{FFN}^{(1)} \right\|_{L^\infty([-2,2]^D)} \\
&\leq \|f\|_{\text{Lip}} \cdot \mathcal{E}_{\text{Attn}} + \mathcal{E}_{\text{FFN}},
\end{aligned}$$

where

$$\mathcal{E}_{\text{FFN}} = \frac{\|f\|_\mathcal{B}}{\sqrt{m}}; \quad \mathcal{E}_{\text{Attn}} = \frac{C(n)M^{n+1}}{H^n}, \quad \text{for both \textbf{Case (A)} and \textbf{Case (B)}.}$$

Due to the arbitrariness of $t$ and $\boldsymbol{X}$, the proof is completed.

$\square$

# F    Key Lemmas about Approximation

## F.1    Approximation by the sum of exponential decay

**Lemma F.1** (Exp decay, fixed Delta function). *For any $T \in \mathbb{N}_+$, $n, m \in \mathbb{N}_+$, there exists and absolute constant $C(n)$ only depending on $n$ and a $\phi_m^{\exp}(t) = \sum_{k=1}^{m} \alpha_k e^{-\beta_k t}$ such that*

$$\|\mathbb{I}(\cdot = T) - \phi_m^{\exp}(\cdot)\|_{\ell_1(\mathbb{N})} \leq \frac{C(n)e^{0.01(n+1)T}}{m^n}.$$

*where $\beta_k > 0$ holds for any $k \in [m]$.*

*Proof of Lemma F.1.*
Let $\alpha, \gamma > 0$ be constants, and they will take specific values at the end of the proof.

First, recall the standard bump function on $[-1, 1]$:

$$\Psi(x) := \begin{cases} \exp\left(-\frac{1}{1-x^2}\right), & x \in (-1, 1) \\ 0, & \text{otherwise} \end{cases},$$

and we can define the following constants for $T \geq 1$:

$$\mu_T = e^{-\alpha T}, \quad \sigma_T = e^{-\alpha T} - e^{-\alpha(T+1)}.$$

Then we consider the following bump function $\Psi_T \in \mathcal{C}^\infty([0,1])$:

$$\Psi_T(x) = \begin{cases} V_T \Psi\left(\frac{x-\mu_T}{\sigma_T}\right), & x \in (\mu_T - \sigma_T, \mu_T + \sigma_T) \\ 0, & \text{otherwise} \end{cases},$$

where $V_T$ is a scaling constant such that $\Psi_T(e^{-\alpha T}) = e^{\gamma T}$.

First, we consider the approximation of $\Psi_T$ on $[0, 1]$.

Notice that $\Psi_T \in \mathcal{C}^\infty([0,1])$, and $\Psi_T^{(k)}(0) = 0$ for any $k \in \mathbb{N}$. For the standard bump function $\Psi$, for any $n \in \mathbb{N}_+$, there exists an absolute constant $M(n) > 0$ only depending on $n$, such that $\max_{0 \leq k \leq 10} \sup_{x \in [-1,1]} \left|\Psi^{(k)}(x)\right| \leq M(n)$.

Notice that for any $k \in \mathbb{N}$ and $x \in [0, 1]$,

$$\Psi_T^{(k)}(x) = \frac{V_T}{\sigma_T^k} \Psi^{(k)}\left(\frac{x - \mu_T}{\sigma_T}\right).$$

Therefore, the following upper bound holds:

$$M_T(n) = \max_{0 \leq k \leq n} \frac{V_T}{\sigma_T^k} M(n) = \frac{V_T}{\sigma_T^n} M(n)$$

$$= \frac{e^{\gamma T} \cdot e}{\left(e^{-\alpha T} - e^{-\alpha(T+1)}\right)^n} M(n) = \frac{M(n)e}{(1 - 1/e)^n} e^{(\gamma + n\alpha)T} := C(n, \alpha)e^{(\gamma + n\alpha)T}.$$

By Lemma G.5, for any $m \in \mathbb{N}_+$, there exists a polynomial $Q_m(x) = \sum_{k=0}^{m-1} \alpha_k x^k$ such that

$$\sup_{x \in [0,1]} |\Psi_T(x) - Q_m(x)| \leq \frac{M_T(n)}{m^n} \leq \frac{C(n, \alpha)e^{(\gamma + n\alpha)T}}{m^n}.$$

Now we use the transform $x = e^{-\alpha t}$ on the function $\Psi$ and consider

$$\Phi_T(t) := e^{-\gamma t}\Psi_T(e^{-\alpha t}), \quad t \in [0, +\infty).$$

It is easy to verify that $\Phi_T$ satisfies that

$$\Phi_T(t)\big|_{\mathbb{N}} = \mathbb{I}(t = T).$$

Moreover, we consider the function

$$P_m(t) := e^{-\gamma t} Q_m(e^{-\alpha t}), \quad t \in [0, +\infty).$$

Then by choosing $\alpha = \gamma = 0.01$, the following error estimate holds:

$$\|P_m(\cdot) - \mathbb{I}(\cdot = T)\|_{\ell_1(\mathbb{N})} = \sum_{t=0}^{+\infty} |P_m(t) - \Phi_T(t)|$$

$$= \sum_{t=0}^{+\infty} e^{-\gamma t} |Q_m(e^{-\alpha t}) - \Psi_T(e^{-\alpha t})| \leq \sum_{t=0}^{+\infty} e^{-\gamma t} \frac{M_T(n)}{m^n}$$

$$\leq \frac{C(n, \alpha) e^{(\gamma + n\alpha)T}}{m^n} \sum_{t=0}^{+\infty} e^{-\gamma t} \leq \frac{C(n) e^{0.01(n+1)T}}{m^n} \frac{1}{1 - e^{-\gamma}}$$

$$= \frac{\tilde{C}(n) e^{0.01(n+1)T}}{m^n}.$$

Finally, notice that $P_m(t) = e^{-\gamma t} Q_m\left(e^{-\alpha t}\right) = \sum_{k=0}^{m-1} \alpha_k e^{-(0.01 + 0.01k)}$, so we can select $\phi_m^{\exp}(t) := P_m(t)$.

$$\square$$

**Lemma F.2** (Exp decay, adaptive Delta function). *For any $T \in \mathbb{N}$, $n, m \in \mathbb{N}_+$, there exists an absolute constant $C(n)$ only depending on $n$ and a $\phi_m^{\exp}(t; B) = \sum_{k=1}^{m} \alpha_k e^{-\beta_k(t-B)}$ such that*

$$\max_{1 \leq B \leq T} \|\mathbb{I}(\cdot = B) - \phi_m^{\exp}(\cdot; B)\|_{\ell_1(\mathbb{N})} \leq \frac{C(n) e^{0.01(n+1)T}}{m^n}.$$

*where $\beta_k > 0$ holds for any $k \in [m]$.*

*Proof of Lemma F.2.*
The **key point** of the proof is to note that the adaptability of $B$ can be eliminated by the **translation operator** $t - B$.

First, recall our proof of Lemma F.1. For the same $\Psi_T(\cdot)$, for any $n, m \in \mathbb{N}_+$, there exists an absolute constant $C(n)$ only depending on $n$ and a polynomial $Q_m(x) = \sum_{k=0}^{m-1} \alpha_k x^k$ such that

$$\sup_{x \in [0,1]} |\Psi_T(x) - Q_m(x)| \leq \frac{C(n) e^{0.01(n+1)T}}{m^n}.$$

Moreover, using the transform $x = e^{-0.01(t-B+T)}$ $(t \geq 0)$ on the function $\Psi$ and consider

$$\Phi_T(t; B) := e^{-0.01(t-B+T)} \Psi_T\left(e^{-0.01(t-B+T)}\right), \quad t \in [0, +\infty).$$

It is easy to verify that $\Phi_T(\cdot; \cdot)$ satisfies that

$$\Phi_T(t; B)\big|_{\mathbb{N}} = \mathbb{I}(t = B).$$

And we consider the function

$$P_m(t; B) := e^{-0.01(t-B+T)} Q_m\left(e^{-0.01(t-B+T)}\right), \quad t \in [0, +\infty).$$

Then, for any $1 \le B \le T$, the following error estimate holds:

$$\|P_m(\cdot; B) - \mathbb{I}(\cdot = B)\|_{\ell_1(\mathbb{N})} = \sum_{t=0}^{+\infty} |P_m(t; B) - \Phi_T(t; B)|$$

$$= \sum_{t=0}^{+\infty} e^{-0.01(t-B+T)} \left| Q_m\left(e^{-0.01(t-B+T)}\right) - \Psi_T\left(e^{-0.01(t-B+T)}\right) \right|$$

$$\le \sum_{t=0}^{+\infty} e^{-0.01t} \sup_{x \in [0,1]} |Q_m(x) - \Psi_T(x)|$$

$$\le \frac{C(n)e^{0.01(n+1)T}}{m^n} \sum_{t=0}^{+\infty} e^{-0.01t} = \frac{\tilde{C}(n)e^{0.01(n+1)T}}{m^n}.$$

Due to the arbitrariness of $B$, the proof is completed.

$\square$

**Lemma F.3** (Exp decay, fixed Delta function). *Consider a exponentially decayed memory $\rho(\cdot)$: there exists $\beta > 0$ such that $\rho(t) = \mathcal{O}(e^{-\beta t})$. Then for any $n \in [\lfloor 99\beta \rfloor]$ and $m \in \mathbb{N}_+$, there exists an absolute constant $C(n)$ only depending on $n$ and a $\phi_m^{\exp}(t) = \sum_{k=1}^{m} \alpha_k e^{-\beta_k t}$ such that*

$$\|\rho(\cdot) - \phi_m^{\exp}(\cdot)\|_{\ell_1(\mathbb{N})} \le \frac{C(n)}{m^n},$$

*where $\beta_k > 0$ holds for any $k \in [m]$.*

*Proof of Lemma F.3.*
There exists $C > 0$ such that $|\rho(t)| \le Ce^{-\beta t}$.

Let $\alpha, \gamma > 0$ be constants, and they will take specific values at the end of the proof.

First, recall the standard bump function on $[-1, 1]$:

$$\Psi(x) := \begin{cases} \exp\left(-\frac{1}{1-x^2}\right), & x \in (-1, 1) \\ 0, & \text{otherwise} \end{cases},$$

and we can define the following constants for $T \ge 1$:

$$\mu_T = e^{-\alpha T}, \quad \sigma_T = \frac{1}{2}\left(e^{-\alpha T} - e^{-\alpha(T+1)}\right),$$

and we consider the following bump function $\Psi_T \in \mathcal{C}^{\infty}([0, 1])$:

$$\Psi_T(x) = \begin{cases} V_T \Psi\left(\frac{x-\mu_T}{\sigma_T}\right), & x \in (\mu_T - \sigma_T, \mu_T + \sigma_T) \\ 0, & \text{otherwise} \end{cases},$$

where $V_T$ is a scaling constant such that $\Psi_T(e^{-\alpha T}) = e^{\gamma T}\rho(T)$.

Consequently, we consider the sum of bump functions on $[0, 1]$:

$$\varphi(x) := \sum_{T=1}^{+\infty} \Psi_T(x).$$

It is easy to verify that $(\mu_{T_1} - \sigma_{T_1}, \mu_{T_1} + \sigma_{T_1}) \cap (\mu_{T_2} - \sigma_{T_2}, \mu_{T_2} + \sigma_{T_2}) = \varnothing$ for any $T_1 \ne T_2$ and

$$\varphi(x) = \begin{cases} \Psi_T(x), & \mu_T - \sigma_T \le x \le \mu_T + \sigma_T \\ 0, & \text{otherwise} \end{cases}.$$

First, we study the property of $\varphi(\cdot)$.

We denote the absolute constants $M_k = \sup_x |\varphi^{(k)}(x)|$. Notice that for any $k \in \mathbb{N}$,

$$\Psi_T^{(k)}(x) = \frac{V_T}{\sigma_T^k} \Psi^{(k)}\left(\frac{x - \mu_T}{\sigma_T}\right).$$

Therefore, it holds that

$$\sup_{x \in (\mu_T - \sigma_T, \mu_T + \sigma_T)} |\varphi^{(k)}(x)| = \sup_{x \in (\mu_T - \sigma_T, \mu_T + \sigma_T)} |\Psi_T^{(k)}(x)|$$

$$\leq \frac{V_T}{\sigma_T^k} M_k = \frac{e^{\gamma T} \rho(T)}{\left(e^{-\alpha T} - e^{-\alpha(T+1)}\right)^k} M_k e \leq \frac{C M_k e}{(1 - e^{-\alpha})^k} e^{(\gamma + k\alpha - \beta)T}.$$

Therefore, if $\beta \geq \gamma + k\alpha$, then the following uniform bounds holds:

$$\sup_{x \in (0,1]} |\varphi^{(k)}(x)| = \sup_{T \geq 1} \sup_{x \in (\mu_T - \sigma_T, \mu_T + \sigma_T)} |\varphi^{(k)}(x)|$$

$$\leq \sup_{T \geq 1} \frac{C M_k e}{(1 - e^{-\alpha})^k} e^{(\gamma + k\alpha - \beta)T} \leq \frac{C M_k e}{(1 - e^{-\alpha})^k} := C(k, \alpha).$$

Consequently, we consider the smoothness of $\Phi$ at $x = 0$.

Recalling the previous results, for any $x \in (0, 1]$, we have

$$\frac{|\varphi^{(k)}(x)|}{x} \leq C(k, \alpha) \frac{e^{(\gamma + k\alpha - \beta)T}}{\mu_T - \sigma_T} = \frac{2C(k, \alpha)}{1 - e^{-\alpha}} e^{(\gamma + (k+1)\alpha - \beta)T}, \ x \in (\mu_T - \sigma_T, \mu_T + \sigma_T);$$

$$\frac{|\varphi^{(k)}(x)|}{x} = 0, \ \text{otherwise}$$

Thus, by induction, it is easy to verify that for any $i < \frac{\beta - \gamma}{\alpha}$ $(i \in \mathbb{N})$,

$$\varphi^{(i)}(0) = 0.$$

Therefore, for any $n < \frac{\beta - \gamma}{\alpha}$ $(n \in \mathbb{N})$, $\varphi^{(k)}(0) = 0$ holds for any $0 \leq k \leq n$. Moreover, there exists absolute constant $C(n, \alpha)$ such that:

$$\max_{0 \leq k \leq n} \sup_{x \in [0,1]} |\varphi^{(k)}(x)| \leq C(n, \alpha).$$

By Lemma G.5, for any $m \in \mathbb{N}_+$, there exists a polynomial $Q_m(x) = \sum_{k=0}^{m-1} \alpha_k x^k$ such that

$$\sup_{x \in [0,1]} |\varphi(x) - Q_m(x)| \leq \frac{C(n, \alpha)}{m^n}.$$

Now we use the transform $x = e^{-\alpha t}$ $(t \geq 0)$ on the function $\varphi$ and consider

$$\Phi(t) := \frac{1}{e^{\gamma t}} \varphi\left(\frac{1}{e^{\alpha t}}\right), \quad t \in [0, +\infty).$$

It is easy to verify that $\Phi$ satisfies that

$$\Phi(t)\big|_{\mathbb{N}} = \rho(t)\big|_{\mathbb{N}}.$$

Moreover, we consider the function

$$P_m(t) := \frac{1}{e^{\gamma t}} Q_m\left(\frac{1}{e^{\alpha t}}\right), \quad t \in [0, +\infty).$$

Then for any $n < \frac{\beta - \gamma}{\alpha}$ ($n \in \mathbb{N}$), the following error estimate holds:

$$\|P_m(\cdot) - \rho(\cdot)\|_{\ell_1(\mathbb{N})} = \sum_{t=0}^{+\infty} |P_m(t) - \Phi(t)|$$

$$= \sum_{t=0}^{+\infty} e^{-\gamma t} \left| Q_m\left(e^{-\alpha t}\right) - \Psi_T\left(e^{-\alpha t}\right) \right| \leq \frac{C(n, \alpha)}{m^n} \sum_{t=0}^{+\infty} e^{-\gamma t}.$$

By choosing $\alpha = 5 \cdot 10^{-3}$ and $\gamma = 10^{-2}\beta$, it holds that $99\beta < \frac{\beta - \gamma}{2\alpha} = \frac{\beta - \gamma}{\alpha}$.

Thus, we obtain our result: for any $n \in [\lfloor 99\beta \rfloor]$ ($\beta \geq 1/99$), the following error estimate holds:

$$\|P_m(\cdot) - \rho(\cdot)\|_{\ell_1(\mathbb{N})} \leq \frac{C(n)}{m^n} \sum_{t=0}^{+\infty} e^{-\gamma t} = \frac{C(n)}{m^n} \frac{1}{1 - e^{-10^{-2}\beta}} = \frac{\tilde{C}(n)}{m^n}.$$

$\square$

## F.2 Approximation by the sum of polynomial decay

**Lemma F.4** (Poly decay, fixed Delta function). *For any $T, n, m \in \mathbb{N}_+$, there exists an absolute constant $C(n)$ only depending on $n$ and a $\phi_m^{\mathrm{poly}}(t) = \sum_{k=1}^{m} \alpha_k t^{-\beta_k}$ such that*

$$\left\| \mathbb{I}(\cdot = T) - \phi_m^{\mathrm{poly}}(\cdot) \right\|_{\ell_1(\mathbb{N}_+)} \leq \frac{C(n) T^{1.01(n+1)}}{m^n},$$

*where $\beta_k > 1$ holds for any $k \in [m]$.*

*Proof of Lemma F.4.*
Let $\alpha, \gamma > 0$ be constants, and they will take specific values at the end of the proof

First, recall the standard bump function on $[-1, 1]$:

$$\Psi(x) := \begin{cases} \exp\left(-\frac{1}{1-x^2}\right), & x \in (-1, 1) \\ 0, & \text{otherwise} \end{cases},$$

and we can define the following constants for $T \geq 1$:

$$\mu_T = \frac{1}{T^\alpha}, \quad \sigma_T = \frac{1}{T^\alpha} - \frac{1}{(T+1)^\alpha}.$$

Then we consider the following bump function $\Psi_T \in \mathcal{C}^\infty([0, 1])$:

$$\Psi_T(x) = \begin{cases} V_T \Psi\left(\frac{x - \mu_T}{\sigma_T}\right), & x \in (\mu_T - \sigma_T, \mu_T + \sigma_T) \\ 0, & \text{otherwise} \end{cases},$$

where $V_T$ is a scaling constant such that $\Psi_T(\frac{1}{T^\alpha}) = T^{1+\gamma}$.

First, we consider the approximation of $\Psi_T$ on $[0, 1]$.

Notice that $\Psi_T \in \mathcal{C}^\infty([0, 1])$, and $\Psi_T^{(k)}(0) = 0$ for any $k \in \mathbb{N}$. For the standard bump function $\Psi$, for any $n \in \mathbb{N}_+$, there exists an absolute constant $M(n) > 0$ only depending on $n$, such that $\max_{0 \leq k \leq n} \sup_{x \in [-1,1]} \left| \Psi^{(k)}(x) \right| \leq M(n)$.

Notice that for any $k \in \mathbb{N}$ and $x \in [0, 1]$,

$$\Psi_T^{(k)}(x) = \frac{V_T}{\sigma_T^k} \Psi^{(k)}\left(\frac{x - \mu_T}{\sigma_T}\right).$$

Therefore, the following upper bound holds:

$$M_T(n) = \max_{0 \le k \le n} \frac{V_T}{\sigma_T^k} M(n) = \frac{V_T}{\sigma_T^n} M(n) = \frac{T^{1+\gamma} e}{(1/T^\alpha - 1/(T+1)^\alpha)^n} M(n)$$

$$\le \frac{T^{1+\gamma}(T+1)^{n(1+\alpha)} M(n) e}{\alpha^n} \le \frac{2^n M(n) e}{\alpha^n} T^{1+\gamma+n(1+\alpha)} := C(n,\alpha) T^{1+\gamma+n(1+\alpha)}.$$

By Lemma G.5, for any $m \in \mathbb{N}_+$, there exists a polynomial $Q_m(x) = \sum_{k=0}^{m-1} \alpha_k x^k$ such that

$$\sup_{x \in [0,1]} |\Psi_T(x) - Q_m(x)| \le \frac{M_T(n)}{m^n} \le \frac{C(n,\alpha) T^{1+\gamma+n(1+\alpha)}}{m^n}.$$

Now we use the transform $x = \frac{1}{t^\alpha}$ $(t \ge 1)$ on the function $\Psi$ and consider

$$\Phi_T(t) := \frac{1}{t^{1+\gamma}} \Psi_T\left(\frac{1}{t^\alpha}\right), \quad t \in [1, +\infty).$$

It is easy to verify that $\Phi_T$ satisfies that

$$\Phi_T(t)\big|_{\mathbb{N}_+} = \mathbb{I}(t = T).$$

Moreover, we consider the function

$$P_m(t) := \frac{1}{t^{1+\gamma}} Q_m\left(\frac{1}{t^\alpha}\right), \quad t \in [1, +\infty).$$

Then by choosing $\alpha = \gamma = 0.01$, the following error estimate holds:

$$\|P_m(\cdot) - \mathbb{I}(\cdot = T)\|_{\ell_1(\mathbb{N}_+)} = \sum_{t=1}^{+\infty} |P_m(t) - \Phi_T(t)|$$

$$= \sum_{t=1}^{+\infty} \frac{1}{t^{1+\gamma}} \left| Q_m\left(\frac{1}{t^\alpha}\right) - \Psi_T\left(\frac{1}{t^\alpha}\right) \right| \le \sum_{t=1}^{+\infty} \frac{1}{t^{1+\gamma}} \frac{M_T(n)}{m^n}$$

$$\le \frac{C(n,\alpha) T^{1+\gamma+n(1+\alpha)}}{m^n} \sum_{t=1}^{+\infty} \frac{1}{t^{1+\gamma}} = \frac{C(n) T^{1.01(n+1)}}{m^n} \sum_{t=1}^{+\infty} \frac{1}{t^{1+0.01}}$$

$$= \frac{\tilde{C}(n) T^{1.01(n+1)}}{m^n}.$$

Finally, notice that $P_m(\cdot)$ satisfies to $P_m(t) = \frac{1}{t^{1+\gamma}} Q_m\left(\frac{1}{t^\alpha}\right) = \sum_{k=0}^{m-1} \alpha_k t^{-(1.01+0.01k)}$, so we can select $\phi_m^{\text{poly}}(t) := P_m(t)$.

$\square$

**Lemma F.5** (Poly decay, adaptive Delta function). *For any $T, n, m \in \mathbb{N}_+$, there exists an absolute constant $C(n)$ only depending on $n$ and a $\phi_m^{\text{poly}}(t; B) = \sum_{k=1}^{m} \alpha_k (t/B)^{-\beta_k}$ such that*

$$\max_{1 \le B \le T} \left\| \mathbb{I}(\cdot = B) - \phi_m^{\text{poly}}(\cdot; B) \right\|_{\ell_1(\mathbb{N}_+)} \le \frac{C(n) T^{1.01(n+1)}}{m^n},$$

*where $\beta_k > 1$ holds for any $k \in [m]$.*

*Proof of Lemma F.5.*
The **key point** of the proof is to note that the adaptability of $B$ can be eliminated by the **rescaling operator** $t/B$.

First, recall our proof of Lemma F.4. For the same $\Psi_T(\cdot)$, for any $n, m \in \mathbb{N}_+$, there exists an absolute constant $C(n)$ only depending on $n$ and a polynomial $Q_m(x) = \sum_{k=0}^{m-1} \alpha_k x^k$ such that

$$\sup_{x \in [0,1]} |\Psi_T(x) - Q_m(x)| \leq \frac{C(n) T^{1.01(n+1)}}{m^n}.$$

We use the transform $x = \frac{1}{t^{0.01}}$ $(t \geq 1)$ on the function $\Psi$ and consider

$$\Phi_T(t; B) := \left( \frac{B}{tT} \right)^{1.01} \Psi_T \left( \left( \frac{B}{tT} \right)^{0.01} \right), \quad t \in [1, +\infty).$$

It is easy to verify that $\Phi_T(\cdot; \cdot)$ satisfies that

$$\Phi_T(t; B) \big|_{\mathbb{N}_+} = \mathbb{I}(t = B).$$

And we consider the function

$$P_m(t; B) := \left( \frac{B}{tT} \right)^{1.01} Q_m \left( \left( \frac{B}{tT} \right)^{0.01} \right), \quad t \in [1, +\infty).$$

Then, for any $1 \leq B \leq T$, the following error estimate holds:

$$\|P_m(\cdot; B) - \mathbb{I}(\cdot = B)\|_{\ell_1(\mathbb{N}_+)} = \sum_{t=1}^{+\infty} |P_m(t; B) - \Phi_T(t; B)|$$

$$= \sum_{t=1}^{+\infty} \left( \frac{B}{tT} \right)^{1.01} \left| Q_m \left( \left( \frac{B}{tT} \right)^{0.01} \right) - \Psi_T \left( \left( \frac{B}{tT} \right)^{0.01} \right) \right|$$

$$\leq \sum_{t=1}^{+\infty} \frac{1}{t^{1.01}} \sup_{x \in [0,1]} |Q_m(x) - \Psi_T(x)|$$

$$\leq \frac{C(n) T^{1.01(n+1)}}{m^n} \sum_{t=1}^{+\infty} \frac{1}{t^{1.01}} = \frac{\tilde{C}(n) T^{1.01(n+1)}}{m^n}.$$

Due to the arbitrariness of $B$, the proof is completed.

$\square$

**Lemma F.6** (Poly decay, fixed Delta function). *Consider a polynomially decayed memory $\rho(\cdot)$: there exists $\beta > 1$ such that $\rho(t) = \mathcal{O}(t^{-\beta})$. Then for any $n \in [\lfloor 0.99\beta \rfloor - 1]$ and $m \in \mathbb{N}_+$, there exists an absolute constant $C(n)$ only depending on $n$ and a $\phi_m^{\mathrm{poly}}(t) = \sum_{k=1}^{m} \alpha_k t^{-\beta_k}$ such that*

$$\left\| \rho(\cdot) - \phi_m^{\mathrm{poly}}(\cdot) \right\|_{\ell_1(\mathbb{N}_+)} \leq \frac{C(n)}{m^n},$$

*where $\beta_k > 1$ holds for any $k \in [m]$.*

*Proof of Lemma F.6.*
There exists $C > 0$ such that $|\rho(t)| \leq C/t^{\beta}$.

Let $\alpha, \gamma > 0$ be constants, and they will take specific values at the end of the proof

First, recall the standard bump function on $[-1, 1]$:

$$\Psi(x) := \begin{cases} \exp\left( -\frac{1}{1-x^2} \right), & x \in (-1, 1) \\ 0, & \text{otherwise} \end{cases} \quad,$$

and we can define the following constants for $T \geq 1$:

$$\mu_T = \frac{1}{T^\alpha}, \quad \sigma_T = \frac{1}{2} \left( \frac{1}{T^\alpha} - \frac{1}{(T+1)^\alpha} \right),$$

and we consider the following bump function $\Psi_T \in C^\infty([0,1])$:

$$\Psi_T(x) = \begin{cases} V_T \Psi\left(\frac{x - \mu_T}{\sigma_T}\right), & x \in (\mu_T - \sigma_T, \mu_T + \sigma_T) \\ 0, & \text{otherwise} \end{cases},$$

where $V_T$ is a scaling constant such that $\Psi_T(\frac{1}{T^\alpha}) = T^{1+\gamma}\rho(T)$.

Consequently, we consider the sum of bump functions on $[0,1]$:

$$\varphi(x) := \sum_{T=1}^{+\infty} \Psi_T(x).$$

It is easy to verify that $(\mu_{T_1} - \sigma_{T_1}, \mu_{T_1} + \sigma_{T_1}) \cap (\mu_{T_2} - \sigma_{T_2}, \mu_{T_2} + \sigma_{T_2}) = \varnothing$ for any $T_1 \neq T_2$ and

$$\varphi(x) = \begin{cases} \Psi_T(x), & \mu_T - \sigma_T \leq x \leq \mu_T + \sigma_T \\ 0, & \text{otherwise} \end{cases}.$$

First, we study the property of $\varphi(\cdot)$.

We denote the absolute constants $M_k = \sup_x |\varphi^{(k)}(x)|$. Notice that for any $k \in \mathbb{N}$,

$$\Psi_T^{(k)}(x) = \frac{V_T}{\sigma_T^k} \Psi^{(k)}\left(\frac{x - \mu_T}{\sigma_T}\right).$$

Therefore, it holds that

$$\sup_{x \in (\mu_T - \sigma_T, \mu_T + \sigma_T)} |\varphi^{(k)}(x)| = \sup_{x \in (\mu_T - \sigma_T, \mu_T + \sigma_T)} |\Psi_T^{(k)}(x)|$$

$$\leq \frac{V_T}{\sigma_T^k} M_k = \frac{T^{1+\gamma}\rho(T)}{\left(\frac{1}{T^\alpha} - \frac{1}{(T+1)^\alpha}\right)^k} 2^k M_k e$$

$$\leq \frac{(T+1)^{k(1+\alpha)} T^{1+\gamma-\beta} C 2^k M_k e}{\alpha^k} \leq \frac{2^{k(2+\alpha)} C M_k e}{\alpha^k} T^{1+\gamma+k(1+\alpha)-\beta}.$$

Therefore, if $k \leq \frac{\beta - (1+\gamma)}{1+\alpha}$, the following uniform bounds hold:

$$\sup_{x \in (0,1]} |\varphi^{(k)}(x)| = \sup_{T \geq 1} \sup_{x \in (\mu_T - \sigma_T, \mu_T + \sigma_T)} |\varphi^{(k)}(x)|$$

$$\leq \sup_{T \geq 1} \frac{2^{k(2+\alpha)} C M_k e}{\alpha^k} T^{1+\gamma+k(1+\alpha)-\beta} \leq \frac{2^{k(2+\alpha)} C M_k e}{\alpha^k} := C(k, \alpha).$$

Consequently, we consider the smoothness of $\Phi$ at $x = 0$.

Recalling the previous results, for any $x \in (0,1]$, we have

$$\frac{|\varphi^{(k)}(x)|}{x} \leq C(k, \alpha) \frac{T^{1+\gamma+k(1+\alpha)-\beta}}{\mu_T - \sigma_T} \leq \frac{C(k,\alpha) 2^{2+\alpha}}{\alpha} T^{1+\gamma+(k+1)(1+\alpha)-\beta}, \quad x \in (\mu_T - \sigma_T, \mu_T + \sigma_T);$$

$$\frac{|\varphi^{(k)}(x)|}{x} = 0, \text{ otherwise}$$

Thus, by induction, it is easy to verify that for any $i < \frac{\beta - (1+\gamma)}{1+\alpha}$ ($i \in \mathbb{N}$),

$$\varphi^{(i)}(0) = 0.$$

Therefore, for any $n < \frac{\beta-(1+\gamma)}{1+\alpha}$ ($n \in \mathbb{N}_+$), $\varphi^{(k)}(0) = 0$ holds for any $0 \leq k \leq n$. Moreover, the following uniform bound holds:

$$\max_{0 \leq k \leq n} \sup_{x \in [0,1]} |\varphi^{(k)}(x)| \leq C(n, \alpha).$$

By Lemma G.5, for any $m \in \mathbb{N}_+$, there exists a polynomial $Q_m(x) = \sum\limits_{k=0}^{m-1} \alpha_k x^k$ such that

$$\sup_{x \in [0,1]} |\varphi(x) - Q_m(x)| \leq \frac{C(n, \alpha)}{m^n}.$$

Now we use the transform $x = \frac{1}{t^\alpha}$ ($t \geq 1$) on the function $\varphi$ and consider

$$\Phi(t) := \frac{1}{t^{1+\gamma}} \varphi\left(\frac{1}{t^\alpha}\right), \quad t \in [1, +\infty).$$

It is easy to verify that $\Phi$ satisfies that

$$\Phi(t)\big|_{\mathbb{N}_+} = \rho(t)\big|_{\mathbb{N}_+}.$$

Moreover, we consider the function

$$P_m(t) := \frac{1}{t^{1+\gamma}} Q_m\left(\frac{1}{t^\alpha}\right), \quad t \in [1, +\infty).$$

Then for any $n < \frac{\beta-(1+\gamma)}{1+\alpha}$ ($n \in \mathbb{N}$), the following error estimate holds:

$$\|P_m(\cdot) - \rho(\cdot)\|_{\ell_1(\mathbb{N}_+)} = \sum_{t=1}^{+\infty} |P_m(t) - \Phi(t)|$$

$$= \sum_{t=1}^{+\infty} \frac{1}{t^{1+\gamma}} \left|Q_m\left(\frac{1}{t^\alpha}\right) - \Psi_T\left(\frac{1}{t^\alpha}\right)\right| \leq \frac{C(n, \alpha)}{m^n} \sum_{t=1}^{+\infty} \frac{1}{t^{1+\gamma}}.$$

By choosing $\alpha = 10^{-2}$ and $\gamma = 10^{-4}\beta$, we have $0.99\beta - 1 = \frac{\beta-\gamma}{1+\alpha} - 1 = \frac{\beta-(1+\gamma+\alpha)}{1+\alpha} < \frac{\beta-(1+\gamma)}{1+\alpha}$. Thus, we obtain our result: for any $n \in \lfloor \lfloor 0.99\beta \rfloor - 1 \rfloor$ ($\beta \geq 2/0.99$), the following error estimate holds:

$$\|P_m(\cdot) - \rho(\cdot)\|_{\ell_1(\mathbb{N}_+)} \leq \frac{C(n)}{m^n} \sum_{t=1}^{+\infty} \frac{1}{t^{1+\gamma}} \leq \frac{C(n)}{m^n} \sum_{t=1}^{+\infty} \frac{1}{t^{1+10^{-4}}} = \frac{\tilde{C}(n)}{m^n}.$$

$\square$

# G   Some Background and Proof Preparation

## G.1   T5's relative positional encoding

The T5's Relative Positional Encoding is primary focus of this study. Its standard form in practical applications (Raffel et al., 2020) adheres to $R_{t,s} = r(t - s)$, where

$$-r(n) = \begin{cases} n, & \text{if } n < \mathcal{B} \\ \mathcal{B} + \lfloor \mathcal{B} \cdot \frac{\log(n/\mathcal{B})}{\log(\mathcal{D}/\mathcal{B})} \rfloor, & \text{if } \mathcal{B} \le n < \mathcal{D} \\ 2B - 1, & \text{if } n \ge \mathcal{D} \end{cases}.$$

Here, $\mathcal{D}$ is a large integer, signifying the longest distance of concern, while $\mathcal{B}$ is a small integer. One can see that for $n < \mathcal{B}$, $r(\cdot)$ exhibits polynomial decay, whereas for $\mathcal{B} < n < \mathcal{D}$, $r(\cdot)$ demonstrates logarithmic decay. Consequently, the *overall decay rate* of $r(\cdot)$ is *logarithmic*.

The following Table further provides an example of standard T5's Relative Positional Encoding.

Table 1: An example of standard T5's Relative Positional Encoding

| $t - s$ | 0 | 1 | 2 | 3 | 4 | 5 | 6 | 7 | 8 | 9 | 10 | 11 | 12 | 13 | 14 | 15 |
|---|---|---|---|---|---|---|---|---|---|---|---|---|---|---|---|---|
| $-r(t - s)$ | 0 | 1 | 2 | 3 | 4 | 5 | 6 | 7 | 8 | 8 | 8 | 8 | 9 | 9 | 9 | 9 |
| $t - s$ | 16 | 17 | 18 | 19 | 20 | 21 | 22 | 23 | 24 | 25 | 26 | 27 | 28 | 29 | 30 | $\cdots$ |
| $-r(t - s)$ | 10 | 10 | 10 | 10 | 10 | 10 | 10 | 11 | 11 | 11 | 11 | 11 | 11 | 11 | 11 | $\cdots$ |

## G.2   Barron space theory

The well-known universal approximation result for 2NNs asserts that 2NNs can approximate any continuous function (Barron, 1992; 1993; 1994). Nonetheless, this result lacks a characterization of the approximation efficiency, i.e., how many neurons are needed to achieve a certain approximation accuracy? This gap was addressed by the Barron space theory (E et al., 2019; 2021). It is established that for any function within Barron space $f \in \mathcal{B}$, 2NNs with $m$ neurons (denoted by $\mathcal{H}_m$) can approximate them efficiently, at a rate of $\inf_{f_m \in \mathcal{H}_m} \|f - f_m\| \le \mathcal{O}(\|f\|_{\mathcal{B}} / \sqrt{m})$, remarkably independent of the input dimension $d$, thus avoiding the *Curse of Dimensionality* (Bellman, 1966; Bach, 2017). Specifically, the Barron space is defined by:

**Definition G.1** (Barron space (E et al., 2019; 2021; Ma et al., 2020))**.** Consider functions $f : X \to \mathbb{R}$ that admit the following representation:

$$f(\boldsymbol{x}) = \int_{\Omega} a\sigma(\boldsymbol{b}^{\top}\boldsymbol{x} + c)\rho(\mathrm{d}a, \mathrm{d}\boldsymbol{b}, \mathrm{d}c), \ \boldsymbol{x} \in X.$$

For any $p \in [1, +\infty]$, we define the Barron norm:

$$\|f\|_{\mathcal{B}_p} := \inf_{\rho} \left( \mathbb{E}_{\rho}\left[ |a|^p (\|\boldsymbol{b}\|_1 + |c|)^p \right] \right)^{1/p}.$$

Then the Barron space are defined as:

$$\mathcal{B}_p := \{f \in \mathcal{C} : \|f\|_{\mathcal{B}_p} < +\infty\}.$$

**Proposition G.2.** *For any* $p \in [1, +\infty]$, $\mathcal{B}_p = \mathcal{B}_{\infty}$ *and* $\|f\|_{\mathcal{B}_p} = \|f\|_{\mathcal{B}_{\infty}}$.

**Remark G.3.** From the Proposition above, the Barron spaces $\mathcal{B}_p$ are equivalent for any $p \in [1, +\infty]$. Consequently, in this paper, we use $\mathcal{B}$ and $\|\cdot\|_{\mathcal{B}}$ to denote the Barron space and Barron norm.

**Remark G.4.** For Barron space $\mathcal{B}$, both Direct and Inverse Approximation Theorems hold (E et al., 2021). In this paper, we mainly utilize the Direct Approximation Theorem, stated in Lemma G.6.

### G.3 Useful approximation lemmas

**Lemma G.5** (Jackson (1930)). *Let $f \in \mathcal{C}^n([0,1])$ with $f(0) = f'(0) = \cdots = f^{(n)}(0) = 0$. Then for any $m \in \mathbb{N}_+$, there exists a polynomial $Q_m(x) = \sum_{k=0}^{m-1} \alpha_k x^k$ such that*

$$\|f - Q_m\|_{L^\infty([0,1])} \leq \frac{M(n)}{m^n},$$

*where $M(n) = \max_{k \leq n} \|f^{(k)}\|_{L^\infty([0,1])}$.*

**Lemma G.6** (Ma et al. (2020)). *For any $f \in \mathcal{B}$ and $m \in \mathbb{N}$, there exists a two-layer ReLU neural network $f_m(\boldsymbol{x}) = \sum_{k=1}^{m} a_k \sigma(\boldsymbol{b}_k^\top \boldsymbol{x} + c_k)$ with $m$ neurons such that*

$$\|f - f_m\|_{L^\infty([0,1]^d)} \leq \tilde{\mathcal{O}}\left(\frac{\|f\|_{\mathcal{B}}}{\sqrt{m}}\right).$$

# H  Experiments

## H.1  Restatement of our theoretical insights

As detailed in Section 1, our theoretical analysis reveals the following novel insights into the expressive power and mechanisms of Transformer:

**Insight (1). The distinct roles of the number of layers, the number of Attn heads, and the width of FFN layers. (1a)** Deeper Transformers can handle tasks with memories with more intricate interrelationships, such as nested relationships (Type II). **(1b)** In contrast, for tasks with memories lacking such interrelationships (Type I), a single-layer Transformer with sufficient Attn heads and FFN width should suffice.

**Insight (2). The different roles of Attn layers and FFN layers. (2a)** FFN layers are tasked with approximating nonlinear memory functions and the readout function, **(2b)** while Attn layers are responsible for extracting the tokens from the memory locations.

**Insight (3). The functionality and necessity of Dot-product (DP). (3a)** For the relatively simple Task I, DP is not necessary and can be omitted. **(3b)** However, for the more complex Task II, DP provides necessary nonlinearity: the cooperation between DP and RPE provides the needed interaction between the temporal space and the token space.

**Insight (4). The efficiency of Relative Positional Encoding (RPE) in modeling long-range correlations.** The primary role of RPE is to approximate the memory kernels. **(4a)** Transformer with log-type RPE can handle heavy-tailed memories. **(4b)** Transformer with lin-type RPE can handle light-tailed memories.

## H.2  Experimental Validation

To validation of our theoretical **insights (1a)∼(4b)**, we conduct 8 experiments, from simple toy models to more complex LLM pre-training. The experiments are conducted on 1 A100.

### H.2.1  Validation of Insight (1a)

*Objective.* As indicated in Section 4, numerous NLP tasks exhibit complex interrelationships among tokens and belong to our Task II. This experiment aims to verify our Insight (1a): for such tasks, increasing the number of layers $L$ is more efficient than increasing the number of Attn heads $H$.

*Setup.* Specifically, we pretrain decoder-only Transformers (Vaswani et al., 2017) with different $L$ and $H$ on the OpenWebText dataset (Gokaslan and Cohen, 2019) for 10,000 iterations (approximately 1B tokens) on 1 A100, using cross-entropy loss and AdamW with the same hyperparameters. To ensure comparability, we meticulously balance the total number of parameters across both experimental setups.

*Results and conclusion.* The final validation losses are shown in Table 2. By comparing these two subtables, the benefits brought by increasing $L$ are much greater than the benefits brought by increasing $H$ (0.802 > 0.136), thereby corroborating our Insight (1a).

Table 2: Results of the experiment supporting Insight (1a).

| $L = 1, H = 8$ (26M) | $L = 1, H = 12$ (29M) | $L = 1, H = 12$ (32M) |
|---|---|---|
| 5.796 (baseline) | 5.689 ($\downarrow$ 0.107) | 5.660 ($\downarrow$ 0.136) |

| $L = 1, H = 8$ (26M) | $L = 4, H = 8$ (29M) | $L = 8, H = 8$ (32M) |
|---|---|---|
| 5.796 (baseline) | 5.374 ($\downarrow$ 0.422) | 4.994 ($\downarrow$ 0.802) |

### H.2.2  Validation of Insight (1b)

*Objective.* As mentioned in Section 3, sparse Boolean functions have no interactions among the memories and belong to our Task I. This experiment aims to verify our Insight (1b): for such tasks, a single-layer Transformer equipped with a sufficient number of Attn heads $H$ and FFN width $m$ suffices, and there is no need to increase the number of layers $L$.

*Setup.* Specifically, we train single-layer DP-free Transformers with different $H$ and $m$ to learn a sparse Boolean target function $f^*$: $f^*(\boldsymbol{x}) := g^*(x_{48}, x_{56}, x_{99}) := \sum_{k=1}^{64} \mathrm{ReLU}(\langle w_k^*, (x_{48}, x_{56}, x_{99}) \rangle)$ for input sequence $\boldsymbol{x} = (x_1, \cdots, x_{1000}) \in \{\pm 1\}^{1000}$, where $\boldsymbol{w}_k^*$ are generated by $\boldsymbol{w}_k^* \sim N(0, I_3)$. Training proceeds for 10,000 iterations (1M samples) using squared loss and AdamW with the same hyperparameters.

*Results and conclusion.* The final validation losses are shown in Table 3. As shown in this table, a single-layer Transformer equipped with a sufficient $H$ (32) and $m$ (256) is adequate for representing this sparse Boolean function. This empirical evidence supports our Insight (1b).

Table 3: Results of the experiment supporting Insight (1b).

| $H = 2, m = 16$ | $H = 8, m = 64$ | $H = 32, m = 256$ |
|---|---|---|
| 0.21 | 0.04 | 0.01 |

### H.2.3 Validation of Insight (2a)

*Objective.* This experiment aims to verify our Insight (2a): to learn a sparse Boolean function with a "complex" readout function and "simple" memories, increasing the FFN width $m$ can significantly improve the performance, whereas increasing the number of Attn heads $H$ brings almost no benefit.

*Setup.* Specifically, we train single-layer DP-free Transformers with different $H$ and $m$ to learn a sparse Boolean function with a "complex" readout function ($g^*$) and a "simple" single memory ($x_{99}$): $f^*(\boldsymbol{x}) := g^*(x_{99}) := \sum_{k=1}^{64} \mathrm{ReLU}(\boldsymbol{w}_k^* \cdot x_{99})$ for any input sequence $\boldsymbol{x} = (x_1, \cdots, x_{1000}) \in \{\pm 1\}^{1000}$, where $\boldsymbol{w}_k^*$ are generated by $\boldsymbol{w}_k^* \sim N(0, 1)$. Training proceeds for 10,000 iterations (1M samples) using squared loss and AdamW with the same hyperparameters.

*Results and conclusion.* The final validation losses are shown in Table 4. The tables indicate that, for learning a sparse Boolean function with a "complex" readout function and "simple" memories, increasing $m$ can significantly improve the performance ($0.49 \rightarrow 0.002$), almost completing this task perfectly. Conversely, increasing $H$ fails to yield substantial improvement. This empirical evidence supports our Insight (2a).

Table 4: Results of the experiment supporting Insight (2a).

|  | $m = 8$ | $m = 64$ | $m = 512$ |  | $H = 8$ | $H = 64$ | $H = 512$ |
|---|---|---|---|---|---|---|---|
| $H = 8$ | 0.49 | 0.006 | 0.002 | $m = 8$ | 0.49 | 0.49 | 0.52 |

### H.2.4 Validation of Insight (2b)

*Objective.* Contrasting with Experiment (2a), this experiment aims to verify our Insight (2b): for learning a sparse Boolean function with a "simple" readout function and "complex" memories, increasing the number of Attn headers $H$ can substantially improve the performance while increasing FFN width $m$ will offer almost no benefit.

*Setup.* Specifically, we train single-layer DP-free Transformers with different $H$ and $m$ to learn a sparse Boolean function with a "simple" linear readout function ($g^*$) and relatively "complex" memories ($x_{48}, x_{56}, x_{99}$): $f^*(\boldsymbol{x}) := g^*(x_{48}, x_{56}, x_{99}) := x_{48} + x_{56} + x_{99}$ for any input sequence $\boldsymbol{x} = (x_1, \cdots, x_{1000}) \in \{\pm 1\}^{1000}$. Training processes for 10,000 iterations (1M samples), using squared loss and AdamW with the same hyperparameters.

*Results and conclusion.* The final validation losses are presented in Table 5. The tables indicate that, for learning a sparse Boolean function with a "simple" readout function and "complex" memories, increasing $m$ can significantly improve the performance ($1.16 \rightarrow 10^{-6}$), closely achieving task perfection. In contrast, increasing $m$ brings almost no benefits. This empirical evidence supports our Insight 2(b).

### H.2.5 Validation of Insight (3a)

*Objective.* As mentioned in Section 3, learning sparse Boolean functions has no interactions among the memories and belongs to our Task I. This experiment aims to verify our insight (3a): for such

Table 5: Results of the experiment supporting Insight (2b).

| | $m = 2$ | $m = 64$ | $m = 256$ | | $H = 2$ | $H = 64$ | $H = 256$ |
|---|---|---|---|---|---|---|---|
| $H = 2$ | 1.16 | 0.81 | 1.23 | $m = 2$ | 1.16 | <1e-6 | <1e-6 |

tasks, a DP-free Transformer equipped with a sufficient number of Attn heads $H$ and FFN width $m$ is sufficiently capable. Moreover, there is no need to use DP structure in Attn.

*Setup.* Specifically, we train single-layer DP-free Transformers with different $H$ and $m$ and with DP or without DP to learn a sparse Boolean target function $f^*$: $f^*(\boldsymbol{x}) := g^*(x_{48}, x_{56}, x_{99}) := \sum_{k=1}^{64} \mathrm{ReLU}(\langle \boldsymbol{w}_k^*, (x_{48}, x_{56}, x_{99}) \rangle)$ for input sequence $\boldsymbol{x} = (x_1, \cdots, x_{1000}) \in \{\pm 1\}^{1000}$, where $\boldsymbol{w}_k^*$ are generated by $\boldsymbol{w}_k^* \sim N(0, I_3)$. Training proceeds for 10,000 iterations (1M samples) using squared loss and AdamW with the same hyperparameters.

*Results and conclusion.* The final validation losses are shown in Table 6. The findings illustrate that a DP-free Transformer equipped with a sufficient $H$ (32) and $m$ (256) is adept at accurately representing the given sparse Boolean function. Additionally, the Incorporation of the DP structure into the layers contributes marginally to performance enhancement. This substantiates our Insight (3a).

Table 6: Results of the experiment supporting Insight (3a).

| | $H = 2, m = 16$ | $H = 8, m = 64$ | $H = 32, m = 256$ |
|---|---|---|---|
| with DP | 0.21 | 0.04 | 0.01 |
| without DP | 0.17 | 0.11 | 0.02 |

### H.2.6 Validation of Insight (3b)

*Objective.* As indicated in Section 4, numerous NLP tasks exhibit complex interrelationships among tokens and belong to our Task II. This experiment aims to verify our Insight (3b): for such tasks, the utilization of DP structure in Attn layers is necessary.

*Setup.* Specifically, we pre-train Transformers with DP or without DP on the OpenWebText dataset for 10,000 iterations (approximately 1B tokens) on 1 A100, using cross-entropy loss and AdamW with the same hyperparameters.

*Results and conclusion.* The final validation losses are presented in Table 7. As shown in the table, for NLP pre-training tasks, Transformer incorporating DP structure is more efficient than Transformer without DP ($5.796 < 5.830, 5.374 < 5.486, 4.994 < 5.274$), thereby supporting our Insight 3(b).

Table 7: Results of the experiment supporting Insight (3b).

| | $L = 1, H = 8$ | $L = 4, H = 8$ | $L = 8, H = 8$ |
|---|---|---|---|
| with DP | 5.796 | 5.374 | 4.994 |
| without DP | 5.830 | 5.486 | 5.274 |

### H.2.7 Validation of Insight (4a)

*Objective.* This experiment aims to verify our Insight (4a): for learning Task III with heavy-tailed memories, Transformers with log-type RPE are efficient, whereas those with lin-type RPE fail.

*Setup.* Specifically, we train single-layer, FFN-free, DP-free Transformers with log-type RPE or lin-type RPE and varying numbers of Attn heads $H$. The target function involves a heavy-tailed memory kernel $\rho(t) = t^{-0.5}$: $f^*(\boldsymbol{x}) := \sum_{s=1}^{1000} x_s \rho(1000 - s)$ for any input sequence $\boldsymbol{x} = (x_1, \cdots, x_{1000}) \in \{\pm 1\}^{1000}$. Training processes for 10,000 iterations (1M samples) using squared loss and AdamW with the same hyperparameters.

*Results and conclusion.* The final validation losses are shown in Table 8. As shown in the table, to learn heavy-tailed memories, even single-head Transformer with log-type RRE can complete it

perfectly ($< 10^{-5}$). Conversely, Transformers employing lin-type RRE exhibit limited improvement even with up to 64 heads (0.19). This empirical evidence supports our Insight (4a).

Table 8: Results of the experiment supporting Insight (4a).

|          | $H = 1$ | $H = 4$ | $H = 16$ | $H = 64$ |
|----------|---------|---------|----------|----------|
| type=log | <1e-5   | <1e-5   | <1e-5    | <1e-5    |
| type=lin | 0.73    | 0.68    | 1.16     | 0.19     |

### H.2.8 Validation of Insight (4b)

*Objective.* In contrast to Experiment (4a), this experiment aims to verify that for learning our Task III with light-tailed memories, Transformers with lin-type RPE are efficient, whereas those with log-type RPE fail.

*Setup.* Specifically, we train single-layer, FFN-free, DP-free Transformers with log-type RPE or lin-type RPE and varying numbers of Attn heads $H$. The target function involves a heavy-tailed memory kernel $\rho(t) = e^{-5t}$: $f^*(\boldsymbol{x}) := \sum_{s=1}^{1000} x_s \rho(1000 - s)$ for any input sequence $\boldsymbol{x} = (x_1, \cdots, x_{1000}) \in \{\pm 1\}^{1000}$. Training processes for 10,000 iterations (1M samples) using squared loss and AdamW with the same hyperparameters.

*Results and conclusion.* The final validation losses are shown in Table 9. As shown in the table, to learn light-tailed memories, even single-head Transformer with lin-type RRE can complete it perfectly ($< 10^{-7}$). Conversely, Transformers employing log-type RRE exhibit limited improvement even with up to 64 heads ($5.3 \times 10^{-4}$). This empirical evidence supports our Insight (4b).

Table 9: Results of the experiment supporting Insight (4b).

|          | $H = 1$ | $H = 4$ | $H = 16$ | $H = 64$ |
|----------|---------|---------|----------|----------|
| type=log | 9.1e-4  | 3.7e-3  | 2.6e-3   | 5.3e-4   |
| type=lin | <1e-7   | <1e-7   | <1e-7    | <1e-7    |

### H.3 Practical Implications

Our theoretical insights and empirical evidence can directly lead to the following 8 practical implications, such as the strategic selection of Transformer hyperparameters for specific tasks.

- **Implication (1a).** (supported by Insight (1a) and Experiment (1a))

  For sequence modeling tasks with complex interrelations between memories, such as many NLP applications, enhancing the number of layers $L$ is more beneficial than increasing the number of Attn heads $H$ and FFN width $m$.

- **Implication (1b).** (supported by Insight (1b) and Experiment (1b))

  For simple sequence modeling tasks with almost no memory intercorrelation, such as the learning of sparse Boolean function, improving performance necessitates only sufficient $H$ and $m$ in a single-layer Transformer, without a need to increase $L$.

- **Implication (2a).** (supported by Insight (2a) and Experiment (2a))

  For sequence modeling tasks with complex readout or memory functions, increasing $m$ can significantly improve performance.

- **Implication (2b).** (supported by Insight (2b) and Experiment (2b))

  For sequence modeling tasks with multiple memories, increasing $H$ can markedly improve performance.

- **Implication (3a).** (supported by Insight (3a) and Experiment (3a))

  For simple sequence modeling tasks with almost no memory correlations, such as learning sparse Boolean functions, omitting the DP structure in Attn layers can still perform well.

- **Implication (3b).** (supported by Insight (3b) and Experiment (3b))

For sequence modeling tasks with complex correlations between memories, such as many NLP tasks, preserving the DP structure in attention layers is crucial for achieving high performance due to its indispensable nonlinearity.

- **Implication (4a).** (supported by Insight (4a) and Experiment (4a))

  For sequence modeling tasks with heavy-tailed memories, the employment of log-type RPE (such as T5's RPE and KERPLE (log)) is recommended over lin-type RPE (such as Alibi).

- **Implication (4b).** (supported by Insight (4b) and Experiment (4b))

  For sequence modeling tasks with light-tailed memories, the employment of lin-type RPE (such as Alibi) is recommended over log-type RPE (such as T5's RPE and KERPLE (log)).

