# OpenReview forum: "Understanding the Expressive Power and Mechanisms of Transformer for Sequence Modeling"
_NeurIPS.cc/2024/Conference — NeurIPS 2024 poster_

### Official Review · Reviewer_Fdzw · 2024-06-14

**Soundness:** 3
**Presentation:** 2
**Contribution:** 3
**Rating:** 6
**Confidence:** 3

**Summary:**

This paper conducts a systematic study on the expressive power and mechanisms of Transformers for sequence modeling. It explores various properties of Transformers, including the number of layers, attention heads, width, and the use of dot production. The theoretical results are supported by experimental evidence.

**Strengths:**

This paper appears to be technically solid. It provides a detailed formulation of the problem and explores the expressive capabilities of Transformers across various tasks. Notably, it includes experiments in Appendix H to validate its theoretical insights. However, I have not verified the correctness of the proofs.

The paper contains some intriguing findings. I particularly want to highlight the exploration between dropout (DP) and relative position encoding (RPE), which I find especially interesting.

**Weaknesses:**

The key weakness of this paper lies in its unclear presentation of motivation, innovation, and contributions. The detailed reasons are as follows:

1) The motivation of this paper appears weak. The authors state their motivation as "to gain a better understanding of how Transformers work in practice." However, there are already numerous studies exploring the expressive power of Transformers, covering most of the topics in this paper. It is unclear why the authors believe the current understanding of Transformers is insufficient and why the community needs this paper.

2) The technical contributions of this work are not clearly presented. It is difficult to discern which conclusions are derived from known techniques and which are based on new techniques proposed by the authors. Additionally, the nature of these new techniques is not clearly explained.

3) The paper's analysis is based on dividing tasks into three categories: modeling fixed, long but sparse memories; modeling adaptive, long but sparse memories; and modeling essentially sparse memories. However, the necessity of this categorization and the relationships between these tasks are unclear. The authors claim these categories are prevalent, but the supporting evidence is weak.

**Questions:**

1) This paper includes numerous proofs, making it challenging to verify their correctness. Could the authors provide a proof sketch for Theorem 4.4?

2) The tightness of the bound is also uncertain. Could the authors provide a numerical result demonstrating the bound in a toy case?

---

> ### Author Rebuttal · Authors · 2024-08-04
>
> We appreciate the reviewer's recognition of our work and helpful comments. Below, we provide detailed responses to the reviewer’s questions.
>
> - **W1.** The motivation of this paper appears weak.
>
>    **Response:** Thank the reviewer for this inquiry.
>    While Transformer shows remarkable capabilities in long-sequence modeling, its underlying mechanisms remain poorly understood. Specifically, several key questions inspire our work, as detailed in l.15–l.30:
>   - How do the key hyper-parameters, for example, the number of layers ($L$), the number of Self-attention (Attn) heads ($H$) and the with of feed-forward network (FFN) layers ($m$), affect the performance of the Transformer network?
>   - How do Attn and FFN layers contribute differently to the overall performance?
>   - How does Dot-product (DP) self-attention work, and is the DP structure necessary?
>   - How efficient is positional encoding (PE) in modeling long-range correlations?
>
>   Furthermore, Appendix A contains a detailed investigation of related works on the expressive power of Transformers. To the best of our knowledge, our manuscript is the **first work** that comprehensively addresses **all the above questions** within the scenario of long-sequence modeling.
>
>
> - **W2.** The technical contributions of this work are not clearly presented.
>
>   **Response:** Thanks for this question. At a high level, our proofs belong to approximation theory, which inevitably involves the use of some standard approximation theory results, such as Lemma G.5 and G.6. However, the most critical step in our main theorems is based on constructive proofs. Our construction employs the specific structure of RPE, DP, and the modeling task, in contrast to many previous proofs such as those in [1][2]. In the revised version, we will add a remark to clarify our technical contributions.
>
>
> - **W3.** The necessity of this categorization and the relationships between these tasks are unclear.
>
>   **Response:** Thanks for this question. We would like to clarify the necessity and relationships of our three task categories:
>   - **Task Complexity and Relevance:** The three tasks exhibit varying complexity, with Task II and III being more complex than Task I. They are relevant to a wide range of application areas: Task I relates to sparse Boolean functions and the traditional n-gram model in NLP; Task II pertains to various NLP tasks, such as dependency parsing, sentiment analysis, part-of-speech tagging, and continuation writing; Task III includes feature representation in CV and wavelet analysis in classical signal processing. Please refer to Section 3-5 in our manuscript for more details.
>   - **Insights from Separate Analysis:** More importantly, analyzing these distinct tasks separately provides insights into different components of the Transformer architecture. For example, studying Task III reveals the efficiency of positional encoding (PE) in modeling long-range correlations, which can not be covered by studying Task I/II; Analyzing Task II illustrates how the number of layers and the number of Attn heads affect the expressivity of Transformers, which can not be revealed by studying Task I/III; Comparing the analysis of Task I and Task II shows when DP is necessary and how DP and PE work together.
>
> - **Q1.** Proof sketch for Theorem 4.4.
>
>   **Response:** Thank the reviewer for this question. Below, we provide a proof sketch for Theorem 4.4. Please refer to our proof route in l.967 to follow this sketch:
>   - Under the setting of Theorem 4.4, the memories are $K$-Adaptive, long but M-sparse (see l.200). We consider the case where $M>K+1$. For the initial $K$ memories, there exits a nested structure, while this nested structure does not exist for the $(K+1,\cdots, M)$-th memories.
>   - In our proof, the nested structure within the initial $K$ memories mandates sequential processing in the first $K$ layers, one by one. Then, in the $(K+1)$-th layer, the remaining $M-K$ non-nested memory functions $t_{K+1},\cdots,t_M$ are processed concurrently.
>   - In each layer, the FFN sublayer is tasked with approximating nonlinear memory functions, such as $t_i=g_i(X)$ and the readout function $f$, while the Attn sublayer is responsible for extracting the tokens from these memory locations, such as $x_{t-t_i}$.
>
>   In our revised version, we will include more proof sketches in the main text for clarity.
>
> - **Q2.** The tightness of the bound is also uncertain. Could the authors provide a numerical result demonstrating the bound in a toy case?
>
>   **Response:** Thank the reviewer for offering this constructive suggestion. We have conducted a new experiment to verify the tightness of our bound. Due to time constraints, we focus on Theorem 3.1 for the case "type=lin".
>   - **Objective.** For simplicity, we consider a single memory $T$. We aim to verify the following two bounds are tight: (i)*$H$ v.s. error*. Given a memory location $T$, the error $\epsilon\lesssim\frac{1}{{\rm poly}(H)}$; (ii) *$H$ v.s. $T$*. To achieve the same error $\epsilon$, we need $H\gtrsim\exp(T)$ heads.
>   - **Setup.** We train single-layer DP-free Transformers with different numbers of heads ($H$) to learn a simple sparse Boolean function, which is within our theoretical framework. Specifically, the input sequence $X=(x_1,\cdots,x_{10})\in\\{\pm1\\}^{10}$, the target output is $x_{11}^*=x_{10-T}$, where $T\in[9]$ reflects the memory location.
>   - **Results & Conclusion.** The results are shown in **the PDF** in our **Global Response to All Reviewers**, supporting the tightness of our bounds: (i) given a memory location $T$, the error $\epsilon\sim\frac{1}{{\rm poly}(H)}$. On the other hand, to achieve the same error $\epsilon$, we need at least $H\sim\exp(T)$ heads.
>
>
> [1] Jiang and Li. Approximation Rate of the Transformer Architecture for Sequence Modeling. (2023)
>
> [2] Edelman et al. Inductive Biases and Variable Creation in Self-Attention Mechanisms. (ICML 2022)

---

> > ### Comment · Reviewer_Fdzw · 2024-08-12
> >
> > I appreciate the authors' detailed response and confirm that they have addressed all the concerns raised. I also find the newly added experiments to be both interesting and insightful. Based on this, I have decided to increase my score to 6.

---

> > > ### Author Response · Authors · 2024-08-12
> > >
> > > We would like to reiterate our gratitude for your valuable recommendation and positive feedback. Thank you for raising the score!

---

### Official Review · Reviewer_iDXk · 2024-07-12

**Soundness:** 4
**Presentation:** 3
**Contribution:** 3
**Rating:** 6
**Confidence:** 3

**Summary:**

The paper investigates the approximation capabilities of transformers with relative positional encodings and derives approximation rates for three types of sequence modeling tasks. Each task corresponds to a particular choice of target function class. Namely, a first class of mappings are fixed, long but sparse memories (equation (5)), and the authors focus on dot-product free transformers. A second class are adaptive, long but sparse memories (equation (8)), and the authors consider deep transformers with specific precision and no SoftMax. The last class of mappings are essentially sparse memories and consist of convolving the input sequence with different kernels where, again, the considered Transformers are dot-product free.
Each approximation rate is commented on the role of the feedforward and self-attention components, the role of dot-product attention (that is, the fact that the attention maps depend on the input sequence through pairwise interactions), and the effect of relative position encoding (whether it is of linear or logarithmic type).

**Strengths:**

- The paper is well written and easy to follow.

- The mathematical results are all supported with detailed and rigorous proofs.

- Each Theorem is structured in the same way which eases the understanding of the whole paper.

- The paper adresses the important question of approximation rates of Transformers in sequence modeling.

- The paper establishes rigorous approximation results for Transformers on sequences modeling tasks. These results are significant and give insights on the role of the different Transformer components on their approximation capabilities.

- The three problem formulations are clearly defined and commented.

**Weaknesses:**

- Unless I missed something, the "true" attention mechanism (line 104) is never used in the paper to derive approximation rates. In sections 3 and 5 the attention is DPF, while in section 4 there is no softmax and quantization is used. I don't think this is a major problem, but it should be emphasised (perhaps by extending remark 2.2) that you never consider the transformer with all the bells and whistles.

- I understand that the dimension of the embeddings grows with $Md$. This is not mentioned in the main text, nor commented on in the supplement. How does this compare with previous work?

- Also, you mention that the number of heads needed grows at least polynomially with the $T_i$'s. I understand that this is due to the particular choice of RPE. Would other types of PE give a better dependency in the $T_i$'s ?  How does the current dependency compare with previous works?

- Paragraph l. 274 to l. 277 is unclear. When do you use this alternative in the rest of the paper?

- Typo in l. 281

- l. 282, I don't understand the notation for the lower bound on $m$.

**Questions:**

- Please see some questions in the Weaknesses part.

- What do you mean by rate $n$ in all the theorems ? Is there an optimal choice for $n$ ?

- What is the main reason why you did not consider the SoftMax in section 4 ?

- How does $C(n)$ behaves with $n$ ?

- Could you approximate the model with specific precision with another Transformer which does not rely on specific precision ?

- Overall, the spirit of the approximation in the paper is always to form augmented tokens $(x_{t}, x_{t-t_1}, \dots x_{t-t_M})$ using attention and to apply the MLP on top of it. Do you think this is the "optimal" way ? And if so, what are the reasons behind it ?

**Limitations:**

Yes, the authors adequately addressed the limitations of their work.

---

> ### Author Rebuttal · Authors · 2024-08-04
>
> We thank the reviewer for the appreciation of our work and insightful comments. We answer the reviewer's questions in the following.
>
> - **W1.** Never consider transformers with all the bells and whistles.
>
>   **Response:** The reviewer is correct. Simplifying models appropriately in different settings helps avoid complex technical proofs and, more importantly, can accurately reveal the distinct roles and necessity of various components in Transformers. In the revised version, we will clarify this point in Remark 2.2.
>
>
> - **W2.** Embeddings dim.
>
>   **Response:** Thank the reviewer for this insightful comment.
>
>   *Comparison with [1]:* For the Transformer $f$ in our paper, we consider embedding dim $D(f)\sim M d$. Notably, this is independent of the sequence length $T$ and represents almost the minimum embedding dim required to extract $M$ $d$-dim memories simultaneously. In contrast, the Transformer $g$ [1] considers a larger embedding dim $D(g)\sim T^2 d$ ($D(g)\gg D(f)$ due to $T\gg M$) and prove that in this setting, 1-layer Transformers are sufficient for sequence modeling, whereas our results require multiple layers for our Task II.
>
>
> - **W3.** The number of heads.
>
>   **Response:** Thank the reviewer for this insightful question. The reviewer is correct that our required number of heads is due to the use of a specific RPE.
>
>   *Comparison with [2]*, which considers Transformer with absolute position encoding (APE) to study Task I. For a fair comparison, let $T:=\max_{i} T_i$. In our work, the model $f$ requires at least $H(f)\sim\text{poly}(T)$ heads, with the number of position parameters $N_p(f)=H(f)\sim\text{poly}(T)$. In contrast, the model $g$ (using APE) in [2] has $N_p(g)\sim Td$ position parameters, and $1$-head Transformers ($H(g)=1$) are sufficient for perform sequence modeling, whereas our model requires $H(f)\sim\text{poly}(T)$ heads. Therefore, model $g$ in [2] requires fewer heads than our model $f$. However, the relationship between the number of position parameters $N_p(f)$ and $N_p(g)$ remains uncertain.
>
> - **W4.** l.274-l.277, alternative.
>
>   **Response:** Thanks for this inquiry. To clarify, this alternative is only used in Prop. 4.3; all other results do not utilize this alternative.
>
> - **W5.** Typo in l. 281.
>
>   **Response:** We are grateful to the reviewer for identifying the typo. We will carefully read through the whole paper and correct all typos.
>
>
> - **W6.** l. 282, lower bound on $m$.
>
>   **Response:** We apologize for the confusion. For example, for type = lin, this lower bound should be $m\gtrsim\max\Big\\{\max_{i\in[K]} ||g_i||_B^2,\sum\_{i=K+1}^M ||g_i||_B^2\Big\\}$.
>
> - **Q1.** Optimal choice for $n$.
>
>   **Response:** Thanks for the inquiry. Given $T_1,\cdots,T_M$, and $H$, there exits an optimal choice for $n$ that minimizes the error term $\mathcal{E}_\text{Attn}(\text{Type})$, although calculating it may not be straightforward.
>
> - **Q2.** The reason for removing SoftMax in section 4 ?
>
>   **Response:** Thank the reviewer for this question. This is a technical operation to simplify the proof. Generally, we use the numerator of SoftMax in a series of attention heads to approximate the memory locations. Hence, we need to cancel the normalization in the SoftMax in each attention head by using $W_V$. However, in Sec.4, the existence of Dot-product results in the normalization in SoftMax depending on the tokens, which cannot be canceled by constant $W_V$. For more details, please refer to our proofs.
>
> - **Q3.** How does $C(n)$ behaves with $n$?
>
>   **Response:** Thanks for the inquiry. $C(n)\leq A^n$, where $A$ is an absolute constant. Notably, although $C(n)$ grows exponentially with $n$, our approximation results remain efficient because the denominator term $H^n$ also grows exponentially, resulting in $\frac{C(n)}{H^n}\leq(\frac{A}{H})^n$.
>
> - **Q4.** Could you approximate the model with specific precision with another Transformer which does not rely on specific precision ?
>
>   **Response:** Thank the reviewer for their insightful comment. First, we would like to clarify that FFN with specific precision is a technical trick to handle the discrete values of the time.
>
>   Let the two-layer FFN with specific precision be $\tilde{f}:=[f]$. Notably, in our setting, we require an *exact representation* of $\tilde{f}$ rather than an approximation. Note that $\tilde{f}$ is a "step function", which is discontinuous at "step points". Thus, any FFN $g$ with continuous activation (such as ReLU and Sigmoid) cannot represent $\tilde{f}$ because $g$ is continuous.
>
>
> - **Q5.** Do you think this is the "optimal" way ? And if so, what are the reasons behind it ?
>
>   **Response:** Thank the reviewer for this enlightening question.
>   -  We believe this approach is general and natural. As the reviewer commented, our main idea is to form $(x_t,x_{t-t_1},\cdots,x_{t-t_M})$ using multi-layer Transformers. In this process, FFN layers and Attn layers work together, each performing their respective strengths: FFN layers approximate complex nonlinear memory functions, and Attn layers are responsible for extracting tokens at the memory locations. Additionally, our Experiment G2 and G3 verify these insights into the roles of FFN and Attn.
>   - However, it should be noted that we make minimal assumptions about the data. Therefore, we do not rule out the existence of better way for special problems. For example, if the data is distributed on a simple low-dimensional manifold $S$ with dimension $r$ ($r<d$), we only need to form $(P_S x_{t-t_1},\cdots,P_S x_{t-t_M})$ using attention and apply an FFN on top of it, where $P_S$ is the projection to the manifold $S$. In this case, the embedding dim only needs to be $rd$, which is smaller than our $Md$.
>
> [1] Jiang and Li. Approximation Rate of the Transformer Architecture for Sequence Modeling. (2023)
>
> [2] Edelman et al. Inductive Biases and Variable Creation in Self-Attention Mechanisms. (ICML 2022)

---

> > ### Comment · Reviewer_iDXk · 2024-08-11
> >
> > Dear authors,
> >
> > I have read the rebuttal. Thank you for responding to my questions. I have noted that the authors will incorporate the conclusions of the discussions with the reviewers in a revised version, and I have decided to maintain my score.
> >
> > Best,
> >
> > Reviewer iDXk

---

> > > ### Author Response · Authors · 2024-08-12
> > >
> > > We are delighted that our responses have addressed your questions. Thank you very much for your support!

---

### Official Review · Reviewer_xF1Y · 2024-07-15

**Soundness:** 3
**Presentation:** 4
**Contribution:** 4
**Rating:** 7
**Confidence:** 3

**Summary:**

The paper provides a thorough analysis of the expressive power of Transformers. It does so by investigating the relative importance of different architectural components, such as the mixing layer (dot product attention), the feed-forwad block and positional embeddings. More specifically, the paper studies three different tasks, characterised by different levels of complexity, sharing some key properties with relevant realistic applications such as n-gram models, sparse Boolean functions (Task I), Adaptive sparse Boolean functions, parsing, part-of-speech tagging (Task II), feature extraction in image and signal processing (Task III). On each of such tasks, the paper studies the approximation rates of transformer models defined in terms of their number of layers $L$ and heads $H$, the width of the feed-forward module $m$ and the type of relative positional embedding utilised (i.e. either Alibi or T5 style). Through its theoretical analysis the paper shows that the necessary model's complexity needs to adapt to the complexity of the task. Moreover by investigating the role of different components, the paper shows that dot-product attention is necessary for more complex tasks (Task II) while its dot-product free variant provably fails. Nevertheless, the authors show that the task can still be solved with a more parameter efficient alternative. In addition, the role of the feed-forward blocks is mainly to approximate readout functions and memory functions, aligning with classical results on approximating Barron functions. The theoretical results are corroborated with empirical results.

**Strengths:**

* The paper meaningfully contributes to the very relevant research line on theoretical analysis of transformers models.
* The paper originally addresses important questions, such as the relative role of different components in the transformer architecture and the necessity of the dot-product attention, through a principled approach and in a controlled framework.
* Generally the paper is very well written (modulo some sparse and minor typos) and pleasant to read.

**Weaknesses:**

* Some concepts could be better introduced: for example, it was not clear to the reviewer what the authors meant by "memories" in the introduction of the paper. While this gets clearer later, it would be helpful to clarify it earlier.
* Layer normalization, an important component of modern transformers, does not seem to be included in the analysis. What effect do you expect it to have on your results?
* The role of the embedding dimension $D$ is not considered in the reported rates. What is the effect of varying $D$ on the approximation rates of the model?

**Questions:**

See weaknesses section.

**Limitations:**

The limitations of the paper are discussed in section 7

---

> ### Author Rebuttal · Authors · 2024-08-04
>
> We appreciate the reviewer's recognition of our work and helpful comments. Below, we offer detailed responses to the reviewer’s questions:
>
> - **W1.** Some concepts could be better introduced: for example, it was not clear to the reviewer what the authors meant by "memories" in the introduction of the paper. While this gets clearer later, it would be helpful to clarify it earlier.
>
>     **Response:** Thank the reviewer for this helpful suggestion. In our paper, "memories" generally refer to the sparse "tokens" that sequence modeling relies on. For example, for the simplest Task I (sequence modeling with fixed memories), the number of memories is $M$, and they are located at fixed positions $t_1,\cdots,t_M$ (which can be very large). In the revised version, we will introduce this concept in Section 1 to enhance clarity.
>
>
> - **W2.** Layer normalization, an important component of modern transformers, does not seem to be included in the analysis. What effect do you expect it to have on your results?
>
>   **Response:** We thank the reviewer for this insightful question.
>   - First, we would like to clarify that *the primary reason for not including Layer normalization (LN)* in our analysis is: LN is typically employed to accelerate and stabilize the training of Transformers, whereas our paper focuses on the issue of expressiveness. In fact, previous works about the expressiveness of Transformers have also omitted LN [1][2].
>   - Now we discuss the *potential impact of LN on our results*. From an approximation theory perspective, LN introduces certain nonlinearity. However, our Transformers include FFN layers, which are already capable of effectively approximating nonlinear functions. Therefore, incorporating LN seems unlikely to significantly improve the network's expressiveness. Technically, LN introduces a coordinate-wise normalization operation, which cannot be analyzed using our current methodology. We leave this analysis for future work.
>
> - **W3.** The role of the embedding dimension $D$ is not considered in the reported rates. What is the effect of varying $D$ on the approximation rates of the model?
>
>   **Response:** We thank the reviewer's insightful question. To clarify, let the dimension of a single token be $d$, the input sequence length be $T$ (where $T=\infty$ in our paper) the number of memories be $M$ ($M\ll T$), and the embedding dimension of Transformer be $D$.
>
>   - *Conjecture: increasing the embedding dim $D$ can effectively reduce the requirement on the number of layers but may not decrease the total number of parameters needed.*
>
>   - *Theoretical evidence (Comparison with [3]).* In our paper, we consider embedding dim $D_1\sim M d$, which is notably independent of the sequence length $T$ and almost the minimum embedding dim required to extract $M$ $d$-dim memories simultaneously. In contrast, in [3], the authors consider a large embedding dim $D_2\sim T^2 d$ ($D_2\gg D_1$ due to $T\gg M$), and prove that, in this setting, 1-layer Transformers are sufficient to perform sequence modeling with sequence length $T$, whereas our results require multiple layers (for Task II).
>
>   - *Empirical evidence:* the "scaling law" (see Fig. 6 in [4]) partially supports this conjecture: 1-layer Transformes indeed demonstrate the scaling law. However, with the same total number of parameters, 1-layer Transformers (with larger $D$) slightly underperform compared to 6-layer Transformers (with smaller $D$). Additionally, [5] highlights that for solving iGSM math problems, given the same total number of parameters, deep Transformers (with smaller $D$) perform much better than shallow Transformer (with larger $D$).
>
>   We will discuss this open issue in Section 7 in the revised version and leave further analysis for future work.
>
> [1] Yun et al. Are transformers universal approximators of sequence-to-sequence functions? (ICLR 2020)
>
> [2] Bai et al. Transformers as statisticians: Provable in-context learning with in-context algorithm selection. (NeurIPS 2023)
>
> [3] Jiang and Li. Approximation Rate of the Transformer Architecture for Sequence Modeling. (2023)
>
> [4] Kaplan et al. Scaling Laws for Neural Language Models. (2020)
>
> [5] Ye et al. Physics of Language Models: Part 2.1, Grade-School Math and the Hidden Reasoning Process. (2024)

---

> > ### Author Response · Authors · 2024-08-12
> >
> > Thanks again for your valuable time and effort in reviewing our work!
> >
> > We are wondering if our responses address your questions or concerns.
> >
> > We are happy to try to address any other comments in the time remaining.

---

> > ### Comment · Reviewer_xF1Y · 2024-08-14
> >
> > Dear authors,
> >
> > I have read the rebuttal. Thank you for your work and the answers to my concerns. I am satisfied with the authors' reply and will keep my score.
> >
> > Best,
> > Reviewer xF1Y

---

> > > ### Author Response · Authors · 2024-08-14
> > >
> > > We thank the reviewer for the positive feedback of our response. We appreciate your support very much!

---

### Author Rebuttal · Authors · 2024-08-04

### **Global Response to All Reviewers.**

- First, we sincerely thank all the reviewers for their appreciation of our results, i.e., theoretical analysis of the expressive power of Transformer for sequence modeling. Our analysis provides valuable insights (also supported by experiments) into the underlying mechanisms of Transformer components, including:
  - The distinct roles of the number of layers ($L$), the number of self-attention (Attn) heads ($H$), and the width of feed-forward network (FFN) layers ($m$).
  - The different roles of Attn layers and FFN layers.
  - The functionality and necessity of dot-product (DP).
  - The efficiency of relative positional encoding (RPE) in modeling long-range correlations.

- We also express our gratitude to the reviewers for their comments and suggestions for improving our paper. In the revised version, we will correct all typos, include the proof sketches, provide complete experimental settings and results, and incorporate the discussions with the reviewers.

- The attached PDF reports a new experimental result addressing Reviewer Fdzw's Question 2 regarding the tightness of our bounds.

- We have addressed each concern raised by the reviewers through separate responses provided below.

---

### Decision · Program_Chairs · 2024-09-25

**Decision:**

Accept (poster)

**Comment:**

This paper is a detailed analysis of the memory mechanisms of transformers. Its analysis relies on the notion of M-sparse memories, which are specific targets of the models, and then approximation properties are studied.

I agree with the reviewers that, despite being simplified, this analysis leads to some interesting insights. The paper is mostly well written: I'd recommend to strengthen its theoretical motivation in the final version of this work.